# FedInverse: Evaluating Privacy Leakage in Federated Learning

**Di Wu**[1][*][†]**, Jun Bai**[2][*]**, Yiliao Song**[3][*][†]**, Junjun Chen**[4]**, Wei Zhou**[5]**, Yong Xiang**[2]**, Atul Sajjanhar**[2]

School of Mathematics, Physics and Computing, University of Southern Queensland[1]
School of Information Technology, Deakin University[2]
The University of Adelaide[3]
Computer Center, Peking University[4]
School of Science, Computing and Engineering Technologies, Swinburne University of Technology[5]
`di.wu@unisq.edu.au,{baijun,yong.xiang,atul.sajjanhar}@deakin.edu.au`
`lia.song@adelaide.edu.au,chenjunjun@pku.edu.cn,weizhou@swin.edu.au`

## Abstract

Federated Learning (FL) is a distributed machine learning technique where multiple devices (such as smartphones or IoT devices) train a shared global model by using their local data. FL promises better data privacy as the individual data isn't shared with servers or other participants. However, this research uncovers a groundbreaking insight: a model inversion (MI) attacker, who acts as a benign participant, can invert the shared global model and obtain the data belonging to other participants. In such scenarios, distinguishing between attackers and benign participants becomes challenging, leading to severe data-leakage risk in FL. In addition, we found even the most advanced defense approaches could not effectively address this issue. Therefore, it is important to evaluate such data-leakage risks of an FL system before using it. Motivated by that, we propose *FedInverse* to evaluate whether the FL global model can be inverted by MI attackers. In particular, FedInverse can be optimized by leveraging the Hilbert-Schmidt independence criterion (HSIC) as a regularizer to adjust the diversity of the MI attack generator. We test FedInverse with three typical MI attackers, GMI, KED-MI, and VMI. The experiments show that FedInverse can effectively evaluate the data leakage risk that attackers successfully obtain the data belonging to other participants. The code of this work is available at https://github.com/Jun-B0518/FedInverse

## 1 Introduction

Federated Learning (FL) is a machine learning technique where multiple participants collaborate to train a global model on a central server while keeping their data locally on devices Li et al. (2020); Kairouz et al. (2021). FL has been applied to many real-world applications such as medical informatics Salim & Park (2023), the Internet of Things Nguyen et al. (2021), and mobile edge computing Lim et al. (2020) because FL is advanced in solving data isolation problems and user privacy-preserving Nguyen et al. (2022). For example, FL is leveraged to preserve patient data privacy for medical informatics Salim & Park (2023). FL claims itself to be able to naturally protect user privacy as each participant trains the model locally McMahan et al. (2017), *i.e.*, different participants do not need to share their private data. In fact, FL might be still vulnerable with regard to its capability for privacy protection Zhu et al. (2019). A few studies found that attackers could hijack the gradients when benign participants communicate with the server Huang et al. (2021) or masquerade as a server to receive gradients uploaded from participants Geiping et al. (2020) to reveal the local data of other participants.

However, no studies discuss whether attackers can reveal data from other participants via *a participant role*. This will lead to severe data-leakage risk in FL because it is difficult to identify attackers from benign participants if the attacker plays a participant role. In this paper, we identify and evaluate this *undiscovered* but *more severe* privacy leakage issue in FL that an attacker, who acts as

---

[*]Equal contribution. [†]Corresponding author:di.wu@unisq.edu.au, lia.song@adelaide.edu.au

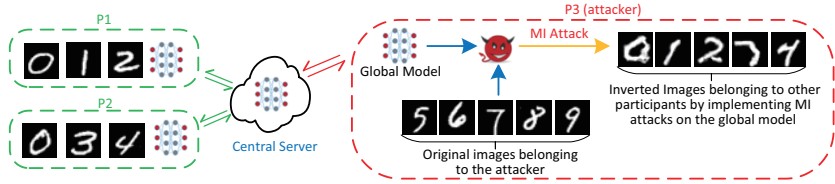

Figure 1: MI attacker can cause FL data-leakage. P1, P2, and P3 denote three FL participants, respectively, wherein P3 is an MI attacker. At the beginning, P3 only has images of digits 5 to 9, however, P3 can invert images of digits 0 to 4 from P1 and P2 through the MI attacks.

a benign participant, can reveal data from other participants with fewer conditions. Motivated by that, we draw attention to the recently developed model inversion (MI) attacks that can expose sensitive private information directly from well pre-trained models Fredrikson et al. (2015); Khosravy et al. (2022); Kahla et al. (2022); Rigaki & Garcia (2020). We let MI attacks act as benign participants and aim to reveal the private image of other participants. Specifically, we use three typical MI attacks—Generative MI attack (GMI) Zhang et al. (2020), Knowledge-Enriched Distributional MI attack (KED-MI) Chen et al. (2021), and Variational MI attack (VMI) Wang et al. (2021a), and find they can successfully recover images of other participants from FL global models. As is shown in Figure. 1, P3 (the attacker) recovers images belonging to the benign participants P1 and P2.

Knowing that FL undergoes such data-leakage risk by MI attacks (Figure. 1), we question *if the current defense approach can address this issue*. We test two advanced defense approaches, MID Wang et al. (2021b) and BiDO Peng et al. (2022) in FL settings, on the three typical datasets MNIST, CelebA, and CIFAR-10, respectively. The results show that these SOTA defense methods cannot defend against the attacks on MNIST and CIFAR-10. We also observe that the FL and defense performance have a significant trade-off on the different parameter settings, indicating that SOTA MI-defense approaches are data-oriented and thus not always applicable to FL (See Appendix A.(7,8,14,15,18,19)).

Given that the data-leakage risk cannot be eliminated, we propose *FedInverse*, a novel privacy leakage evaluation method to evaluate the boundaries of privacy protection in the FL system from the participant's perspective, whether one participant can obtain data from other participants. An attacker pretending to be a participant in FedInverse can conduct the Black-box attacks via global model prediction in each federated training round. In addition, MI attacks sometimes obtain less diverse data when inverting the model, making it challenging to evaluate the data-leakage risk. To verify the efficacy of the attack performance, we propose a dependency constraint in FedInverse by introducing the *Hilbert-Schmidt independence criterion* (HSIC) Gretton et al. (2005) to adjust the diversity of the attacker-generated images Radford et al. (2015).

We test FedInverse with three typical MI attackers, GMI, KED-MI, and VMI on two typical datasets including CelebA Liu et al. (2015) and MNIST LeCun et al. (1998). The experimental results show that FedInverse can effectively evaluate the data leakage risk that attackers successfully obtain the data belonging to other participants. Specially, GMI, KED-MI, and VMI can achieve *high attack performance on target global models* including VGG16 Simonyan & Zisserman (2014), ResNet-34 He et al. (2016), and MCNN Cui et al. (2016) and reveal the data from other participants. By the end, we compare the performance of FedInverse with and without HSIC, and find that the *attack performance improves* significantly when *increasing the diversity* of images that attackers generated.

## 2 PRELIMINARY

### 2.1 TRAINING PROCEDURE OF FEDERATED LEARNING

FL has made significant benefits to the fields of the Internet of Things Savazzi et al. (2020), network security Chen et al. (2022), and medical care Huang et al. (2019), but it faces the challenge that the global model in FL can be poisoned by uploading malicious parameters to the server Zhang et al. (2019); Zhu et al. (2019). However, no studies pay attention to the data leakage problem when the attackers are pretended to be benign users. This paper mainly discusses the vulnerability of federated learning from the perspective of attackers obtaining user privacy as the FL participants.

Unlike traditional server-client training schemes, in each training round of FL, $k$ clients are selected as participants and receive the global model $\omega_t$. All the participants train the model parameters on

their own local private data and return trained parameters $\omega_t^k$ back to the server. The server averages all the parameters $\omega_t^k$ and constructs the new global model $\omega_{t+1}$ for the next training round. In the FL training mechanism, every participant contributes local updates $\omega_t^k$ trained by their sensitive local data. Even though the new global model $\omega_{t+1}$ is generated after the average algorithm, it still contains the key information from all the participants which could be a good target for MI attacks. This paper aims to evaluate and quantify the risk of sensitive information being leaked in FL by MI attacks.

## 2.2 Model Inversion Attacks

According to different attack strategies, privacy attacks on machine learning can be divided into membership inference attacks Shokri et al. (2017), parameter extraction attacks Ateniese et al. (2015), and model inversion attacks Liu et al. (2020). This paper mainly focuses on model inversion attacks. The first model inversion attack was proposed by Fredrikson et al. (2014) which demonstrates that even if an attacker only has access privilege to the global model, it is possible to obtain users' sensitive data. Hitaj et al. Hitaj et al. (2017) proposed a model inversion attack in collaborative learning scenarios showing that as long as the local model accuracy of the participant is high, a good attack performance can be achieved. Ateniese et al. Ateniese et al. (2015) constructed a new meta-classifier (meta-classifier) and trained it to attack other classifiers to obtain sensitive information about their training data sets. Wang et al. Wang et al. (2019) proposed a model inversion attack for FL. This method designed a multi-task generative confrontation model as the attack model and successfully realized the user-level privacy attack. Recent model inversion attack are optimization-based methods, such as GMI Zhang et al. (2020), KED-MI Chen et al. (2021), and VMI Wang et al. (2021a), which obtain private data in the global model by training GAN. The details of these attacks will be described in the next section.

## 2.3 Hilbert-Schmidt Independence Criterion

HSIC Gretton et al. (2005) is a kernel-based measure to evaluate the statistical dependence between various random variables. Let $\phi : \mathcal{X} \to \mathcal{F}$ represent a nonlinear feature transformation and $k_x(x, x') = \langle \phi(x), \phi(x') \rangle$ denote a positive kernel function showing the inner product between features. So the structure of a reproducing kernel Hilbert space (RKHS) can be represented by feature space $\mathcal{F}$. We can also define another transformation $\psi : \mathcal{Y} \to \mathcal{G}$ and the corresponding positive definite kernel function $k_y(y, y') = \langle \psi(y), \psi(y') \rangle$, which has a similar process with the former transformation. Then, a cross-covariance operator $\mathcal{C}_{xy} : \mathcal{G} \to \mathcal{F}$ between the two transformations exists and can be defined linearly in the following equation:

$$\mathcal{C}_{xy} = E_{xy} \left[ (\phi(x) - \mu_x) \otimes (\psi(y) - \mu_y) \right],  \quad (1)$$

wherein $\otimes$ denotes a tensor product between vector space $\mu_x = E_x[\phi(x)]$ and vector space $\mu_y = E_y[\psi(y)]$. Then HSIC, which is a squared norm of the cross-covariance operator $\mathcal{C}_{xy}$, can be represented as

$$\text{HSIC}(\mathcal{F}, \mathcal{G}, P_{xy}) = \|\mathcal{C}_{xy}\|_{HS}^2  \quad (2)$$

Given $m$ pairs of data sets $Z = \{(x_1, y_1), \ldots, (x_m, y_m)\}$ from datasets $X \in R^{m \times d_x}$ and $Y \in R^{m \times d_y}$, the empirical estimator of HSIC can be written as

$$\text{HSIC}(\mathcal{F}, \mathcal{G}, P_{xy}) = \frac{1}{m^2} \text{Tr}(K_x H K_y H),  \quad (3)$$

wherein $m \times m$ is the size of the empirical HSIC, $K_x$ and $K_y$ represent the corresponding kernel matrices for $x$ and $y$ with $(k_x)_{i,j} = k_x(x_i, x_j)$ and $(k_y)_{i,j} = k_y(y_i, y_j)$, Tr is the trace of the matrix, and $H$ centers $x$ and $y$ in feature space $\mathcal{F}$ and $\mathcal{G}$.

## 3 Methodology

This section first introduces the proposed FedInverse method and how it embeds MI attackers. We have shown cases of FedInverse using three typical MI attackers—Generative MI (GMI) Zhang et al. (2020), Knowledge-Enriched Distributional MI (KED-MI) Chen et al. (2021), and Variational MI (VMI) Wang et al. (2021a), which will be presented separately in the following sections. Then, we

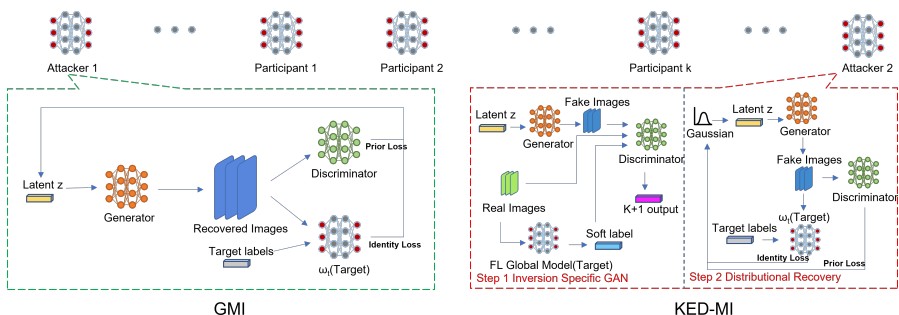

Figure 2: FedInverse: Model inversion attacks against FL, taking GMI and KED-MI as examples. VMI is not shown in this figure because it has a similar training schema to KED-MI.

introduce optimized FedInverse which leverages the Hilbert-Schmidt independence criterion (HSIC) as a regularizer to increase the diversity of MI attack generators and thus improve the evaluation performance.

## 3.1 FedInverse

The main idea of FedInverse method is presented in Figure. 2, where Attackers 1 (GMI) and Attacker 2 (KED-MI) are pretending to be benign participants to get the necessary information from the central server, *i.e.* pre-trained models, training tasks, target labels, etc. In the meantime, Attacker 1 and 2 also pre-train GAN models by leveraging public images that are structurally similar to the target images to launch an attack in a white-box setting. Clearly, we can obtain qualified public images by using historical target images. Next, we will analyze the attacker case by case. Attacker 1 is GMI and Attacker 2 is KED-MI. We will also analyze Attacker 3: VMI , which has a similar scheme to KED-MI, and thus is not shown in Figure. 2.

**Attacker 1: GMI attack against FL.** To reconstruct the sensitive image from other participants, Attacker 1 utilizes public structurally similar images to train a Wasserstein-GAN Arjovsky et al. (2017) as shown in Equation.4, which learns a generic prior knowledge.

$$\min_{G} \max_{D} L_{wgan}(G, D) = \mathbb{E}_x[D(x)] - \mathbb{E}_z[D(G(z))]. \tag{4}$$

The goal of Attacker 1 is to find the latent vector $z'$ that maximizes the likelihood under the FL global model $\omega_t$ limited to the data manifold learned by G as $z' = \arg\min_z L_{prior}(z) + \lambda_i L_{id}(z)$. Prior loss $L_{prior}(z)$ penalizes unrealistic images and the identity loss $L_{id}(z)$ promotes the generated images to have a high likelihood under $\omega_t$. Equation.5 gives the details of $z'$.

$$z' = \arg\min_{z}(-D(G(z))) + \lambda_i(-log[F_{\omega_t}(G(z))]), \tag{5}$$

wherein $F_{\omega_t}(G(z))$ indicates the probabiltiy of $G(z)$ output by the FL global model $\omega_t$.

**Attacker 2: KED-MI Attack against FL.** To distill better private information from other FL participants, Attacker 2 launches the attack in two steps. In the first step, a customized GAN is trained. To adopt the discriminator $D$ that can discriminate the class labels under FL global model $\omega_t$, discriminator $D$ includes $(K + 1)$ classes, where $K$ classes correspond to the labels of the $\omega_t$, and $(K + 1)$-th class indicates fake samples. A soft label $F_{\omega_t}(x)$ is generated for each image from the public set. The training loss for $D$ is represented in Equation.6.

$$L_D = -\mathbb{E}_{x \sim p_{\text{data}}(x)}[\Sigma_{k=1}^{K} F_{\omega_t^k}(x) \log p_{\text{disc}}(y = k \mid x)]$$
$$-\{\mathbb{E}_{x \sim p_{\text{data}}(x)}[\log D(x)] + \mathbb{E}_{z \sim p_{\text{noise}}}[\log(1 - D(G(z)))]\}, \tag{6}$$

wherein $p_{data}$ represents the distribution of public structurally similar images, and $p_{disc}(y \mid x)$ indicates the probability that $D$ predicts $x$ as class $y$. $F_{\omega_t^k}(x)$ is the $k$-th dimension of the soft label produced by the global model $\omega_t$. The training loss of generator $G$ is illustrated in Equation.7.

$$L_G = \|\mathbb{E}_{x \sim p_{\text{data}}}[\mathbf{f}(x)] - \mathbb{E}_{z \sim p_{\text{noise}}}[\mathbf{f}(G(z))]\|_2^2 + \lambda_h L_{\text{entropy}}, \tag{7}$$

wherein $\mathbf{f}(x)$ is the learned features encoded in an intermediate layer of the discriminator and $L_{entropy}$ is an entropy regularizer Grandvalet & Bengio (2004).

---

**Algorithm 1** *FedInverse Algorithm.* $K$ indicates the number of participants and $k$ represents the participant number; $B$ represents the local batch size, $E$ indicates the local training epochs, $C$ is the participation rate of participants, while $\eta$ is learning rate; $G$ and $D$ denote Generator and Discriminator respectively, $\mathcal{P}_{aux}$ represents the auxiliary dataset used to pre-train GAN, $\mathcal{N}$ denotes the Gaussian distribution, while $\mathcal{Q}_t$ indicates the set of generated images by FedInverse.

---

| | |
|---|---|
| 1: Server Initialization: $\omega_0$ | 20:          $x \leftarrow G(z)$ |
| 2: **for** each training round $t$ = 1,2 ... **do** | 21:          split $x$ into $x_1$ and $x_2$ |
| 3:      $m \leftarrow \max(C \cdot K, 1)$ | 22:          compute HSIC($x_1$, $x_2$) |
| 4:      $S_t \leftarrow$ (random set of $m$ participants including a single Attacker) | 23:          update $z'$ for diversity optimization |
| | 24:       **end for** |
| 5:      **for** each participant $k \in S_t$ **in parallel do** | 25:      $x' \leftarrow G(z')$ |
| 6:          $\omega_{t+1}^k \leftarrow$ ParticipantUpdate($k, \omega_t$) | 26:      $\mathcal{Q}_t \leftarrow \mathcal{Q}_t \cup \{x'\}$ |
| 7:          **evaluate** on $\mathcal{Q}_t \leftarrow$ Attacker($\omega_t$) | 27:     **end for** |
| 8:      **end for** | 28:      return $\mathcal{Q}_t$ |
| 9:      $\omega_{t+1} \leftarrow \Sigma_{k=1}^K \frac{n_k}{n} \omega_{t+1}^k$ | 29: **end function** |
| 10: **end for** | 30: |
| 11: | 31: **function** ATTACKER($\omega_t$): |
| 12: **function** ATTACKER($\omega_t$): | 32:      $\mathcal{B} \leftarrow$ (split $\mathcal{P}_k$ into batches of Size $B$) |
| 13:      **if** needed **then** | 33:      **for** each local epoch $i$ from 1 to $E$ **do** |
| 14:          pretrain $G$ and $D$ with $\omega_t$ on $\mathcal{P}_{aux}$ | 34:          **for** batch $b \in \mathcal{B}$ **do** |
| 15:      **else** | 35:              $\omega_t \leftarrow \omega_t - \eta \nabla l(\omega_t; b)$ |
| 16:          load pretrained $G$ and $D$ | 36:          **end for** |
| 17:      **end if** | 37:      **end for** |
| 18:      **for** each attack epoch **do** | 38:      return $\omega_t$ to server |
| 19:          **for** batch $z \in \mathcal{N}$ **do** | 39: **end function** |

---

To launch the attack, given a class label $k$ from the global model $\omega_t$, the attack loss is $L = L_{prior} + L_{id}$, which is similar to GMI, and the details are represented in Equation.8.

$$L = -\mathbb{E}_{z' \sim p_{\text{gen}}}[\log D\left(G\left(z'\right)\right)] - \mathbb{E}_{z' \sim p_{\text{gen}}}[F_{\omega_t^k}\left(G\left(z'\right)\right)], \tag{8}$$

wherein $z'$ is sampled from $p_{gen} = \mathcal{N}(\mu, \sigma^2)$. After reparameterization, $z'$ can be represented as Equation.9 to directly estimate $\mu$ and $\sigma^2$ through back-propagation.

$$z' = \sigma\epsilon + \mu, \epsilon \sim \mathcal{N}(0, I). \tag{9}$$

**Attacker 3: VMI attack against FL.** Attacker 3 demonstrates a similar attack procedure to Attacker 2. Compare to KED-MI, a StyleGAN is used for VMI attack due to its capability of "style mixing". The synthesis network can generate a "mixed" image when given a mixture of two **w**'s in the expanded **w** space, where **w** represents the mapping vector of the input **z** via $f : \mathbf{z} \rightarrow \mathbf{w}$.

The objective of VMI with StyleGAN is shown in Equation.10

$$\begin{aligned} L_{\text{S}-\text{VMI}}^{\gamma}(q) :=& \mathbb{E}_{q(\mathbf{z}_1,...,\mathbf{z}_L)}\left[-\log F_{\omega_t}\left(y \mid S\left(\{f(\mathbf{z}_l)\}_{l=1}^L\right)\right)\right] \\ &+ \frac{\gamma}{L}\Sigma_{l=1}^L D_{\text{KL}}\left(q_l(\mathbf{z}_l) \| p_{aux}(\mathbf{z}_l)\right), \end{aligned} \tag{10}$$

wherein $q(z_1, ..., z_L)$ is the joint density over $z_1, ..., z_L$, and $q_l(z_l)$ is the marginal density over $z_l$. Additionally, $p_{aux}$ represents the auxiliary data used to pre-train GAN.

### 3.2 FEDINVERSE WITH DIVERSITY OPTIMIZATION

It is more difficult to invert the model when the MI attacks obtain less diverse images. To alleviate this issue, we use the Hilbert-Schmidt independency criterion (HSIC) as a regularizer with MI attack (termed MI-HSIC) to adjust the diversity of the attack generator, as shown in Figure 3. For each attack epoch, the attack generator will generate $x$ images from $G(z)$ based on the latent $z$ from data distribution $\mathcal{Q}_t$, we equally split $x$ into $x_1$ and $x_2$ and compute HSIC($x_1$, $x_2$) which is used to update the $z' \in \mathcal{Q}'_t$ for diversity optimization. The optimized attack loss $L = L_{prior} + L_{id} + HSIC(x_1, x_2)$ which is shown in Equation.11.

$$L = L_{prior} + L_{id} - \lambda\Sigma_{j=1}^m d(x_1, x_2), \tag{11}$$

wherein $m$ is half of the batch size of the generated images, and $d$ is the dependency measure. To evaluate the relationship between HSIC and the diversity of generated images, we can adjust the

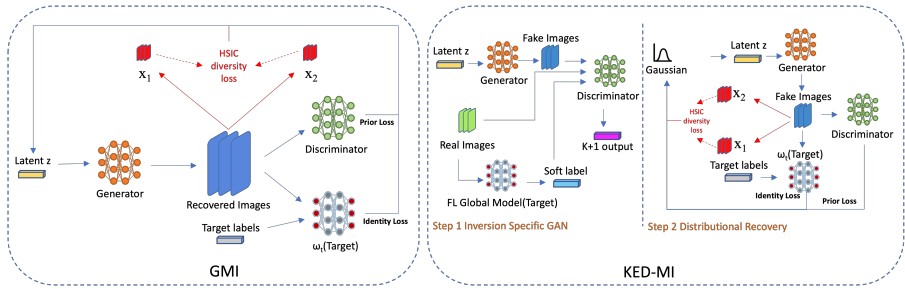

Figure 3: FedInverse with Diversity Optimization by Using HSIC on GMI and KED-MI.

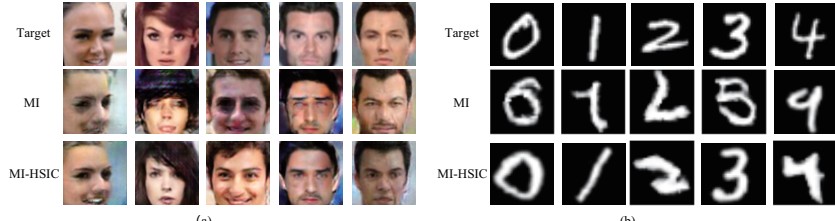

Figure 4: (a) Qualitative comparison of the proposed MI-HSIC attack with the corresponding MI attack against FL on CelebA. (b) Qualitative comparison of the proposed MI-HSIC attack with the corresponding MI attack against FL on MNIST.

parameter $\lambda$ to optimize the performance of generated image diversity. The algorithm of FedInverse is shown in Algorithm.1.

Figure. 4(a) and Figure. 4(b) compare some sample images generated from CelebA and MNIST respectively. According to Figure. 4(a), the generator on MI has lower quality on the generated images compared to MI-HSIC, *i.e.*, regarding the generating quality in the fourth column from the left side, MI-HSIC is better than MI itself. Even the inverted image cannot be completely alike the original image. However, the biometric identity (e.g. face) can be successfully reconstructed to break into otherwise secure systems even if humans do not recognize the model-inverted examples look much alike the true examples Wang et al. (2021a). Figure. 4(b) compares different generation performance between MI and MI-HSIC on MNIST. The first line indicates the target images from MNIST digits 0 to 4, and the attackers have the prior images from MNIST digits 5 to 9. The result shows that MI has less diversity compared with MI-HSIC, which means the generator always has a bias on the prior images with attackers, *i.e.* the generator with MI generates the digit 4 as digit 9. In conclusion, the HISC can affect the generating diversity and quality for MI attacking purposes.

## 4 EXPERIMENT SETTINGS

In this section, we discuss the experimental settings for verifying the efficacy of FedInverse against FL in white-box settings. We do not focus on black-box attacks because in this study the attacker plays a participant role which can naturally obtain the model structure. Therefore, the training task cannot be in a black-box setting.

### 4.1 DATASETS

We use three typical datasets, CelebFaces Attributes Dataset (CelebA) Liu et al. (2015), MNIST dataset LeCun et al. (1998), and CIFAR-10 Krizhevsky et al. (2009) (More experiment results on different datasets See Appendix A) to evaluate the FedInverse attack performance with different classification tasks. The three datasets cover, face recognition(CelebA), handwriting recognition (MNIST), and object detection (CIFAR-10). For all datasets, we randomly select a part as the historical target images and use these images as the prior public dataset to pre-train the GAN, as we have discussed in Section 3.1. Specifically, for CelebA, we first randomly select 1000 identities and select all images belonging to these identities from CelebA. These images will be used as private data in FL. In this paper, we finally have 30,027 images in the private data set. In addition, we also randomly select 30,000 images from the rest part of CelebA. This data is used to pre-train the GAN. For MNIST, as it has fewer classes, we directly select the images from MNIST digits 5 to 9 as the prior public data and use images from MNIST digits 0 to 4 as the private data. CIFAR-10 has the same settings as MNIST, five classes (airplane, automobile, bird, cat, deer) are selected as the private data, and the rest of the classes (dog, frog, horse, ship, truck) are used as the prior public data.

### 4.2 FEDERATED LEARNING GLOBAL MODELS

We adopt different global models with different classification tasks to evaluate the performance of different FedInverse attacks. For CelebA (face recognition tasks), we apply VGG16 Simonyan & Zisserman (2014) to evaluate the performance of GMI and KED-MI attacks in FedInverse and apply RezNet-34 He et al. (2016) to evaluate the performance of VMI attacks in FedInverse. For MNIST(handwriting recognition tasks), we apply MCNN Cui et al. (2016) as the global model to evaluate the performance of GMI and KED-MI attacks.

### 4.3 FEDERATED LEARNING SETTINGS

We adopt different FL settings for CelebA and MNIST to evaluate FedInverse.

**For CelebA**, we choose 5 participants joining every training round. The local training batch size is 64 and the local training epoch is 50. We evaluate all the FedInverse —GMI, KED-MI, and VMI. We choose VGG16 as the FL global model for GMI and KED-MI, and ResNet34 for VMI, respectively. Every participant except the attacker averagely shares the private data set in FL training setting.

**For MNIST**, 100 participants are chosen to join the FL, However, only 10 out of 100 participants can join the training in each training round. The local training batch is 10, and the local training epoch is 5. We evaluate the FedInverse leveraging GMI and KED-MI. We choose MCNN as the FL global model. The FL data training setting is similar to CelebA.

### 4.4 EVALUATION METRICS

To evaluate the performance of FedInverse, we assess the extent of sensitive information regarding a target label that is disclosed through the synthesized images. Our evaluation approach involves both quantitative metrics and visual examination. The quantitative metrics utilized for evaluating the attack performance are presented below.

**Attack accuracy (Attack Acc)** To evaluate the attack effectiveness, we construct an "evaluation classifier" to identify the identities of the reconstructed images. The evaluation classifier is well-trained by using the whole dataset with a very high testing accuracy which achieved around 98%. These metrics evaluate the similarity of the generated samples to the target class. If the evaluation classifier exhibits high accuracy, the attack is considered successful. To guarantee a comprehensive and impartial evaluation, the evaluation classifier should be highly accurate across all classes.

**Fréchet inception distance (FID)** We utilize the commonly used FID metric Heusel et al. (2017) to evaluate the quality and diversity of reconstructed images. This metric allows us to gauge the level of detailed information that may be present in the reconstructed images. FID determines the likeness between real and fake images within the embedding space, which is based on the features of a convolutional neural network (*i.e.*, the evaluation classifiers in the defense task). Essentially, FID calculates the variances and means of the features, assuming a multivariate normal distribution, and compares the differences between them.

## 5 EXPERIMENT EVALUATIONS

### 5.1 FEDINVERSE ATTACK PERFORMANCE ON CELEBA

Face recognition is widely applied in different real scenarios for public security purposes. We evaluate and compare the efficacy of FedInverse attack methods with and without HSIC on CelebA for privacy leakage on face recognition data in FL.

The results of FedInverse using GMI and GMI-HSIC are illustrated in Figure 5(a), GMI-HSIC consistently achieves better attack performance and FID since the first FL training round, indicated by improvement of 10% of the attack accuracy, 2% of the top-5 attack accuracy, and FID. FL training accuracy has been increased and stabilized in the second round from 66.05 to 80.48, and it is worth noting that GMI-HSIC achieves the highest attack accuracy and top-5 attack accuracy, which outperforms GMI by 5% of the attack accuracy, 5% of the top-5 attack accuracy, and smaller FID. When compared with the GMI and GMI-HSIC, GMI can partially leak images from other participants, and the attack performance can be significantly improved by GMI-HSIC, in which the diversity of the generated images has been optimized by HSIC.

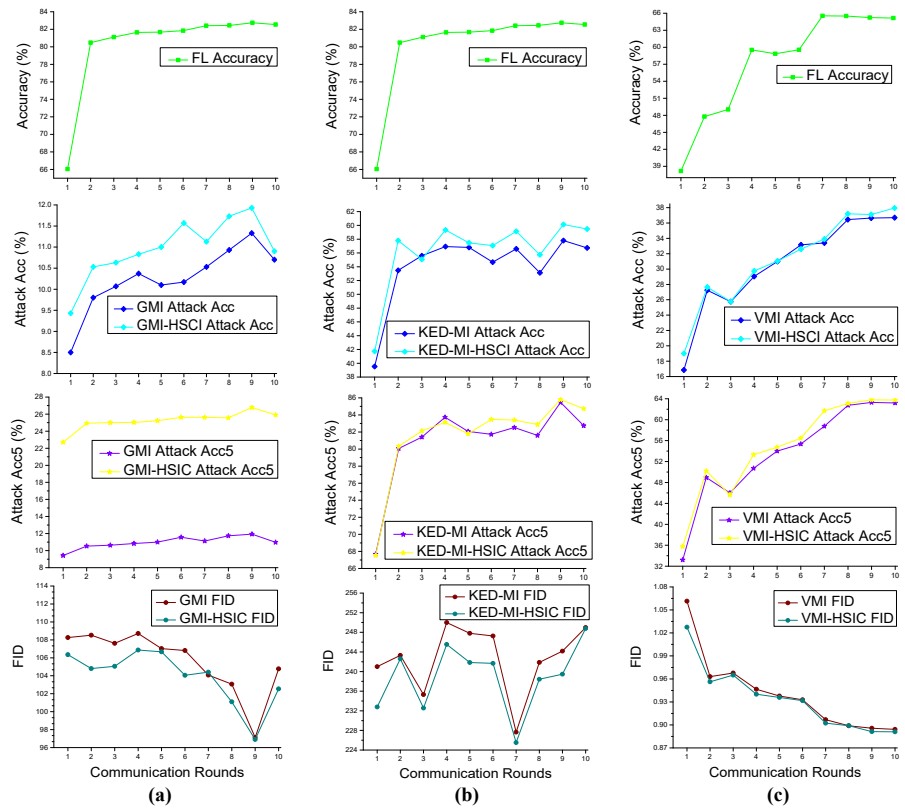

Figure 5: FedInverse on CelebA. Columns (a)-(c) present the relevant curves for three chosen MI/MI-HSIC attacks on CelebA under specific FL conditions. The first row of subplots illustrates global model accuracy changes over communication rounds. Rows two to four display comparative results using Attack Acc, Attack Acc5, and FID metrics for these attacks across ten federated rounds.

The performance evaluation of FedInverse using KED-MI attack on CelebA is presented in Figure 5(b), where we use the same attack settings as the GMI attack. Compared to GMI, KED-MI completely penetrates the mechanism of FL personal privacy protection, and the best attack accuracy dramatically increases from 11.33 in Figure 5(a) to 57.80 in Figure 5(b), and the best top-5 attack accuracy booms from 25.40 Figure 5(a) to 85.47 Figure 5(b). The best result of attack accuracy and top-5 attack accuracy also increases from 11.93 in Figure 5(a) to 60.13 in Figure 5(b), and 26.77 in Figure 5(a) to 85.80 in Figure 5(b), respectively. When we compare the attack performance between KED-MI and KED-MI-HSIC, KED-MI-HSIC improves 4% of the attack accuracy, 2% of the top-5 accuracy, and smaller FID in FL training rounds 9 and 10, respectively. According to the results, participants' privacy in FL can be easily revealed by the KED-MI, and HSIC plays an important role in increasing the diversity of the generated images where the attack accuracy can be improved.

We also evaluate the FedInverse performance of VMI in Figure 5(c) with ResNet-34. The highest attack accuracy of VMI appears in FL training round 10 is 36.70 and the highest top-5 attack accuracy appears in FL training round 9 is 63.30. The VMI-HSIC further improves the attack performance by 3% and the top-5 attack accuracy by 0.7%, which are 37.95 and 63.80, respectively, with smaller FID. The results validate the promised privacy leakage by VMI attacks.

## 5.2 FEDINVERSE ATTACK PERFORMANCE ON MNIST

Handwriting is another important user privacy information in real scenarios, therefore, we further evaluate the FedInverse using GMI and KED-MI attack performance on MNIST dataset in Table 1 and Table 2 respectively. As shown in Table 1, GMI performs better on MNIST than the attack performance on CelebA, which achieves the attack accuracy of 56.00 and the top-5 accuracy of 98.00 in FL training round 5. GMI-HSIC further improves 7% of the attack accuracy to 60.00 and 4% of the top-5 accuracy to 100.00 in FL training round 5 and 4 respectively. The results demonstrate that GMI performs better on the handwriting dataset. As shown in Table 2, the attack accuracy reaches

Table 1: FL privacy leakage indicated by Attack Acc/Acc5± standard deviation(%) and FID on MNIST via FedInverse using GMI and GMI-HSIC with prior training dataset MNIST. Bold values denote the best metric results obtained by GMI or GMI-HSIC throughout the FL training epoch. The symbol ↓(↑) denotes that smaller (larger) values are favored.

| Metrics | Methods | FL#R01 | FL#R02 | FL#R03 | FL#R04 | FL#R05 |
|---|---|---|---|---|---|---|
| Accuracy ↑ | | 83.34 | 97.59 | 98.27 | 98.4 | 98.52 |
| Attack Acc ↑ | GMI | 34.00±9.66 | 38.00±22.01 | 34.00±16.47 | 50.00±10.54 | **56.00±20.66** |
| | GMI-HSIC | 44.00±15.78 | 44.00±12.65 | 42.00±14.76 | 56.00±8.43 | **60.00±9.43** |
| Attack Acc5 ↑ | GMI | 94.00±9.66 | **98.00±6.32** | **98.00±6.32** | 96.00±8.43 | **98.00±6.32** |
| | GMI-HSIC | 96.00±8.43 | 98.00±6.32 | 98.00±6.32 | **100.00±0.00** | 98.00±6.32 |
| FID ↓ | GMI | 20.1373 | 23.3598 | 22.3839 | 17.1018 | **16.7486** |
| | GMI-HSIC | 19.0845 | 21.1116 | 21.5377 | 15.6066 | **14.469** |

Table 2: FL privacy leakage indicated by Attack Acc/Acc5± standard deviation(%) and FID on MNIST via FedInverse using KED-MI and KED-MI-HSIC with prior training dataset MNIST. Bold values denote the best metric results obtained by KED-MI or KED-MI-HSIC throughout the FL training epoch. The symbol ↓(↑) denotes that smaller (larger) values are favored.

| Metrics | Methods | FL#R01 | FL#R02 | FL#R03 | FL#R04 | FL#R05 |
|---|---|---|---|---|---|---|
| Accuracy ↑ | | 83.34 | 97.59 | 98.27 | 98.4 | 98.52 |
| Attack Acc ↑ | KED-MI | 64.60±8.46 | 60.60±4.45 | **80.00±0.00** | **80.00±0.00** | 79.80±2.00 |
| | KED-MI-HSIC | 80.00±0.00 | 64.40±8.33 | 80.00±0.00 | **80.20±2.00** | **80.20±2.00** |
| Attack Acc5 ↑ | KED-MI | 100.00±0.00 | 100.00±0.00 | 100.00±0.00 | 100.00±0.00 | 100.00±0.00 |
| | KED-MI-HSIC | 100.00±0.00 | 100.00±0.00 | 100.00±0.00 | 100.00±0.00 | 100.00±0.00 |
| FID ↓ | KED-MI | 209.1448 | 206.0789 | 195.1807 | 184.995 | **175.9532** |
| | KED-MI-HSIC | 204.5017 | 198.6938 | 175.9532 | 161.0252 | **160.9891** |

80.00 and the top-5 accuracy keeps 100.00 with KED-MI attacks on MNIST dataset, which are extremely high, and KED-MI-HSIC can still slightly improve the attack accuracy to 80.20 which is 0.2% of improvement to the attack accuracy.

## 6 RELATED WORKS

FL has made significant benefits to the fields of the Internet of Things Savazzi et al. (2020), network security Chen et al. (2022), and medical care Huang et al. (2019), but it also faces some challenges. No studies pay attention to the data leakage problem when the attackers pretend to be benign users and attack by the model inversion attacks. Typical model inversion attacks, including Zhang et al. (2020) Chen et al. (2021) Wang et al. (2021a), can be leveraged by the attacker to attack the FL system. In addition, to evaluate the impact on the diversity of the generated data, Hilbert-Schmidt independency criterion (HSIC) Gretton et al. (2005) is introduced, which is a statistic dependency measure metric that is well established in statistics. For more detailed related works please refer to Appendix B.

## 7 CONCLUSION

This is the first study that discovers model inversion (MI) attackers, who act as normal participants, can invert the FL global model and obtain the data belonging to other participants. This finding indicates a severe data-leakage risk in FL, especially considering FL claims itself to be naturally privacy-protected. To evaluate this data-leakage risk, we propose FedInverse that can evaluate whether MI attackers can invert the FL global model. We test FedInverse by leveraging three typical attackers, including GMI, KED-MI, and VMI on face recognition and handwriting recognition datasets. The experiment results show that the privacy-preserving mechanism of FL is vulnerable to MI attacks and it is difficult to prevent this risk if the attackers are acting as normal participants in FL. We further evaluate the efficacy of MI attacks with the diversity of generated images by using Hilbert-Schmidt independence criterion (HSIC) as the regularizer. The results prove that the attack performance is significantly improved when increasing the diversity of the generated images. Based on that, we consider promising future works for this topic would focus on further increasing the diversity of the generated images of an attacker and how to protect the user privacy from the MI attacks on FL.

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

# A APPENDIX

## A.1 EXPERIMENT IMPLEMENTATION DETAILS

We will elaborate more on the experiment implementation details in the appendix. We conducted
the FedInverse experiments on the extra datasets such as EMNIST Cohen et al. (2017), and Fashion-
MNIST (FMNIST) Xiao et al. (2017) rather than MNIST, CelebA and CIFAR-10 with different
attack settings. EMNIST is an expansion of the original MNIST dataset, encompasses a richer
variety of handwritten characters. It goes beyond digits to include letters, both lowercase and up-
percase, thereby offering a more comprehensive view of handwritten character recognition. The
dataset has been carefully partitioned into multiple subsets, each tailored to specific tasks — rang-
ing from digit recognition, balanced character sets, to datasets designed for by-class and by-merge
recognition tasks. With the inclusion of alphabetic characters, the complexity and variability in the
data increase, offering a more challenging playground for machine learning models. Additionally,
Fashion-MNIST is an alternative to the original MNIST digit dataset, curated to serve as a more
challenging problem in the domain of image classification. Designed by Zalando, a European e-
commerce company, the dataset contains grayscale images of 28x28 pixels each, encompassing 10
fashion categories, such as T-shirts, trousers, pullovers, dresses, and more. Each category is popu-
lated with 7,000 images, leading to a training set of 60,000 images and a test set of 10,000 images,
mimicking the exact structure of the classic MNIST.

## A.2 MORE EXPERIMENTAL RESULTS ON MNIST TYPES DATASETS

### A.2.1 IMPACT OF PRIOR DATA WITH FEDINVERSE+GMI ON EMNIST AND FMNIST

To assess the influence of prior data on the efficacy of GMI attacks on FL, we employ EMNIST
and FMNIST datasets as prior data for GMI, respectively. Pertinent findings from our empirical
investigations are presented in Table 3. The results of FedInverse attack using GMI and GMI-HSIC
are illustrated in Table 3, where we have the attack performance with varying GMI attack settings
and FL training rounds. As shown in Table 3, Similar to the results reported on MNIST and CelebA
in our paper, GMI-HSIC consistently achieves better attack performance and FID since the first FL
training round on EMNIST, indicated by improvement of the attack accuracy (60.00 vs 62.00), the
top-5 attack accuracy (92.00 vs 96.00), and FID (13.4053 vs 11.064). FL training accuracy has
been increased and stabilized in the second round from 83.34 to 98.52, and it is worth noting that
GMI-HSIC achieves the highest attack accuracy and top-5 attack accuracy, which outperforms GMI
as attack accuracy (70.00 vs 72.00), both reaches 100% top-5 attack accuracy and FID (6.3623 vs
5.5718). On FMNIST, the results show the same trend, in which GMI-HSIC outperforms the GMI
as as attack accuracy (52.00 vs 58.00), top-5 attack accuracy (98.00 vs 90.00), and FID (18.3514 vs
16.8608)

### A.2.2 IMPACT OF PRIOR DATA ON FEDINVERSE+KED-MI ON EMNIST AND FASHION-MNIST

To assess the influence of prior data on the efficacy of KED-MI attacks on FL, we employ EMNIST
and FMNIST datasets as prior data for KED-MI, respectively. Pertinent findings from our empir-
ical investigations are presented in Table 4. Compare to GMI, KED-MI completely penetrates the
mechanism of FL personal privacy protection on both EMNIST and FMNIST. For EMNIST, the best
attack accuracy dramatically increases from 70.00 in Table 3 to 88.15 in Table 4. The best result of
KED-MI-HSIC attack accuracy also increases from 72.00 in Table 3 to 99.99 in Table 4, where all
the best top-5 accuracy can achieve 100%. The results of FMNIST have a similar trend as EMNIST,

Table 3: FL privacy leakage indicated by Attack Acc/Acc5± standard deviation(%) and FID on MNIST via FedInverse using GMI and GMI-HSIC with two diverse prior training datasets: EMNIST and FMNIST. Bold values denote the best metric results obtained by GMI or GMI-HSIC throughout the FL training epoch. The symbol ↓(↑) denotes that smaller (larger) values are favored.

| Metrics | Methods | FL#R01 | FL#R02 | FL#R03 | FL#R04 | FL#R05 | FL#R01 | FL#R02 | FL#R03 | FL#R04 | FL#R05 |
|---|---|---|---|---|---|---|---|---|---|---|---|
| *Prior Data* | | EMNIST | | | | | FMNIST | | | | |
| Accuracy ↑ | | 83.34 | 97.59 | 98.27 | 98.40 | 98.52 | 83.49 | 97.87 | 98.36 | 98.73 | 98.94 |
| Attack Acc ↑ | GMI | 60.00±23.09 | 64.00±20.66 | 66.00±18.97 | 68.00±16.87 | **70.00±19.44** | 38.00±11.35 | 42.00±14.76 | 44.00±18.38 | 46.00±9.66 | **52.00±13.98** |
| | GMI-HSIC | 62.00±22.01 | 66.00±16.47 | 68.00±19.32 | 70.00±19.44 | **72.00±25.30** | 46.00±18.97 | 50.00±14.14 | 50.00±19.44 | 56.00±15.78 | **58.00±14.76** |
| Attack Acc5 ↑ | GMI | 92.00±10.33 | 94.00±9.66 | 96.00±8.43 | 98.00±6.32 | **100.00±0.00** | 82.00±14.76 | 84.00±15.78 | 88.00±10.33 | **90.00±10.54** | 90.00±10.54 |
| | GMI-HSIC | 96.00±8.43 | 98.00±6.32 | **100.00±0.00** | **100.00±0.00** | **100.00±0.00** | 84.00±15.78 | 88.00±10.33 | 90.00±10.54 | 92.00±10.33 | **98.00±6.32** |
| FID ↓ | GMI | 13.4053 | 11.3012 | 11.1458 | 9.9356 | **6.3623** | 20.9196 | 19.0771 | 18.6007 | 18.4105 | **18.3514** |
| | GMI-HSIC | 11.064 | 10.2641 | 9.9451 | 9.1239 | **5.5718** | 19.2656 | 17.9561 | 17.1093 | 17.6389 | **16.8608** |

in which the best attack accuracy dramatically increases from 52.00 in Table 3 to 86.83 in Table 4. The best result of KED-MI-HSIC attack accuracy also increases from 58.00 in Table 3 to 99.80 in Table 4. In addition, we also compare the attack performance between KED-MI and KED-MI-HSIC, KED-MI-HSIC improves the attack accuracy (88.15 vs 99.99), and smaller FID (127.9106 vs 116.5144) on EMNIST. and attack accuracy (86.83 vs 99.80), and smaller FID (192.0721 vs 185.9508) on FMNIST, respectively.

Table 4: FL privacy leakage indicated by Attack Acc/Acc5± standard deviation(%) and FID on MNIST via FedInverse using KED-MI and KED-MI-HSIC with two diverse prior training datasets: EMNIST and FMNIST. Bold values denote the best metric results obtained by GMI or GMI-HSIC throughout the FL training epoch. The symbol ↓(↑) denotes that smaller (larger) values are favored.

| Metrics | Methods | FL#R01 | FL#R02 | FL#R03 | FL#R04 | FL#R05 | FL#R01 | FL#R02 | FL#R03 | FL#R04 | FL#R05 |
|---|---|---|---|---|---|---|---|---|---|---|---|
| *Prior Data* | | EMNIST | | | | | FMNIST | | | | |
| Accuracy ↑ | | 83.34 | 97.59 | 98.27 | 98.40 | 98.52 | 83.49 | 97.87 | 98.36 | 98.73 | 98.94 |
| Attack Acc ↑ | KED-MI | 74.17±2.21 | 80.00±0.00 | 84.24±3.21 | 85.60±8.34 | **88.15±2.77** | 73.80±4.49 | 78.23±4.72 | 81.49±3.07 | 83.04±11.51 | **86.83±1.71** |
| | KED-MI-HSIC | 79.99±0.29 | 86.67±0.00 | 87.80±5.83 | 99.11±3.08 | **99.99±0.29** | 76.24±7.48 | 82.46±7.20 | 83.69±4.28 | 87.00±9.56 | **99.80±1.82** |
| Attack Acc5 ↑ | KED-MI | 84.76±8.33 | 98.07±4.57 | **100.00±0.00** | **100.00±0.00** | **100.00±0.00** | 100.00±0.00 | 100.00±0.00 | 100.00±0.00 | 100.00±0.00 | 100.00±0.00 |
| | KED-MI-HSIC | 92.12±6.90 | 99.12±3.59 | **100.00±0.00** | **100.00±0.00** | **100.00±0.00** | 100.00±0.00 | 100.00±0.00 | 100.00±0.00 | 100.00±0.00 | 100.00±0.00 |
| FID ↓ | KED-MI | 155.4555 | 151.3656 | 141.9426 | 139.4965 | **127.9106** | 235.5265 | 213.8955 | 210.1306 | 197.6505 | **192.0721** |
| | KED-MI-HSIC | 146.2689 | 140.6991 | 133.3661 | 116.8525 | **116.5144** | 223.582 | 202.285 | 199.2425 | 198.5916 | **185.9508** |

## A.3 IMPACT OF PARALLELISM ON FEDINVERSE+GMI ON MNIST

We investigate the influence of FL parallelism by modulating the active fraction of participants on the attack performance of GMI within the context of FL. Pertinent observations are detailed in Table 5. We adjust the numbers of the participants in FL settings from 10% to 100% to monitor the parallelism impact on attacks. We observe that the attack performance varies, and GMI has similar results when 10%, 20%, and 50% participants join each training round from 56.00 to 58.00. However, the attack performance decreased to 48.00 when 100% participants joined the FL round. The same trend happens on GMI-HSIC, the attack accuracy varies from 60.00 to 58.00 when 10%, 20%, and 50% participants join each training round and decrease to 49.20 when 100% participants join the FL round. In addition, the top-5 attack accuracy remains stable from 98.00 to 100.00 on GMI and keeps 100 on GMI-HSIC, respectively.

Table 5: FL privacy leakage indicated by Attack Acc/Acc5± standard deviation(%) and FID on MNIST via FedInverse using GMI and GMI-HSIC with varying active fractions of participants. Bold values denote the best metric results obtained by GMI or GMI-HSIC throughout the FL training epoch. The symbol ↓(↑) denotes that smaller (larger) values are favored.

| Metrics | Methods | FL#R01 | FL#R02 | FL#R03 | FL#R04 | FL#R05 | FL#R01 | FL#R02 | FL#R03 | FL#R04 | FL#R05 |
|---|---|---|---|---|---|---|---|---|---|---|---|
| *Fraction of Participants* | | $C = 0.1$ | | | | | $C = 0.2$ | | | | |
| Accuracy ↑ | | 83.34 | 97.59 | 98.27 | 98.40 | 98.52 | 83.49 | 97.87 | 98.36 | 98.73 | 98.94 |
| Attack Acc ↑ | GMI | 34.00±9.66 | 38.00±22.01 | 34.00±16.47 | 50.00±10.54 | **56.00±20.66** | 34.00±16.47 | 42.00±14.76 | 44.00±15.78 | 46.00±16.47 | **56.00±18.38** |
| | GMI-HSIC | 44.00±15.78 | 44.00±12.65 | 42.00±14.76 | 56.00±8.43 | **60.00±9.43** | 36.00±15.78 | 44.00±15.78 | 46.00±13.50 | 48.00±21.50 | **58.00±17.51** |
| Attack Acc5 ↑ | GMI | 94.00±9.66 | **98.00±6.32** | **98.00±6.32** | 96.00±8.43 | **98.00±6.32** | 96.00±8.43 | 96.00±8.43 | 98.00±6.32 | **100.00±0.00** | **100.00±0.00** |
| | GMI-HSIC | 96.00±8.43 | 98.00±6.32 | 98.00±6.32 | **100.00±0.00** | 98.00±6.32 | 96.00±8.43 | 98.00±6.32 | **100.00±0.00** | **100.00±0.00** | **100.00±0.00** |
| FID ↓ | GMI | 20.1373 | 23.3598 | 22.3839 | 17.1018 | **16.7486** | 20.9378 | 19.8334 | 18.8355 | 17.5003 | **14.791** |
| | GMI-HSIC | 19.0845 | 21.1116 | 21.5377 | 15.6066 | **14.469** | 18.9679 | 18.5146 | 18.4547 | 16.5750 | **12.7691** |
| *Fraction of Participants* | | $C = 0.5$ | | | | | $C = 1.0$ | | | | |
| Accuracy ↑ | | 87.37 | 97.78 | 98.46 | 98.87 | 99.10 | 80.38 | 97.89 | 98.52 | 98.85 | 99.02 |
| Attack Acc ↑ | GMI | 38.00±25.73 | 40.00±16.33 | 46.00±18.97 | 50.00±14.14 | **58.00±23.94** | 32.00±16.87 | 38.00±11.35 | 38.00±17.51 | 44.00±26.33 | **48.00±13.98** |
| | GMI-HSIC | 44.00±22.71 | 46.00±23.19 | 48.00±16.87 | 52.00±21.50 | **58.00±22.01** | 42.00±11.35 | 44.00±18.38 | 46.00±13.50 | 46.00±9.66 | **49.20±6.50** |
| Attack Acc5 ↑ | GMI | 96.00±8.43 | 96.00±8.43 | 94.00±9.66 | **98.00±6.32** | **98.00±6.32** | 94.00±9.66 | 96.00±8.43 | 98.00±6.32 | 98.00±6.32 | **100.00±0.00** |
| | GMI-HSIC | 98.00±6.32 | 98.00±6.32 | 96.00±8.43 | 98.00±6.32 | **100.00±0.00** | 98.00±6.32 | 96.00±8.43 | **100.00±0.00** | **100.00±0.00** | **100.00±0.00** |
| FID ↓ | GMI | 20.3550 | 20.1785 | 19.7692 | 19.3438 | **14.0115** | 24.7496 | 22.3330 | 20.7582 | 20.6225 | **18.5714** |
| | GMI-HSIC | 19.9560 | 19.4090 | 18.8257 | 17.3110 | **12.0485** | 23.1585 | 20.9507 | 19.8762 | 19.4196 | **17.8631** |

## A.4    IMPACT OF PARALLELISM ON FEDINVERSE+KED-MI ON MNIST

We investigate the influence of FL parallelism by modulating the active fraction of participants on the attack performance of KED-MI within the context of FL. Pertinent observations are detailed in Table 6. Unlike GMI, the KED-MI shows more stabilized attack performance, the attack accuracy varies from 80.00 to 79.97, when 10%, 20%, and 50% participants join each training round. And only decrease to 73.33 when 100% participants join each training round. Like KED-MI, the KED-MI-HSIC has the same results, varying from 80.20 to 79.95, when 10%, 20%, and 50% participants join each training round. And decrease to 75.85 when 100% participants join each training round. However, the top-5 attack accuracy remains 100.00 on both KED-MI and KED-MI-HSIC.

## A.5    IMPACT OF LOCAL COMPUTATION ON FEDINVERSE+GMI ON MNIST

The influence of local computation on each client plays a crucial role in determining the incorporation of local knowledge present within client data. This influence is governed by two primary hyperparameters $E$ and $B$. Consequently, we manipulate the values of $E$ and $B$ across a range of settings to assess the performance of GMI attacks on FL global models across various communication rounds. The results and pertinent observations from these experiments are presented in Table 7. We adjusted the local computation hyperparameters and observed that the local computation on each client has less impact on FedInverse using GMI and GMI-HSIC. The highest GMI attack accuracy is 56.00 when we set $E = 1$ and $B = 10$. The attack accuracy on the rest of the settings varies from 42.00 to 48.00 except the $E = 2$ and $B = 60$, which reaches 52.00 attack accuracy. For the GMI-HSIC, the highest attack accuracy is 60.00 when we set $E = 1$ and $B = 10$ or $E = 2$ and $B = 60$. The top-5 attack accuracy for GMI and GMI-HSIC are stabilized between 98.00 to 100.00.

## A.6    IMPACT OF LOCAL COMPUTATION ON FEDINVERSE+KED-MI ON MNIST

Table 8 shows the results and pertinent observations from experiments by adjusting the values of $E$ and $B$ to assess the performance of KED-MI attacks on FL global models across various communication rounds. Similar to GMI, We adjusted the local computation hyperparameters and observed that the local computation on each client has less impact on FedInverse using KED-MI and KED-

Table 6: FL privacy leakage indicated by Attack Acc/Acc5± standard deviation(%) and FID on MNIST via FedInverse using KED-MI and KED-MI-HSIC with varying active fractions of participants. Bold values denote the best metric results obtained by KED-MI or KED-MI-HSIC throughout the FL training epoch. The symbol ↓(↑) denotes that smaller (larger) values are favored.

| Metrics | Methods | FL#R01 | FL#R02 | FL#R03 | FL#R04 | FL#R05 | FL#R01 | FL#R02 | FL#R03 | FL#R04 | FL#R05 |
|---|---|---|---|---|---|---|---|---|---|---|---|
| *Fraction of Participants* | | $C = 0.1$ | | | | | $C = 0.2$ | | | | |
| Accuracy ↑ | | 83.34 | 97.59 | 98.27 | 98.4 | 98.52 | 83.49 | 97.87 | 98.36 | 98.73 | 98.94 |
| Attack Acc ↑ | KED-MI | 64.60±8.46 | 60.60±4.45 | **80.00±0.00** | **80.00±0.00** | 79.80±2.00 | 57.75±4.88 | 66.09±2.95 | 66.67±0.00 | 67.45±6.93 | **79.97±0.42** |
| | KED-MI-HSIC | 80.00±0.00 | 64.40±8.33 | 80.00±0.00 | **80.20±2.00** | 80.20±2.00 | 59.63±2.70 | 74.32±3.19 | 79.37±3.32 | 79.59±1.60 | **79.95±0.59** |
| Attack Acc5 ↑ | KED-MI | 100.00±0.00 | 100.00±0.00 | 100.00±0.00 | 100.00±0.00 | 100.00±0.00 | 100.00±0.00 | 100.00±0.00 | 100.00±0.00 | 100.00±0.00 | 100.00±0.00 |
| | KED-MI-HSIC | 100.00±0.00 | 100.00±0.00 | 100.00±0.00 | 100.00±0.00 | 100.00±0.00 | 100.00±0.00 | 100.00±0.00 | 100.00±0.00 | 100.00±0.00 | 100.00±0.00 |
| FID ↓ | KED-MI | 209.1448 | 206.0789 | 195.1807 | 184.995 | **175.9532** | 206.817 | 196.3008 | 199.594 | 189.9246 | **170.2652** |
| | KED-MI-HSIC | 204.5017 | 198.6938 | 175.9532 | 161.0252 | **160.9891** | 200.2681 | 195.9936 | 190.2452 | 187.61 | **131.5852** |
| *Fraction of Participants* | | $C = 0.5$ | | | | | $C = 1.0$ | | | | |
| Accuracy ↑ | | 87.37 | 97.78 | 98.46 | 98.87 | 99.10 | 80.38 | 97.89 | 98.52 | 98.85 | 99.02 |
| Attack Acc ↑ | KED-MI | 60.00±0.00 | 64.04±6.53 | 68.85±11.64 | 73.87±10.70 | **80.00±0.00** | 51.39±7.35 | 59.91±1.10 | 58.91±6.52 | 60.63±2.40 | **73.33±0.00** |
| | KED-MI-HSIC | 60.03±0.42 | 62.96±6.80 | 69.91±11.53 | 75.25±8.80 | **80.00±0.00** | 58.12±6.30 | 60.00±0.00 | 60.08±1.08 | 66.09±6.50 | **75.85±2.57** |
| Attack Acc5 ↑ | KED-MI | 99.96±0.71 | **100.00±0.00** | 100.00±0.00 | 100.00±0.00 | 100.00±0.00 | 100.00±0.00 | 100.00±0.00 | 100.00±0.00 | 100.00±0.00 | 100.00±0.00 |
| | KED-MI-HSIC | 99.80±1.93 | **100.00±0.00** | 100.00±0.00 | 100.00±0.00 | 100.00±0.00 | 100.00±0.00 | 100.00±0.00 | 100.00±0.00 | 100.00±0.00 | 100.00±0.00 |
| FID ↓ | KED-MI | 200.696 | 179.1579 | 173.4662 | 158.6515 | **156.9115** | 223.7763 | 218.7452 | 219.0277 | 199.7431 | **195.0272** |
| | KED-MI-HSIC | 187.5081 | 165.0061 | 158.7922 | **138.1317** | 145.5453 | 220.3384 | 198.8252 | 193.7816 | 191.4331 | **184.3157** |

MI-HSIC. The highest KED-MI attack accuracy is 86.67 when we set $E = 1$ and $B = 60$. The attack accuracy on the rest of the settings varies from 78.89 to 83.17. For the KED-MI-HSIC the highest attack accuracy appears $E = 1$ and $B = 120$, which is 93.33. Others vary from 80.00 to 87.64. The top-5 attack accuracy keeps 100.00 on all settings.

## A.7 IMPACT OF DEFENSE METHODS ON FEDINVERSE+GMI ON MNIST

To evaluate the performance of FedInverse against SOTA defense methods, we use two latest prevailing defense methods, MID Wang et al. (2021b) and BiDO Peng et al. (2022), to train the FL models. Meanwhile, we launch the GMI attacks on the updated global models in each federated communication round to see if these defense training schemes can still work in FL settings. The relevant observations are summarized in Table 9. The FedInverse with GMI and GMI-HSIC successfully attacked the FL with MID on the fourth training round and reached attack accuracy of 50.00 and 58.00, respectively. Similar to MID settings, GMI and GMI-HSIC successfully attacked the FL with BiDO on the fourth training round and reached attack accuracy of 40.00 and 42.00, respectively. For top-5 attack accuracy, GMI reached 100% in round 3, and GMI-HSIC reached 100% in round 2 with both MID and BiDO defense settings. The results show that the SOTA defense approach cannot defend against FedInverse+GMI attacks on FL settings on MNIST dataset.

## A.8 IMPACT OF DEFENSE METHODS ON FEDINVERSE+KED-MI ON MNIST

Table 10 illustrates the impact of defense methods on FedInverse+KED-MI on MNIST. The FedInverse with KED-MI and KED-MI-HSIC successfully attacked the FL with MID on the fourth training round and reached attack accuracy of 71.57 and 74.03, respectively. Similar to MID settings, KED-MI and KED-MI-HSIC successfully attacked the FL with BiDO on the fourth training round and reached attack accuracy of 61.01 and 66.85, respectively. For top-5 attack accuracy, KED-MI reached 100% since round 1, and KED-MI-HSIC also reached 100% in round 1 with both MID and BiDO defense settings. The results show that the SOTA defense approach cannot defend against FedInverse+KED-MI attacks on FL settings on MNIST dataset.

Table 7: FL privacy leakage indicated by Attack Acc/Acc5± standard deviation(%) and FID on MNIST via FedInverse using GMI and GMI-HSIC with varying local computation. Bold values denote the best metric results obtained by GMI or GMI-HSIC throughout the FL training epoch. The symbol ↓(↑) denotes that smaller (larger) values are favored.

| Metrics | Methods | FL#R01 | FL#R02 | FL#R03 | FL#R04 | FL#R05 | FL#R01 | FL#R02 | FL#R03 | FL#R04 | FL#R05 |
|---|---|---|---|---|---|---|---|---|---|---|---|
| *Local Computation* | | $(E, B) = (1, 10)$ | | | | | $(E, B) = (2, 10)$ | | | | |
| Accuracy ↑ | | 83.34 | 97.59 | 98.27 | 98.4 | 98.52 | 97.64 | 98.15 | 98.81 | 98.94 | 99.02 |
| Attack Acc ↑ | GMI | 34.00±9.66 | 38.00±22.01 | 34.00±16.47 | 50.00±10.54 | **56.00±20.66** | 32.00±19.32 | 34.00±18.97 | 38.00±19.89 | 40.00±9.43 | **42.00±17.51** |
| | GMI-HSIC | 44.00±15.78 | 44.00±12.65 | 42.00±14.76 | 56.00±8.43 | **60.00±9.43** | 36.00±12.65 | 40.00±16.33 | 42.00±19.89 | 44.00±15.78 | **48.00±21.50** |
| Attack Acc5 ↑ | GMI | 94.00±9.66 | **98.00±6.32** | **98.00±6.32** | 96.00±8.43 | **98.00±6.32** | 92.00±13.98 | 94.00±9.66 | 96.00±8.43 | 94.00±9.66 | **98.00±6.32** |
| | GMI-HSIC | 96.00±8.43 | 98.00±6.32 | 98.00±6.32 | 100.00±0.00 | 98.00±6.32 | 96.00±8.43 | 98.00±6.32 | 98.00±6.32 | 100.00±0.00 | 100.00±0.00 |
| FID ↓ | GMI | 20.1373 | 23.3598 | 22.3839 | 17.1018 | **16.7486** | 23.7213 | 23.6720 | 22.8779 | 21.4082 | **20.7127** |
| | GMI-HSIC | 19.0845 | 21.1116 | 21.5377 | 15.6066 | **14.469** | 21.3812 | 21.4646 | 21.2353 | 20.7013 | **19.7279** |
| *Local Computation* | | $(E, B) = (1, 30)$ | | | | | $(E, B) = (2, 30)$ | | | | |
| Accuracy ↑ | | 85.30 | 95.77 | 97.37 | 98.22 | 98.38 | 95.07 | 98.03 | 98.61 | 98.46 | 98.92 |
| Attack Acc ↑ | GMI | 36.00±18.38 | 38.00±17.51 | 42.00±22.01 | 42.00±14.76 | **46.00±13.50** | 34.00±18.97 | 36.00±22.71 | 38.00±19.89 | 40.00±18.86 | **44.00±15.78** |
| | GMI-HSIC | 38.00±14.76 | 40.00±18.86 | 42.00±14.76 | 44.00±15.78 | **46.58±18.97** | 36.00±22.71 | 38.00±19.89 | 40.00±18 | 42.00±14.76 | **48.00±21.50** |
| Attack Acc5 ↑ | GMI | 94.00±9.66 | 96.00±8.43 | 98.00±6.32 | 98.00±6.32 | **100.00±0.00** | 94.00±9.66 | **98.00±6.32** | **98.00±6.32** | 96.00±8.43 | **98.00±6.32** |
| | GMI-HSIC | **100.00±0.00** | 100.00±0.00 | 98.00±6.32 | 100.00±0.00 | 100.00±0.00 | 94.00±9.66 | 98.00±6.32 | 100.00±0.00 | 98.00±6.32 | 100.00±0.00 |
| FID ↓ | GMI | 23.9145 | 22.6867 | 21.1273 | 20.2517 | **18.1435** | 20.4175 | 19.0410 | 18.6176 | 18.2479 | **16.4464** |
| | GMI-HSIC | 22.7109 | 20.5478 | 18.8782 | 20.1115 | **17.4279** | 19.6434 | 17.3692 | 18.1270 | 17.3496 | **16.0987** |
| *Local Computation* | | $(E, B) = (1, 60)$ | | | | | $(E, B) = (2, 60)$ | | | | |
| Accuracy ↑ | | 86.35 | 90.40 | 93.79 | 96.40 | 97.52 | 90.48 | 96.65 | 97.17 | 98.09 | 98.38 |
| Attack Acc ↑ | GMI | 34.00±23.19 | 36.00±15.78 | 44.00±20.66 | 42.00±14.76 | **48.00±19.32** | 38.00±17.51 | 42.00±19.89 | 44.00±15.78 | 50.00±17.00 | **52.00±21.50** |
| | GMI-HSIC | 36.00±18.38 | 38.00±14.76 | 48.00±16.87 | 44.00±15.78 | **50.00±17.00** | 42.00±11.35 | 46.00±13.50 | 50.00±17.00 | 52.00±16.87 | **60.00±16.33** |
| Attack Acc5 ↑ | GMI | 86.00±18.97 | 88.00±13.98 | **96.00±8.43** | 94.00±9.66 | **96.00±8.43** | 94.00±9.66 | 96.00±8.43 | 98.00±6.32 | 100.00±0.00 | 96.00±8.43 |
| | GMI-HSIC | 94.00±9.66 | 92.00±10.33 | 98.00±6.32 | 96.00±8.43 | **98.00±6.32** | 96.00±8.43 | 98.00±6.32 | **100.00±0.00** | **100.00±0.00** | 100.00±0.00 |
| FID ↓ | GMI | 22.6974 | 19.9287 | 17.6055 | 16.6408 | **15.4819** | 22.8858 | 21.0912 | 17.7921 | 15.3665 | **12.3089** |
| | GMI-HSIC | 20.9824 | 18.3025 | 17.4393 | 15.6629 | **15.4757** | 21.2270 | 20.7513 | 15.6794 | 15.5562 | **10.6041** |
| *Local Computation* | | $(E, B) = (1, 120)$ | | | | | $(E, B) = (2, 120)$ | | | | |
| Accuracy ↑ | | 51.34 | 56.95 | 68.90 | 78.90 | 92.08 | 61.04 | 82.81 | 92.33 | 95.99 | 97.27 |
| Attack Acc ↑ | GMI | 30.00±10.54 | 32.00±21.50 | 40.00±13.33 | 42.00±23.94 | **44.00±22.71** | 30.00±19.44 | 32.00±13.98 | 32.00±21.50 | 38.00±19.89 | **42.00±16.47** |
| | GMI-HSIC | 34.00±13.50 | 38.00±14.76 | 42.00±14.76 | 46.00±18.97 | **48.00±19.32** | 30.00±17.00 | 36.00±15.78 | 38.00±22.01 | 40.00±23.09 | **44.00±20.66** |
| Attack Acc5 ↑ | GMI | 92.00±13.98 | 96.00±12.65 | **98.00±6.32** | 96.00±8.43 | **98.00±6.32** | 90.00±10.54 | **96.00±8.43** | **96.00±8.43** | 94.00±9.66 | **96.00±8.43** |
| | GMI-HSIC | 96.00±12.65 | 98.00±6.32 | 100.00±0.00 | 100.00±0.00 | 100.00±0.00 | 98.00±6.32 | **98.00±6.32** | 96.00±8.43 | **98.00±6.32** | 98.00±6.32 |
| FID ↓ | GMI | 21.2109 | 21.4954 | 19.8025 | 19.1230 | **18.9499** | 25.0463 | 23.8215 | 23.4771 | 23.5279 | **22.0935** |
| | GMI-HSIC | 19.5419 | 20.4106 | 18.4925 | 18.7281 | **17.7968** | 22.6472 | 22.4640 | 22.4177 | 21.9816 | **21.0760** |

## A.9 FEDINVERSE ATTACK PERFORMANCE ON CELEBA

The results of FedInverse using GMI and GMI-HSIC are illustrated in Table 11, where we have the attack performance with varying GMI attack settings and FL training rounds. As shown in Table 11, GMI-HSIC consistently achieves better attack performance and FID since the first FL training round, indicated by improvement of 10% of the attack accuracy (8.50 vs 9.43), 2% of the top-5 attack accuracy (22.07 vs 22.70), and FID (97.1281 vs 96.9064). FL training accuracy has been increased and stabilized in the second round from 66.05 to 80.48, and it is worthy noting that GMI-HSIC achieves the highest attack accuracy and top-5 attack accuracy which outperforms GMI by 5% of the attack accuracy (11.33 vs 11.93), 5% of the top-5 attack accuracy (25.49 vs 26,77), and smaller FID (104.0733 vs 101.0956). When compared with the GMI and GMI-HSIC, GMI can partially leak images from other participants and the attack performance can be significantly improved by GMI-HSIC, in which the diversity of the generated images has been optimized by HSIC.

Table 8: FL privacy leakage indicated by Attack Acc/Acc5± standard deviation(%) and FID on MNIST via FedInverse using KED-MI and KED-MI-HSIC with varying local computation. Bold values denote the best metric results obtained by KED-MI or KED-MI-HSIC throughout the FL training epoch. The symbol ↓(↑) denotes that smaller (larger) values are favored.

| Metrics | Methods | FL#R01 | FL#R02 | FL#R03 | FL#R04 | FL#R05 | FL#R01 | FL#R02 | FL#R03 | FL#R04 | FL#R05 |
|---|---|---|---|---|---|---|---|---|---|---|---|
| *Local Computation* | | $(E, B) = (1, 10)$ | | | | | $(E, B) = (2, 10)$ | | | | |
| Accuracy ↑ | | 83.34 | 97.59 | 98.27 | 98.4 | 98.52 | 97.64 | 98.15 | 98.81 | 98.94 | 99.02 |
| Attack Acc ↑ | KED-MI | 64.60±8.46 | 60.60±4.45 | **80.00±0.00** | **80.00±0.00** | 79.80±2.00 | 60.15±1.73 | 61.35±4.91 | 74.17±9.26 | 78.44±5.77 | **78.89±2.48** |
| | KED-MI-HSIC | 80.00±0.00 | 64.40±8.33 | 80.00±0.00 | **80.20±2.00** | **80.20±2.00** | 60.52±3.16 | 61.85±5.71 | 79.81±1.10 | 79.99±6.63 | **85.09±4.14** |
| Attack Acc5 ↑ | KED-MI | 100.00±0.00 | 100.00±0.00 | 100.00±0.00 | 100.00±0.00 | 100.00±0.00 | 100.00±0.00 | 100.00±0.00 | 100.00±0.00 | 100.00±0.00 | 100.00±0.00 |
| | KED-MI-HSIC | 100.00±0.00 | 100.00±0.00 | 100.00±0.00 | 100.00±0.00 | 100.00±0.00 | 100.00±0.00 | 100.00±0.00 | 100.00±0.00 | 100.00±0.00 | 100.00±0.00 |
| FID ↓ | KED-MI | 209.1448 | 206.0789 | 195.1807 | 184.995 | **175.9532** | 189.0635 | 182.2778 | 178.8578 | 173.078 | **167.2463** |
| | KED-MI-HSIC | 204.5017 | 198.6938 | 175.9532 | 161.0252 | **160.9891** | 182.813 | 180.9875 | 173.1939 | 169.6834 | **162.9299** |
| *Local Computation* | | $(E, B) = (1, 30)$ | | | | | $(E, B) = (2, 30)$ | | | | |
| Accuracy ↑ | | 85.30 | 95.77 | 97.37 | 98.22 | 98.38 | 95.07 | 98.03 | 98.61 | 98.46 | 98.92 |
| Attack Acc ↑ | KED-MI | 62.69±5.75 | 64.55±5.96 | 71.20±3.11 | 78.35±2.88 | **80.00±0.00** | 60.00±0.00 | 64.60±11.99 | 66.88±6.62 | 76.53±6.47 | **83.17±7.21** |
| | KED-MI-HSIC | 66.73±0.66 | 65.52±2.51 | 72.55±3.70 | **80.00±0.00** | **80.00±0.00** | 62.67±6.39 | 65.29±11.05 | 68.85±9.77 | 79.40±3.32 | **83.64±7.69** |
| Attack Acc5 ↑ | KED-MI | 100.00±0.00 | 100.00±0.00 | 100.00±0.00 | 100.00±0.00 | 100.00±0.00 | 100.00±0.00 | 100.00±0.00 | 100.00±0.00 | 100.00±0.00 | 100.00±0.00 |
| | KED-MI-HSIC | 100.00±0.00 | 100.00±0.00 | 100.00±0.00 | 100.00±0.00 | 100.00±0.00 | 100.00±0.00 | 100.00±0.00 | 100.00±0.00 | 100.00±0.00 | 100.00±0.00 |
| FID ↓ | KED-MI | 191.7317 | 186.0394 | 175.8877 | 169.0031 | **162.7329** | 217.0403 | 210.2183 | 204.235 | 183.4337 | **180.9140** |
| | KED-MI-HSIC | 186.9849 | 185.2320 | 172.8691 | 164.9772 | **160.6046** | 206.0624 | 203.1168 | 196.6989 | 178.4203 | **162.7005** |
| *Local Computation* | | $(E, B) = (1, 60)$ | | | | | $(E, B) = (2, 60)$ | | | | |
| Accuracy ↑ | | 86.35 | 90.40 | 93.79 | 96.40 | 97.52 | 90.48 | 96.65 | 97.17 | 98.09 | 98.38 |
| Attack Acc ↑ | KED-MI | 59.92±0.72 | 61.19±4.02 | 67.31±6.49 | 77.89±3.10 | **86.67±0.00** | 59.72±2.30 | 71.00±12.85 | 75.33±8.54 | 79.72±2.2898 | **80.00±0.00** |
| | KED-MI-HSIC | 62.17±4.45 | 66.67±0.00 | 68.00±3.3821 | 80.00±0.00 | **87.64±0.42** | 60.00±0.00 | 73.15±1.89 | 76.93±9.23 | 79.69±2.25 | **80.00±0.00** |
| Attack Acc5 ↑ | KED-MI | 100.00±0.00 | 100.00±0.00 | 100.00±0.00 | 100.00±0.00 | 100.00±0.00 | 100.00±0.00 | 100.00±0.00 | 100.00±0.00 | 100.00±0.00 | 100.00±0.00 |
| | KED-MI-HSIC | 100.00±0.00 | 100.00±0.00 | 100.00±0.00 | 100.00±0.00 | 100.00±0.00 | 100.00±0.00 | 100.00±0.00 | 100.00±0.00 | 100.00±0.00 | 100.00±0.00 |
| FID ↓ | KED-MI | 194.3525 | 188.9867 | 198.0894 | 179.0765 | **170.3807** | 203.035 | 192.8893 | 179.7266 | 173.2627 | **154.8115** |
| | KED-MI-HSIC | 186.6046 | 163.9228 | 163.4239 | 148.2969 | **131.0784** | 185.0541 | 173.8115 | 171.704 | 163.2503 | **153.6553** |
| *Local Computation* | | $(E, B) = (1, 120)$ | | | | | $(E, B) = (2, 120)$ | | | | |
| Accuracy ↑ | | 51.34 | 56.95 | 68.90 | 78.90 | 92.08 | 61.04 | 82.81 | 92.33 | 95.99 | 97.27 |
| Attack Acc ↑ | KED-MI | 47.56±4.47 | 60.03±0.42 | 66.91±1.50 | 79.65±1.48 | **80.00±0.00** | 46.89±1.82 | 53.80±10.53 | 66.35±1.42 | 66.67±0.00 | **80.00±0.00** |
| | KED-MI-HSIC | 60.00±0.00 | 66.67±0.00 | 73.33±0.00 | 80.00±0.00 | **93.33±0.00** | 60.33±2.50 | 63.73±11.20 | 66.67±0.00 | 75.77±3.21 | **80.01±0.29** |
| Attack Acc5 ↑ | KED-MI | 100.00±0.00 | 100.00±0.00 | 100.00±0.00 | 100.00±0.00 | 100.00±0.00 | 100.00±0.00 | 100.00±0.00 | 100.00±0.00 | 100.00±0.00 | 100.00±0.00 |
| | KED-MI-HSIC | 100.00±0.00 | 100.00±0.00 | 100.00±0.00 | 100.00±0.00 | 100.00±0.00 | 100.00±0.00 | 100.00±0.00 | 100.00±0.00 | 100.00±0.00 | 100.00±0.00 |
| FID ↓ | KED-MI | 243.4233 | 230.6205 | 212.6278 | 192.4296 | **185.4447** | 243.9973 | 228.1217 | 214.8240 | 200.3790 | **184.5743** |
| | KED-MI-HSIC | 216.9471 | 215.4928 | 203.0319 | 182.6337 | **170.4385** | 196.2384 | 187.2983 | 179.7032 | 162.6797 | **160.1287** |

The performance evaluation of FedInverse using KED-MI attack on CelebA is presented in Table 12, where we use the same attack settings as the GMI attack. Compare to GMI, KED-MI completely penetrates the mechanism of FL personal privacy protection, and the best attack accuracy dramatically increases from 11.33 in Table 11 to 57.80 in Table 12, and the best top-5 attack accuracy booms from 25.40 Table 11 to 85.47 Table 12. The best result of attack accuracy and top-5 attack accuracy also increases from 11.93 in Table 11 to 60.13 in Table 12, and 26.77 in Table 11 to 85.80 in Table 12, respectively, compared to GMI-HISC and KED-MI-HSIC. When we compare the attack performance between KED-MI and KED-MI-HSIC, KED-MI-HSIC improves 4% of the attack accuracy (57.80 vs 60.13), 2% of the top-5 accuracy (82.73 vs 84.73), and smaller FID (244.159 vs 239.4701) in FL training round 9 and 10 respectively. According to the results, participants' privacy in FL can be easily revealed by the KED-MI, and HSIC plays an important role in increasing the diversity of the generated images where the attack accuracy can be improved.

Table 9: FL privacy leakage indicated by Attack Acc/Acc5± standard deviation(%) and FID on MNIST via FedInverse using GMI and GMI-HSIC with two diverse defense methods: MID and BiDO. Bold values denote the best metric results obtained by GMI or GMI-HSIC throughout the FL training epoch. The symbol ↓(↑) denotes that smaller (larger) values are favored.

| Metrics | Methods | FL#R01 | FL#R02 | FL#R03 | FL#R04 | FL#R05 | FL#R01 | FL#R02 | FL#R03 | FL#R04 | FL#R05 |
|---|---|---|---|---|---|---|---|---|---|---|---|
| *Defense Method* | | | | MID | | | | | BiDO | | |
| Accuracy ↑ | | 86.69 | 97.85 | 97.97 | 98.59 | 98.54 | 95.79 | 96.20 | 97.47 | 97.93 | 96.51 |
| Attack Acc ↑ | GMI | 38.00±22.01 | 40.00±9.43 | 42.00±14.76 | **50.00±10.54** | 40.00±18.86 | 32.00±19.32 | 34.00±13.49 | 34.00±21.18 | **40.00±13.33** | 38.00±19.88 |
| | GMI-HSIC | 40.00±21.08 | 42.00±14.76 | 44.00±18.38 | **58.00±11.35** | 40.00±23.09 | 34.00±18.97 | 36.00±12.65 | 36.00±15.78 | **42.00±19.89** | 40.00±16.33 |
| Attack Acc5 ↑ | GMI | 96.00±8.43 | 98.00±6.32 | **100.00±0.00** | **100.00±0.00** | **100.00±0.00** | 96.00±8.43 | 96.00±8.43 | 98.00±6.32 | **100.00±0.00** | **100.00±0.00** |
| | GMI-HSIC | 98.00±6.32 | **100.00±0.00** | **100.00±0.00** | **100.00±0.00** | **100.00±0.00** | 98.00±6.32 | **100.00±0.00** | **100.00±0.00** | **100.00±0.00** | **100.00±0.00** |
| FID ↓ | GMI | 23.9977 | 21.4223 | 20.1756 | 20.9959 | **19.9425** | 26.3337 | 25.0333 | 24.2407 | 23.7489 | **21.6887** |
| | GMI-HSIC | 23.4958 | 20.1026 | 19.8077 | 19.2872 | **18.2523** | 25.5310 | 24.3883 | 23.1513 | 22.8132 | **20.1309** |

Table 10: FL privacy leakage indicated by Attack Acc/Acc5± standard deviation(%) and FID on MNIST via FedInverse using KED-MI and KED-MI-HSIC with two diverse defense methods: MID and BiDO. Bold values denote the best metric results obtained by GMI or GMI-HSIC throughout the FL training epoch. The symbol ↓(↑) denotes that smaller (larger) values are favored.

| Metrics | Methods | FL#R01 | FL#R02 | FL#R03 | FL#R04 | FL#R05 | FL#R01 | FL#R02 | FL#R03 | FL#R04 | FL#R05 |
|---|---|---|---|---|---|---|---|---|---|---|---|
| *Defense Method* | | | | MID | | | | | BiDO | | |
| Accuracy ↑ | | 86.69 | 97.85 | 97.97 | 98.59 | 98.54 | 95.79 | 96.20 | 97.47 | 97.93 | 96.51 |
| Attack Acc ↑ | KED-MI | 58.35±11.99 | 62.63±3.85 | 64.60±6.99 | **71.57±10.33** | 67.88±4.06 | 43.12±7.23 | 46.67±0.00 | 49.76±8.77 | **61.01±5.61** | 60.00±0.00 |
| | KED-MI-HSIC | 59.67±6.21 | 64.93±7.62 | 65.57±8.69 | **74.03±8.47** | 73.23±13.19 | 45.45±8.33 | 48.85±8.12 | 56.32±8.53 | **66.85±3.64** | 64.11±9.34 |
| Attack Acc5 ↑ | KED-MI | 100.00±0.00 | 100.00±0.00 | 100.00±0.00 | 100.00±0.00 | 100.00±0.00 | 100.00±0.00 | 100.00±0.00 | 100.00±0.00 | 100.00±0.00 | 100.00±0.00 |
| | KED-MI-HSIC | 100.00±0.00 | 100.00±0.00 | 100.00±0.00 | 100.00±0.00 | 100.00±0.00 | 100.00±0.00 | 100.00±0.00 | 100.00±0.00 | 100.00±0.00 | 100.00±0.00 |
| FID ↓ | KED-MI | 205.2744 | 190.9697 | 184.2229 | 173.4399 | **172.9889** | 256.3345 | 237.0378 | 233.1303 | 220.3506 | **196.9305** |
| | KED-MI-HSIC | 200.1239 | 187.4048 | 179.6035 | **165.1414** | 166.0872 | 235.8799 | 235.8799 | 213.6232 | 209.4665 | **179.0025** |

We also evaluate the FedInverse performance of VMI in Table 13 with ResNet-34. The highest attack accuracy of VMI appears in FL training round 10 is 36.70 and the highest top-5 attack accuracy appears in FL training round 9 is 63.30. The VMI-HSIC further improves the attack performance by 3% and the top-5 attack accuracy by 0.7%, which are 37.95 and 63.80, respectively, with smaller FID (0.8942 vs 0.891). The results validate the promised privacy leakage by VMI attacks.

Table 11: FL privacy leakage indicated by Attack Acc/Acc5±standard deviation(%) and FID on CelebA via FedInverse using GMI and GMI-HSIC. Bold values denote the best metric results obtained by GMI or GMI-HSIC throughout the FL training epoch. The symbol ↓(↑) denotes that smaller (larger) values are favored.

| Metrics | Methods | FL#R01 | FL#R02 | FL#R03 | FL#R04 | FL#R05 | FL#R06 | FL#R07 | FL#R08 | FL#R09 | FL#R10 |
|---|---|---|---|---|---|---|---|---|---|---|---|
| Accuracy ↑ | | 66.05 | 80.48 | 81.11 | 81.64 | 81.68 | 81.84 | 82.41 | 82.44 | 82.74 | 82.54 |
| Attack Acc ↑ | GMI | 8.50±3.26 | 9.80±4.08 | 10.07±3.91 | 10.37±4.10 | 10.10±3.96 | 10.17±3.87 | 10.53±3.90 | 10.93±3.82 | **11.33±4.67** | 10.70±3.62 |
| | GMI-HSIC | 9.43±3.78 | 10.53±3.35 | 10.63±3.66 | 10.83±3.58 | 11.00±3.91 | 11.57±4.06 | 11.13±3.98 | 11.73±4.17 | **11.93±4.33** | 10.97±4.91 |
| Attack Acc5 ↑ | GMI | 22.07±4.62 | 23.80±5.24 | 23.80±5.63 | 23.93±6.03 | 24.07±5.73 | 24.20±5.42 | 24.47±4.99 | 24.83±6.57 | **25.40±7.35** | 25.20±5.15 |
| | GMI-HSIC | 22.70±5.46 | 24.93±5.47 | 25.00±5.60 | 25.03±5.26 | 25.23±5.18 | 25.63±5.68 | 25.63±5.03 | 25.57±6.18 | **26.77±5.84** | 25.90±6.54 |
| FID ↓ | GMI | 108.2603 | 108.5283 | 107.6128 | 108.7106 | 107.0249 | 106.814 | 104.0733 | 103.0706 | **97.1281** | 104.7775 |
| | GMI-HSIC | 106.3502 | 104.8022 | 105.0613 | 106.8688 | 106.6671 | 104.0582 | 104.404 | 101.0956 | **96.9064** | 102.5457 |

Table 12: FL privacy leakage indicated by Attack Acc/Acc5±standard deviation(%) and FID on CelebA via FedInverse using KED-MI and KED-MI-HSIC. Bold values denote the best metric results obtained by KED-MI or KED-MI-HSIC throughout the FL training epoch. The symbol ↓(↑) denotes that smaller (larger) values are favored.

| Metrics | Methods | FL#R01 | FL#R02 | FL#R03 | FL#R04 | FL#R05 | FL#R06 | FL#R07 | FL#R08 | FL#R09 | FL#R10 |
|---|---|---|---|---|---|---|---|---|---|---|---|
| Accuracy ↑ | | 66.05 | 80.48 | 81.11 | 81.64 | 81.68 | 81.84 | 82.41 | 82.44 | 82.74 | 82.54 |
| Attack Acc ↑ | KED-MI | 39.53±1.87 | 53.47±2.91 | 55.60±2.64 | 56.93±3.19 | 56.80±4.54 | 54.67±3.40 | 56.60±2.97 | 53.13±3.33 | **57.80±4.04** | 56.73±2.62 |
| | KED-MI-HSIC | 41.73±1.76 | 57.80±3.07 | 55.07±4.31 | 59.33±3.45 | 57.47±2.42 | 57.07±3.88 | 59.13±3.95 | 55.73±2.87 | **60.13±3.53** | 59.47±3.30 |
| Attack Acc5 ↑ | KED-MI | 67.67±2.90 | 80.07±2.32 | 81.40±2.93 | 83.73±2.38 | 82.07±2.75 | 81.73±2.66 | 82.53±3.27 | 81.60±2.17 | **85.47±2.39** | 82.73±2.02 |
| | KED-MI-HSIC | 67.53±2.07 | 80.33±3.26 | 82.13±2.74 | 83.13±2.88 | 81.80±2.18 | 83.47±2.00 | 83.40±2.49 | 82.87±2.17 | **85.80±2.27** | 84.73±0.97 |
| FID ↓ | KED-MI | 241.0025 | 243.3093 | 235.3072 | 249.9985 | 247.8074 | 247.2511 | **227.6555** | 241.8614 | 244.159 | 248.9721 |
| | KED-MI-HSIC | 232.7846 | 242.5915 | 232.5486 | 245.5304 | 241.8464 | 241.6832 | **225.5088** | 238.4409 | 239.4701 | 248.7714 |

Table 13: FL privacy leakage indicated by Attack Acc/Acc5±standard deviation(%) and FID on CelebA via FedInverse using VMI and VMI-HSIC. Bold values denote the best metric results obtained by VMI or VMI-HSIC throughout the FL training epoch. The symbol ↓(↑) denotes that smaller (larger) values are favored.

| Metrics | Methods | FL#R01 | FL#R02 | FL#R03 | FL#R04 | FL#R05 | FL#R06 | FL#R07 | FL#R08 | FL#R09 | FL#R10 |
|---|---|---|---|---|---|---|---|---|---|---|---|
| Accuracy ↑ | | 38.17 | 47.78 | 49.03 | 59.53 | 58.85 | 59.55 | 65.54 | 65.5 | 65.24 | 65.16 |
| Attack Acc ↑ | VMI | 16.85±14.60 | 27.25±17.98 | 25.75±17.66 | 29.05±16.85 | 31.00±20.62 | 33.15±20.81 | 33.40±21.24 | 36.45±21.23 | 36.65±20.83 | **36.70±20.31** |
| | VMI-HSIC | 19.00±18.31 | 27.65±18.06 | 25.75±18.42 | 29.75±16.85 | 31.05±21.40 | 32.60±20.25 | 33.90±21.29 | 37.20±20.47 | 37.10±21.08 | **37.95±20.35** |
| Attack Acc5 ↑ | VMI | 33.20±23.84 | 48.95±27.10 | 46.00±27.71 | 50.70±22.53 | 54.00±28.94 | 55.35±26.17 | 58.75±25.61 | 62.75±23.26 | **63.30±23.55** | 63.20±22.89 |
| | VMI-HSIC | 35.75±23.97 | 50.20±26.89 | 45.65±28.84 | 53.30±22.66 | 54.75±28.35 | 56.45±26.85 | 61.70±23.54 | 63.10±23.78 | **63.80±21.75** | 63.75±23.58 |
| FID ↓ | VMI | 1.0615 | 0.963 | 0.9677 | 0.9466 | 0.9377 | 0.9328 | 0.9069 | 0.8989 | 0.8957 | **0.8942** |
| | VMI-HSIC | 1.0275 | 0.9562 | 0.965 | 0.9401 | 0.9359 | 0.9317 | 0.9023 | 0.899 | 0.8912 | **0.8910** |

Table 14: FL privacy leakage indicated by Attack Acc/Acc5±standard deviation(%) and FID on CelebA via FedInverse using GMI and GMI-HSIC with varying active fractions of participants. Bold values denote the best metric results obtained by GMI or GMI-HSIC throughout the FL training epoch. The symbol ↓(↑) denotes that smaller (larger) values are favored.

| Metrics | Methods | FL#R01 | FL#R02 | FL#R03 | FL#R04 | FL#R05 | FL#R06 | FL#R07 | FL#R08 | FL#R09 | FL#R10 |
|---|---|---|---|---|---|---|---|---|---|---|---|
| *Fraction of participants* | | $C = 0.4$ | | | | | | | | | |
| Accuracy ↑ | | 67.65 | 80.35 | 81.68 | 82.08 | 82.44 | 82.41 | 82.67 | 82.91 | 83.11 | 82.54 |
| Attack Acc ↑ | GMI | 10.67±4.12 | 11.17±3.87 | 10.33±3.40 | 12.67±4.90 | 11.33±3.86 | 9.00±4.68 | 10.50±3.78 | 12.33±3.32 | 8.83±4.38 | **13.00±3.87** |
| | GMI-HSIC | 12.00±3.73 | **12.83±5.35** | 12.17±4.64 | 12.00±2.65 | **12.83±6.57** | 11.83±3.06 | 11.00±3.91 | 11.67±4.61 | 12.00±4.80 | **12.83±4.18** |
| Attack Acc5 ↑ | GMI | 25.83±6.03 | 25.33±5.37 | 25.83±7.20 | 23.00±8.61 | 25.33±4.72 | 23.17±3.84 | 24.17±7.25 | 25.67±7.17 | 23.67±7.70 | **27.17±6.64** |
| | GMI-HSIC | 25.00±5.60 | 26.50±6.01 | 25.50±6.73 | 26.33±8.98 | 27.83±6.96 | 25.90±6.54 | 26.50±4.27 | 29.00±8.45 | **29.33±8.88** | 26.83±7.03 |
| FID ↓ | GMI | 157.5960 | 157.9935 | 148.8573 | 154.4589 | 155.4286 | 150.9164 | 150.3659 | 146.8032 | **141.5987** | 144.3315 |
| | GMI-HSIC | 152.7228 | 149.6889 | 147.5066 | 147.9429 | 147.9600 | 147.2307 | 144.5968 | 146.9599 | **135.7852** | 141.4002 |
| *Fraction of participants* | | $C = 0.6$ | | | | | | | | | |
| Accuracy ↑ | | 69.28 | 80.68 | 81.38 | 81.51 | 82.01 | 82.18 | 82.24 | 82.44 | 82.64 | 82.71 |
| Attack Acc ↑ | GMI | 9.50±4.46 | 9.83±4.02 | 11.50±4.44 | 8.33±5.26 | 10.33±4.46 | 7.83±3.06 | 10.50±5.19 | 11.33±4.61 | 11.67±4.98 | **13.00±5.65** |
| | GMI-HSIC | 10.17±4.47 | 10.67±5.17 | 10.50±3.23 | 11.67±3.98 | 11.17±4.51 | 12.17±5.37 | 12.67±7.07 | 13.00±6.07 | **14.33±4.77** | 13.67±4.66 |
| Attack Acc5 ↑ | GMI | 21.83±7.01 | 21.50±6.67 | 21.67±6.23 | 24.50±8.16 | 24.00±6.26 | 26.83±7.91 | 26.50±6.51 | 27.00±5.26 | **27.33±5.18** | 26.50±5.99 |
| | GMI-HSIC | 25.33±4.92 | 26.33±9.00 | 27.33±8.07 | 27.50±6.00 | 27.67±4.95 | 27.67±6.11 | 28.33±5.46 | 28.17±10.37 | **29.00±5.04** | 28.17±6.28 |
| FID ↓ | GMI | 151.0542 | 152.1294 | 152.2443 | 150.9182 | 149.3314 | 150.4244 | 147.8318 | 142.7556 | **140.6171** | 143.4548 |
| | GMI-HSIC | 149.0826 | 146.6641 | 144.3291 | 141.4876 | 143.3411 | 142.9232 | 143.2222 | 141.6792 | **134.0685** | 139.9209 |
| *Fraction of participants* | | $C = 1.0$ | | | | | | | | | |
| Accuracy ↑ | | 66.05 | 80.48 | 81.11 | 81.64 | 81.68 | 81.84 | 82.41 | 82.44 | 82.74 | 82.54 |
| Attack Acc ↑ | GMI | 8.50±3.26 | 9.80±4.08 | 10.07±3.91 | 10.37±4.10 | 10.10±3.96 | 10.17±3.87 | 10.53±3.90 | 10.93±3.82 | **11.33±4.67** | 10.70±3.62 |
| | GMI-HSIC | 9.43±3.78 | 10.53±3.35 | 10.63±3.66 | 10.83±3.58 | 11.00±3.91 | 11.57±4.06 | 11.13±3.98 | 11.73±4.17 | **11.93±4.33** | 10.97±4.91 |
| Attack Acc5 ↑ | GMI | 22.07±4.62 | 23.80±5.24 | 23.80±5.63 | 23.93±6.03 | 24.07±5.73 | 24.20±5.42 | 24.47±4.99 | 24.83±6.57 | **25.40±7.35** | 25.20±5.15 |
| | GMI-HSIC | 22.70±5.46 | 24.93±5.47 | 25.00±5.60 | 25.03±5.26 | 25.23±5.18 | 25.63±5.68 | 25.63±5.03 | 25.57±6.18 | **26.77±5.84** | 25.90±6.54 |
| FID ↓ | GMI | 108.2603 | 108.5283 | 107.6128 | 108.7106 | 107.0249 | 106.814 | 104.0733 | 103.0706 | **97.1281** | 104.7775 |
| | GMI-HSIC | 106.3502 | 104.8022 | 105.0613 | 106.8688 | 106.6671 | 104.0582 | 104.404 | 101.0956 | **96.9064** | 102.5457 |

## A.10 IMPACT OF PARALLELISM ON FEDINVERSE+GMI ON CELEBA

We investigate the influence of FL parallelism by modulating the active fraction of participants on the attack performance of GMI within the context of FL. Pertinent observations are detailed in Table 14. The results show similar trends as MNIST parallelism experiments. The number of participants does not greatly impact the attack performance, whereas GMI has a similar attack accuracy of 13.00 when 40% and 60% of the participants join the FL training round. The results also show that GMI-HSIC has less impact on the attack performance, where the highest attack accuracy is 14.33 when 60% of the participants join the FL training round, and the lowest attack accuracy is 11.93 when 100% of the participants join the FL training round. Comparing top-5 attack accuracy between the number of training participants, the highest GMI top-5 attack accuracy is 27.33 when 60% of the participants join the FL training round, and the lowest top-5 attack accuracy of GMI is 25.40. Same as GMI-HSIC, the highest top-5 attack accuracy is 29.33, and lowest top-5 attack accuracy is 26.77.

## A.11 IMPACT OF PARALLELISM ON FEDINVERSE+KED-MI ON CELEBA

Table 15 shows that KED-MI and KED-MI-HSIC achieved better attack performance than GMI and GMI-HSIC. The number of participants does not greatly impact the attack performance as well. The highest attack accuracy of KED-MI is 61.93 when 40% of participants join the FL training round, and the lowest attack accuracy is 57.80 when 100% of participants join the FL training round. KED-MI-HSIC reaches the highest attack accuracy of 64.80 at 40% of participants and the lowest attack accuracy of 60.13 at 100% of of participants. For top-5 attack accuracy, both KED-MI and KED-MI-HSIC have similar patterns. The highest top-5 attack accuracy appears at participant = 10%,

Table 15: FL privacy leakage indicated by Attack Acc/Acc5±standard deviation(%) and FID on CelebA via FedInverse using KED-MI and KED-MI-HSIC with varying active fractions of participants. Bold values denote the best metric results obtained by KED-MI or KED-MI-HSIC throughout the FL training epoch. The symbol ↓(↑) denotes that smaller (larger) values are favored.

| Metrics | Methods | FL#R01 | FL#R02 | FL#R03 | FL#R04 | FL#R05 | FL#R06 | FL#R07 | FL#R08 | FL#R09 | FL#R10 |
|---|---|---|---|---|---|---|---|---|---|---|---|
| *Fraction of participants* | | $C = 0.4$ | | | | | | | | | |
| Accuracy ↑ | | 67.65 | 80.35 | 81.68 | 82.08 | 82.44 | 82.41 | 82.67 | 82.91 | 83.11 | 82.54 |
| Attack Acc ↑ | KED-MI | 42.13±2.64 | 53.53±3.59 | 54.60±3.70 | 55.27±4.89 | 56.53±3.94 | 58.87±4.96 | **61.93±5.26** | 54.00±4.85 | 58.73±3.34 | 61.07±3.68 |
| | KED-MI-HSIC | 45.07±3.58 | 59.53±4.22 | 55.73±3.72 | 58.13±3.66 | 60.13±4.33 | 59.33±4.33 | **64.80±4.12** | 55.80±3.57 | 57.53±2.72 | 61.07±2.34 |
| Attack Acc5 ↑ | KED-MI | 71.67±2.90 | 81.73±3.30 | 80.33±2.90 | 81.53±3.46 | 83.20±2.82 | 82.53±3.31 | **87.73±3.20** | 81.07±2.94 | 83.40±2.48 | 85.67±2.50 |
| | KED-MI-HSIC | 73.40±3.67 | 83.47±3.99 | 81.73±2.57 | 82.73±2.86 | 83.53±2.73 | 85.20±3.06 | **88.73±2.55** | 82.20±3.38 | 82.13±3.10 | 85.40±2.85 |
| FID ↓ | KED-MI | 236.8327 | 241.7872 | 229.6966 | 232.1761 | 226.3052 | 232.3101 | 247.5734 | 227.7192 | **225.6284** | 232.9999 |
| | KED-MI-HSIC | 228.8435 | 227.7244 | 250.3745 | 241.7729 | 242.425 | 233.486 | 224.7654 | **221.1877** | 222.9146 | 221.6223 |
| *Fraction of participants* | | $C = 0.6$ | | | | | | | | | |
| Accuracy ↑ | | 69.28 | 80.68 | 81.38 | 81.51 | 82.01 | 82.18 | 82.24 | 82.44 | 82.64 | 82.71 |
| Attack Acc ↑ | KED-MI | 49.47±4.53 | 57.60±4.81 | 53.47±4.24 | 59.13±3.39 | 58.33±5.02 | 56.80±3.56 | 56.67±4.47 | 55.47±4.47 | 56.53±4.08 | **59.73±3.16** |
| | KED-MI-HSIC | 50.60±3.95 | 58.67±3.71 | 56.40±3.73 | 59.13±4.82 | 59.20±4.89 | **60.40±4.93** | 60.33±5.18 | 56.53±3.09 | 56.40±3.24 | 58.47±3.97 |
| Attack Acc5 ↑ | KED-MI | 77.67±2.35 | 82.00±2.78 | 79.07±3.25 | 81.73±3.30 | 83.33±2.95 | 84.27±3.61 | 84.40±2.60 | 82.27±2.56 | 80.87±2.20 | **84.73±3.70** |
| | KED-MI-HSIC | 78.40±3.08 | 83.33±2.59 | 82.40±2.77 | 82.20±2.51 | 82.53±2.54 | 83.87±3.51 | **84.60±3.50** | 83.13±2.78 | 82.00±2.32 | 82.73±3.34 |
| FID ↓ | KED-MI | 255.9901 | 237.3064 | 238.6459 | 239.4178 | 254.6535 | 251.7509 | 240.175 | 243.6964 | 240.3951 | **234.203** |
| | KED-MI-HSIC | 229.8925 | 231.906 | 239.4226 | 249.6141 | 245.5274 | 230.4824 | 240.9596 | 235.8868 | **226.1562** | 227.872 |
| *Fraction of participants* | | $C = 1.0$ | | | | | | | | | |
| Accuracy ↑ | | 66.05 | 80.48 | 81.11 | 81.64 | 81.68 | 81.84 | 82.41 | 82.44 | 82.74 | 82.54 |
| Attack Acc ↑ | KED-MI | 39.53±1.87 | 53.47±2.91 | 55.60±2.64 | 56.93±3.19 | 56.80±4.54 | 54.67±3.40 | 56.60±2.97 | 53.13±3.33 | **57.80±4.04** | 56.73±2.62 |
| | KED-MI-HSIC | 41.73±1.76 | 57.80±3.07 | 55.07±4.31 | 59.33±3.45 | 57.47±2.42 | 57.07±3.88 | 59.13±3.95 | 55.73±2.87 | **60.13±3.53** | 59.47±3.30 |
| Attack Acc5 ↑ | KED-MI | 67.67±2.90 | 80.07±2.32 | 81.40±2.93 | 83.73±2.38 | 82.07±2.75 | 81.73±2.66 | 82.53±3.27 | 81.60±2.17 | **85.47±2.39** | 82.73±2.02 |
| | KED-MI-HSIC | 67.53±2.07 | 80.33±3.26 | 82.13±2.74 | 83.13±2.88 | 81.80±2.18 | 83.47±2.00 | 83.40±2.49 | 82.87±2.17 | **85.80±2.27** | 84.73±0.97 |
| FID ↓ | KED-MI | 241.0025 | 243.3093 | 235.3072 | 249.9985 | 247.8074 | 247.2511 | **227.6555** | 241.8614 | 244.159 | 248.9721 |
| | KED-MI-HSIC | 232.7846 | 242.5915 | 232.5486 | 245.5304 | 241.8464 | 241.6832 | **225.5088** | 238.4409 | 239.4701 | 248.7714 |

which are 87.73 and 88.73, respectively. The lowest top-5 attack accuracy appears at participant = 60%, which are 84.73 and 84.60, respectively.

## A.12 IMPACT OF LOCAL COMPUTATION ON FEDINVERSE+GMI ON CELEBA

The relevant experimental results are summarized in Table 16. We adjusted the local computation hyperparameters $E$ and $B$ and observed that the local computation on each client has less impact on FedInverse using GMI and GMI-HSIC. The highest GMI attack accuracy is 12.67 when we set $E = 30$ and $B = 128$. The attack accuracy on the rest of the settings varies from 10.50 to 12.17. For the GMI-HSIC the highest attack accuracy appears $E = 50$ and $B = 32$, which is 13.17. Others vary from 11.50 to 13.17. The top-5 attack accuracy varies from 24.00 to 27.80 and 24.67 to 30.00 for GMI and GMI-HSIC, respectively.

## A.13 IMPACT OF LOCAL COMPUTATION ON FEDINVERSE+KED-MI ON CELEBA

The relevant experimental results are summarized in Table 17, KED-MI and KED-MI-HSIC performed better than GMI and GMI-HSIC on CelebA. The highest KED-MI attack accuracy is 66.60 when we set $E = 50$ and $B = 32$. The attack accuracy on the rest of the settings varies from 57.80 to 61.40. For the KED-MI-HSIC the highest attack accuracy appears $E = 30$ and $B = 128$, which is 82.80. Others vary from 60.13 to 66.67. The top-5 attack accuracy varies from 84.27 to 88.40 and 84.87 to 89.27 for KED-MI and KED-MI-HSIC, respectively.

Table 16: FL privacy leakage indicated by Attack Acc/Acc5±standard deviation(%) and FID on CelebA via FedInverse using GMI and GMI-HSIC with varying local computation. Bold values denote the best metric results obtained by GMI or GMI-HSIC throughout the FL training epoch. The symbol ↓(↑) denotes that smaller (larger) values are favored.

| Metrics | Methods | FL#R01 | FL#R02 | FL#R03 | FL#R04 | FL#R05 | FL#R06 | FL#R07 | FL#R08 | FL#R09 | FL#R10 |
|---|---|---|---|---|---|---|---|---|---|---|---|
| *Local Computation* | | $(E, B) = (30, 32)$ | | | | | | | | | |
| Accuracy ↑ | | 67.25 | 81.15 | 81.44 | 81.68 | 82.38 | 82.51 | 82.67 | 82.57 | 82.67 | 82.74 |
| Attack Acc ↑ | GMI | 8.00±3.68 | 8.83±2.55 | 8.83±3.43 | 9.67±3.80 | 9.83±4.52 | 9.33±3.98 | 9.83±3.01 | 11.33±5.52 | **11.83±4.79** | 10.17±3.29 |
| | GMI-HSIC | 9.83±4.81 | 10.00±3.41 | 10.33±3.84 | 10.83±5.39 | 11.33±3.99 | 11.17±3.98 | 11.50±4.11 | 12.00±2.85 | **12.17±4.91** | 11.33±5.30 |
| Attack Acc5 ↑ | GMI | 22.33±4.96 | 22.00±6.71 | 24.33±8.04 | 23.17±5.54 | 22.17±4.81 | 21.17±4.62 | 23.50±4.84 | 23.00±5.69 | 26.33±6.26 | **27.83±7.42** |
| | GMI-HSIC | 24.67±4.66 | 25.50±6.83 | 26.83±4.92 | 24.17±6.58 | 24.83±4.69 | 27.00±9.11 | 24.33±5.70 | 24.00±6.66 | 26.83±6.66 | **27.50±6.55** |
| FID ↓ | GMI | 150.3791 | 151.9222 | 148.9057 | 153.0103 | 155.1253 | 154.3783 | 148.0305 | 149.3648 | 140.6815 | **139.4663** |
| | GMI-HSIC | 142.8629 | 148.9581 | 144.2583 | 149.5401 | 147.4151 | 150.0513 | 151.2198 | 148.6511 | 135.5822 | **126.5875** |
| *Local Computation* | | $(E, B) = (30, 64)$ | | | | | | | | | |
| Accuracy ↑ | | 66.05 | 80.48 | 81.11 | 81.64 | 81.68 | 81.84 | 82.41 | 82.44 | 82.74 | 82.54 |
| Attack Acc ↑ | GMI | 8.50±3.26 | 9.80±4.08 | 10.07±3.91 | 10.37±4.10 | 10.10±3.96 | 10.17±3.87 | 10.53±3.90 | 10.93±3.82 | **11.33±4.67** | 10.70±3.62 |
| | GMI-HSIC | 9.43±3.78 | 10.53±3.35 | 10.63±3.66 | 10.83±3.58 | 11.00±3.91 | 11.57±4.06 | 11.13±3.98 | 11.73±4.17 | **11.93±4.33** | 10.97±4.91 |
| Attack Acc5 ↑ | GMI | 22.07±4.62 | 23.80±5.24 | 23.80±5.63 | 23.93±6.03 | 24.07±5.73 | 24.20±5.42 | 24.47±4.99 | 24.83±6.57 | **25.40±7.35** | 25.20±5.15 |
| | GMI-HSIC | 22.70±5.46 | 24.93±5.47 | 25.00±5.60 | 25.03±5.26 | 25.23±5.18 | 25.63±5.68 | 25.63±5.03 | 25.57±6.18 | **26.77±5.84** | 25.90±6.54 |
| FID ↓ | GMI | 108.2603 | 108.5283 | 107.6128 | 108.7106 | 107.0249 | 106.814 | 104.0733 | 103.0706 | **97.1281** | 104.7775 |
| | GMI-HSIC | 106.3502 | 104.8022 | 105.0613 | 106.8688 | 106.6671 | 104.0582 | 104.404 | 101.0956 | **96.9064** | 102.5457 |
| *Local Computation* | | $(E, B) = (30, 128)$ | | | | | | | | | |
| Accuracy ↑ | | 68.21 | 81.11 | 82.01 | 82.24 | 82.61 | 82.54 | 82.67 | 82.81 | 82.91 | 82.94 |
| Attack Acc ↑ | GMI | 8.83±3.09 | 8.17±5.15 | 6.17±3.82 | 8.67±2.70 | **12.67±4.82** | 9.83±4.37 | 10.17±3.84 | 9.50±4.72 | 9.83±5.05 | 10.50±5.22 |
| | GMI-HSIC | 10.50±4.49 | 10.83±5.77 | 12.83±6.27 | 10.33±4.46 | 12.67±4.58 | 12.67±3.98 | 12.67±4.84 | **13.83±4.53** | 9.67±3.08 | 11.83±3.99 |
| Attack Acc5 ↑ | GMI | 21.67±5.74 | 22.67±7.03 | 21.33±7.34 | 23.50±6.08 | 25.17±6.77 | 24.17±8.57 | 24.50±6.47 | 27.50±7.36 | **27.67±6.99** | 24.67±5.55 |
| | GMI-HSIC | 25.50±6.80 | 25.33±6.65 | 25.83±6.48 | 25.33±5.55 | 26.33±4.93 | 26.00±6.20 | 26.83±5.28 | 28.17±8.47 | **30.00±6.27** | 27.67±6.94 |
| FID ↓ | GMI | 157.4958 | 155.0116 | 151.5519 | 149.7776 | 149.2006 | 147.1125 | 147.2644 | 148.8404 | **138.1305** | 143.8598 |
| | GMI-HSIC | 148.6452 | 147.9373 | 146.0720 | 145.2726 | 144.5028 | 143.5818 | 141.7064 | 142.0568 | **135.0851** | 138.9605 |
| *Local Computation* | | $(E, B) = (50, 32)$ | | | | | | | | | |
| Accuracy ↑ | | 68.05 | 82.08 | 82.51 | 82.81 | 83.44 | 83.64 | 83.74 | 83.97 | 83.97 | 84.17 |
| Attack Acc ↑ | GMI | 10.50±4.08 | 10.50±4.78 | 10.50±4.94 | 10.33±4.30 | 10.83±3.12 | 11.00±5.46 | 11.33±3.66 | **12.17±4.29** | 11.50±5.22 | 12.00±4.03 |
| | GMI-HSIC | 10.50±5.58 | 10.83±5.68 | 11.33±4.57 | 12.67±4.45 | 11.17±4.27 | 12.50±5.17 | 12.83±5.21 | **13.17±4.36** | 12.00±4.50 | 2.00±3.62 |
| Attack Acc5 ↑ | GMI | 24.83±5.20 | 22.83±7.10 | 24.83±7.21 | 25.00±4.45 | 24.00±6.99 | 22.83±6.22 | 23.67±4.57 | 26.67±5.64 | **27.67±8.17** | 24.33±6.19 |
| | GMI-HSIC | 25.67±7.35 | 26.17±5.08 | 25.17±7.86 | 25.17±7.99 | 24.83±7.30 | 26.17±5.32 | 27.50±8.19 | **28.33±7.47** | 26.50±5.78 | 27.83±6.76 |
| FID ↓ | GMI | 155.4550 | 152.0929 | 148.4526 | 144.9996 | 149.4272 | 151.4114 | 140.2202 | 144.2916 | **139.5381** | 147.8779 |
| | GMI-HSIC | 146.0104 | 143.0022 | 146.3147 | 151.5006 | 146.7224 | 143.4877 | 143.0034 | 153.7981 | 145.4033 | **135.2066** |
| *Local Computation* | | $(E, B) = (50, 64)$ | | | | | | | | | |
| Accuracy ↑ | | 67.58 | 80.11 | 80.75 | 81.05 | 81.58 | 81.54 | 81.91 | 82.08 | 82.14 | 82.48 |
| Attack Acc ↑ | GMI | 6.83±4.14 | 10.17±4.59 | 8.33±3.56 | 9.00±2.76 | 10.17±2.48 | 10.50±6.10 | 9.17±4.17 | **11.17±4.85** | 9.17±4.65 | 9.83±3.74 |
| | GMI-HSIC | 9.83±3.82 | 10.83±3.19 | 9.67±2.97 | 9.83±3.92 | 10.67±4.11 | 11.33±5.15 | 10.83±3.94 | **11.67±3.89** | 10.00±2.66 | 10.00±3.05 |
| Attack Acc5 ↑ | GMI | 20.50±6.23 | 22.83±5.11 | 21.00±4.48 | 22.83±6.19 | **25.50±5.34** | 23.50±7.69 | 25.33±4.64 | 25.33±4.13 | 23.50±7.55 | 21.83±6.55 |
| | GMI-HSIC | 22.33±5.77 | 24.33±6.69 | 23.33±3.33 | 24.33±5.91 | 23.83±4.52 | 23.00±5.95 | 24.33±5.24 | 24.33±5.15 | **26.00±7.36** | 24.67±6.64 |
| FID ↓ | GMI | 161.0902 | 158.4809 | 159.8758 | 149.9713 | 154.8924 | 154.7554 | 157.3184 | **142.5553** | 145.9331 | 154.8426 |
| | GMI-HSIC | 154.5773 | 154.9725 | 151.9411 | 156.5869 | 151.3138 | 146.3411 | 150.6333 | 149.2384 | 144.2459 | **143.9670** |
| *Local Computation* | | $(E, B) = (50, 128)$ | | | | | | | | | |
| Accuracy ↑ | | 68.21 | 79.42 | 80.35 | 80.71 | 81.21 | 81.48 | 81.44 | 81.54 | 81.78 | 81.94 |
| Attack Acc ↑ | GMI | 8.50±3.30 | 8.17±3.13 | 9.17±5.62 | 7.83±4.36 | 9.00±2.78 | 8.83±4.99 | 10.17±5.05 | 9.33±3.90 | 8.33±4.34 | **10.50±4.76** |
| | GMI-HSIC | 10.50±4.46 | 10.83±3.65 | 9.17±3.86 | 9.67±3.67 | 9.50±4.60 | 10.83±6.02 | 11.17±5.58 | **11.50±4.82** | 10.83±4.78 | 11.17±3.63 |
| Attack Acc5 ↑ | GMI | 20.83±4.34 | 20.33±5.49 | 22.83±7.05 | 21.83±3.33 | 22.33±8.31 | 22.67±6.03 | 24.17±7.43 | 22.33±8.93 | **24.00±7.17** | 23.50±4.48 |
| | GMI-HSIC | 22.83±5.53 | 22.67±8.51 | 22.83±5.14 | 22.83±6.40 | 24.33±7.90 | 23.33±7.03 | 24.00±6.90 | 24.33±6.75 | **24.67±4.86** | 24.33±4.20 |
| FID ↓ | GMI | 155.7808 | 157.3235 | 156.1449 | 153.3072 | 153.2049 | 150.1078 | 140.1049 | 147.2408 | 150.4086 | **144.1966** |
| | GMI-HSIC | 150.0434 | 146.9458 | 149.4768 | 148.7402 | 148.6756 | 148.4312 | 147.1038 | 146.8517 | 145.7551 | **135.8389** |

Table 17: FL privacy leakage indicated by Attack Acc/Acc5±standard deviation(%) and FID on CelebA via FedInverse using KED-MI and KED-MI-HSIC with varying local computation. Bold values denote the best metric results obtained by GMI or GMI-HSIC throughout the FL training epoch. The symbol ↓(↑) denotes that smaller (larger) values are favored.

| Metrics | Methods | FL#R01 | FL#R02 | FL#R03 | FL#R04 | FL#R05 | FL#R06 | FL#R07 | FL#R08 | FL#R09 | FL#R10 |
|---|---|---|---|---|---|---|---|---|---|---|---|
| *Local Computation* | | $(E, B) = (30, 32)$ | | | | | | | | | |
| Accuracy ↑ | | 67.25 | 81.15 | 81.44 | 81.68 | 82.38 | 82.51 | 82.67 | 82.57 | 82.67 | 82.74 |
| Attack Acc ↑ | KED-MI | 35.40±3.31 | 56.40±4.23 | 58.00±4.41 | 57.47±4.91 | 56.13±3.08 | 59.00±5.79 | 54.47±5.76 | 55.20±4.62 | 60.27±3.66 | **61.40±3.72** |
| | KED-MI-HSIC | 37.00±2.56 | 53.73±5.20 | 56.07±3.36 | 56.33±5.27 | 58.27±4.64 | 59.73±3.11 | 55.27±4.46 | 56.67±2.64 | 58.87±4.74 | **61.07±4.43** |
| Attack Acc5 ↑ | KED-MI | 64.07±3.92 | 81.53±3.35 | 82.07±3.75 | 82.07±3.79 | 81.27±4.17 | 83.47±3.20 | 82.87±3.29 | 82.33±3.78 | 84.20±3.12 | **84.27±2.53** |
| | KED-MI-HSIC | 66.13±3.10 | 81.07±3.76 | 80.93±3.18 | 82.60±3.18 | 82.60±3.18 | 83.27±2.90 | 82.93±2.22 | 82.67±4.50 | 84.20±2.52 | **84.87±2.33** |
| FID ↓ | KED-MI | 250.8308 | 237.7906 | 239.5953 | 225.7642 | 231.51 | 247.7834 | **219.4812** | 236.7981 | 230.0569 | 232.8372 |
| | KED-MI-HSIC | 242.823 | 235.9567 | 232.2535 | 223.7007 | 233.6536 | 244.4438 | **214.8585** | 233.2556 | 223.4137 | 230.3397 |
| *Local Computation* | | $(E, B) = (30, 64)$ | | | | | | | | | |
| Accuracy ↑ | | 66.05 | 80.48 | 81.11 | 81.64 | 81.68 | 81.84 | 82.41 | 82.44 | 82.74 | 82.54 |
| Attack Acc ↑ | KED-MI | 39.53±1.87 | 53.47±2.91 | 55.60±2.64 | 56.93±3.19 | 56.80±4.54 | 54.67±3.40 | 56.60±2.97 | 53.13±3.33 | **57.80±4.04** | 56.73±2.62 |
| | KED-MI-HSIC | 41.73±1.76 | 57.80±3.07 | 55.07±4.31 | 59.33±3.45 | 57.47±2.42 | 57.07±3.88 | 59.13±3.95 | 55.73±2.87 | **60.13±3.53** | 59.47±3.30 |
| Attack Acc5 ↑ | KED-MI | 67.67±2.90 | 80.07±2.32 | 81.40±2.93 | 83.73±2.38 | 82.07±2.75 | 81.73±2.66 | 82.53±3.27 | 81.60±2.17 | **85.47±2.39** | 82.73±2.02 |
| | KED-MI-HSIC | 67.53±2.07 | 80.33±3.26 | 82.13±2.74 | 83.13±2.88 | 81.80±2.18 | 83.47±2.00 | 83.40±2.49 | 82.87±2.17 | **85.80±2.27** | 84.73±0.97 |
| FID ↓ | KED-MI | 241.0025 | 243.3093 | 235.3072 | 249.9985 | 247.8074 | 247.2511 | **227.6555** | 241.8614 | 244.159 | 248.9721 |
| | KED-MI-HSIC | 232.7846 | 242.5915 | 232.5486 | 245.5304 | 241.8464 | 241.6832 | **225.5088** | 238.4409 | 239.4701 | 248.7714 |
| *Local Computation* | | $(E, B) = (30, 128)$ | | | | | | | | | |
| Accuracy ↑ | | 68.21 | 81.11 | 82.01 | 82.24 | 82.61 | 82.54 | 82.67 | 82.81 | 82.91 | 82.94 |
| Attack Acc ↑ | KED-MI | 40.13±4.23 | 59.00±4.31 | 51.73±5.32 | 54.33±4.31 | 55.13±4.45 | 54.73±4.75 | 58.00±3.32 | **60.00±5.15** | 58.40±4.58 | 59.80±3.21 |
| | KED-MI-HSIC | 44.00±3.16 | 56.13±4.05 | 57.53±4.37 | 56.73±3.81 | 57.73±2.90 | 58.47±3.30 | 58.20±3.47 | **82.80±3.34** | 58.13±4.24 | 59.73±3.74 |
| Attack Acc5 ↑ | KED-MI | 67.47±3.89 | 83.93±2.73 | 77.20±4.48 | 82.00±3.93 | 84.53±3.54 | 84.27±2.43 | 84.33±3.64 | **84.87±2.12** | 83.73±3.85 | 80.4±3.45 |
| | KED-MI-HSIC | 70.47±3.34 | 83.33±2.90 | 83.87±3.78 | 83.67±5.08 | 84.13±2.52 | 82.87±3.69 | 84.07±2.07 | **85.40±4.08** | 83.67±3.12 | 84.47±3.03 |
| FID ↓ | KED-MI | 250.4548 | 247.27 | 233.0923 | 257.5443 | 239.2134 | 236.8182 | 244.7302 | **226.5913** | 239.1755 | 234.661 |
| | KED-MI-HSIC | 250.0051 | 231.995 | 242.9628 | 248.1224 | 235.3339 | 227.7765 | 233.6334 | 233.3824 | 239.8843 | **225.5299** |
| *Local Computation* | | $(E, B) = (50, 32)$ | | | | | | | | | |
| Accuracy ↑ | | 68.05 | 82.08 | 82.51 | 82.81 | 83.44 | 83.64 | 83.74 | 83.97 | 83.97 | 84.17 |
| Attack Acc ↑ | KED-MI | 40.80±3.80 | 57.73±3.51 | 55.20±5.83 | 61.33±4.05 | 58.60±3.47 | 56.40±4.99 | 60.53±3.67 | 60.47±3.50 | 62.00±5.13 | **66.60±2.94** |
| | KED-MI-HSIC | 42.80±4.90 | 57.67±3.42 | 57.53±5.26 | 62.73±4.59 | 58.93±2.52 | 59.53±3.24 | 64.67±4.09 | 62.87±4.11 | 62.80±3.75 | **66.67±5.43** |
| Attack Acc5 ↑ | KED-MI | 68.60±3.45 | 82.67±2.97 | 80.93±4.42 | 86.27±3.42 | 82.33±3.64 | 82.07±3.85 | 84.27±2.63 | 84.00±3.11 | 85.33±3.14 | **88.40±2.73** |
| | KED-MI-HSIC | 69.53±3.34 | 82.67±3.10 | 84.13±3.46 | 85.93±3.83 | 85.47±3.11 | 83.47±2.16 | 87.33±1.73 | 86.07±2.34 | 86.00±2.89 | **89.27±3.01** |
| FID ↓ | KED-MI | 233.0825 | 237.9213 | 219.7000 | 231.326 | 245.0408 | 235.0954 | 234.5587 | **216.0964** | 220.189 | 209.7383 |
| | KED-MI-HSIC | 228.4920 | 234.2204 | 221.8825 | 228.2933 | 242.7171 | 230.3700 | 226.0333 | 215.3089 | 220.931 | **210.3917** |
| *Local Computation* | | $(E, B) = (50, 64)$ | | | | | | | | | |
| Accuracy ↑ | | 67.58 | 80.11 | 80.75 | 81.05 | 81.58 | 81.54 | 81.91 | 82.08 | 82.14 | 82.48 |
| Attack Acc ↑ | KED-MI | 39.93±2.87 | 51.93±4.53 | 53.93±4.93 | 55.87±3.39 | 51.00±3.28 | 54.73±4.55 | 55.47±3.49 | **58.40±3.47** | 57.93±3.48 | 57.20±4.99 |
| | KED-MI-HSIC | 43.33±3.07 | 54.53±3.69 | 52.60±3.91 | 57.80±4.54 | 53.73±4.04 | 56.13±5.13 | 58.80±4.67 | 60.00±3.60 | **60.33±4.76** | 59.80±5.00 |
| Attack Acc5 ↑ | KED-MI | 66.53±2.26 | 78.53±3.30 | 79.60±3.48 | 82.07±2.81 | 79.60±3.12 | 81.67±2.85 | 80.20±3.81 | **84.33±3.10** | 82.53±2.41 | 83.07±3.36 |
| | KED-MI-HSIC | 69.73±3.76 | 80.60±4.29 | 79.47±2.65 | 82.60±2.77 | 80.27±4.05 | 83.53±3.04 | 82.20±3.68 | 83.73±2.01 | **84.97±2.48** | 83.53±2.57 |
| FID ↓ | KED-MI | 230.4009 | 231.4905 | 261.1751 | 241.4455 | 228.279 | 233.7109 | 235.3763 | 245.8801 | **214.8504** | 246.8346 |
| | KED-MI-HSIC | 231.0736 | 228.222 | 258.8327 | 239.9759 | 221.3416 | 231.2333 | 230.7373 | 236.4309 | **211.564** | 242.5042 |
| *Local Computation* | | $(E, B) = (50, 128)$ | | | | | | | | | |
| Accuracy ↑ | | 68.21 | 79.42 | 80.35 | 80.71 | 81.21 | 81.48 | 81.44 | 81.54 | 81.78 | 81.94 |
| Attack Acc ↑ | KED-MI | 43.00±3.09 | 53.60±3.47 | 53.73±3.93 | 54.00±3.40 | 56.60±3.85 | **59.60±4.30** | 56.53±3.86 | 57.87±3.74 | 56.00±4.97 | 56.93±4.02 |
| | KED-MI-HSIC | 42.20±4.02 | 53.67±4.72 | 55.00±3.78 | 58.07±4.97 | 56.20±3.22 | **61.20±4.35** | 58.27±3.34 | 58.67±2.89 | 57.80±4.25 | 57.00±3.54 |
| Attack Acc5 ↑ | KED-MI | 69.67±3.44 | 79.60±2.68 | 81.67±3.59 | 81.07±3.59 | 81.33±4.34 | 83.00±4.27 | 82.73±2.51 | **85.13±2.25** | 82.07±3.62 | 81.60±3.33 |
| | KED-MI-HSIC | 71.80±2.99 | 79.80±3.15 | 83.53±3.45 | 84.00±3.05 | 82.73±2.74 | **85.27±3.68** | 83.53±2.58 | 84.40±2.30 | 83.60±2.90 | 82.33±3.19 |
| FID ↓ | KED-MI | 251.7551 | 245.1883 | 240.308 | 245.6284 | 244.8232 | **226.9266** | 236.3251 | 229.9813 | 238.5785 | 244.5931 |
| | KED-MI-HSIC | 241.6745 | 243.0175 | 233.0712 | 244.5804 | 237.807 | **222.5282** | 233.1085 | 227.386 | 231.779 | 238.2023 |

## A.14 IMPACT OF DEFENSE METHODS ON FEDINVERSE+GMI ON CELEBA

We used two latest prevailing defense methods, MID and BiDO, to train the FL models. Meanwhile, we launch the GMI attacks on the updated global models in each federated communication round to see if these defense training schemes can still work in FL settings. The relevant observations are summarized in Table 18. The FedInverse with GMI and GMI-HSIC attacked the FL with MID on the third training round and reached attack accuracy of 35.67 and 38.67, respectively. Similar to MID settings, GMI and GMI-HSIC attacked the FL with BiDO on the third training round and reached attack accuracy of 34.33 and 36.83, respectively. For top-5 attack accuracy, GMI reached 56.00 in round 3, and GMI-HSIC reached 56.83 in round 3 with MID. Additionally, GMI reached 56.83 in round 3, and GMI-HSIC reached 59.00 in round 3 with BiDO. The attack performance decreased dramatically, which shows that the SOTA MI defense approaches can partially defend against FedInverse+GMI attacks on FL settings on CelebA dataset. According to the results from MNIST; the experiments show that the SOTA MI defense approaches are data and model-oriented to defend against FedInverse+GMI attacks.

Table 18: FL privacy leakage indicated by Attack Acc/Acc5± standard deviation(%) and FID on CelebA via FedInverse using GMI and GMI-HSIC with two diverse defense methods: MID and BiDO. Bold values denote the best metric results obtained by GMI or GMI-HSIC throughout the FL training epoch. The symbol ↓(↑) denotes that smaller (larger) values are favored.

| Metrics | Methods | FL#R01 | FL#R02 | FL#R03 | FL#R04 | FL#R05 | FL#R06 | FL#R07 | FL#R08 | FL#R09 | FL#R10 |
|---|---|---|---|---|---|---|---|---|---|---|---|
| *Defense Method* | | MID | | | | | | | | | |
| Accuracy ↑ | | 0.13 | 0.09 | 62.1 | 82.38 | 84.77 | 84.77 | 85.2 | 85.8 | 85.6 | 86.13 |
| Attack Acc ↑ | GMI | 0.00±0.00 | 3.67±3.41 | **35.67±7.01** | 15.33±4.39 | 7.67±3.01 | 7.33±2.81 | 8.33±4.98 | 6.17±4.36 | 6.33±5.21 | 7.00±2.17 |
| | GMI-HSIC | 0.33±0.46 | 5.83±3.81 | **38.67±8.50** | 15.17±4.38 | 10.67±4.36 | 8.83±4.55 | 9.67±3.63 | 7.50±3.88 | 8.00±3.34 | 8.00±4.11 |
| Attack Acc5 ↑ | GMI | 0.17±0.37 | 14.00±4.37 | **56.00±7.40** | 32.17±4.85 | 20.00±5.14 | 17.67±4.78 | 20.50±6.74 | 17.83±6.73 | 16.83±6.52 | 19.00±6.90 |
| | GMI-HSIC | 0.50±0.83 | 15.00±4.29 | **56.83±5.96** | 33.50±6.20 | 25.83±7.25 | 24.83±9.07 | 23.33±4.85 | 20.67±4.26 | 19.17±4.90 | 18.33±4.47 |
| FID ↓ | GMI | 262.1850 | 304.7803 | **144.9108** | 151.2462 | 169.7602 | 175.6377 | 170.8262 | 179.5037 | 178.1989 | 183.8218 |
| | GMI-HSIC | 239.9734 | 303.0943 | **140.8861** | 147.7921 | 155.8281 | 163.6606 | 171.9034 | 173.3850 | 177.6978 | 181.2256 |
| *Defense Method* | | BiDO | | | | | | | | | |
| Accuracy ↑ | | 0.09 | 56.88 | 83.87 | 84.90 | 85.03 | 85.53 | 85.50 | 86.00 | 86.07 | 85.97 |
| Attack Acc ↑ | GMI | 0.33±0.75 | **34.33±8.74** | 8.50±3.79 | 5.50±3.13 | 6.50±2.54 | 6.17±3.56 | 7.83±3.24 | 5.83±3.05 | 5.50±2.70 | 5.00±4.02 |
| | GMI-HSIC | 0.33±0.46 | **36.83±4.77** | 9.67±4.19 | 8.67±4.00 | 8.50±4.65 | 6.83±2.93 | 7.67±3.96 | 8.17±3.56 | 7.67±3.26 | 7.67±3.88 |
| Attack Acc5 ↑ | GMI | 0.50±0.83 | **56.50±7.19** | 22.33±4.06 | 16.83±5.09 | 19.67±6.71 | 19.33±5.05 | 19.00±5.62 | 18.67±4.71 | 19.17±5.52 | 19.67±7.51 |
| | GMI-HSIC | 0.50±0.83 | **59.00±5.54** | 24.50±5.12 | 20.50±4.74 | 22.83±5.84 | 19.00±5.10 | 20.67±4.95 | 20.67±6.73 | 21.50±5.26 | 19.17±5.29 |
| FID ↓ | GMI | 307.7120 | **154.6588** | 181.7940 | 209.6793 | 198.4003 | 195.2061 | 196.4863 | 199.7902 | 202.7657 | 206.6921 |
| | GMI-HSIC | 280.9487 | **154.2095** | 180.0081 | 196.7448 | 193.8378 | 187.7461 | 193.5657 | 198.1558 | 194.2287 | 195.9553 |

## A.15 IMPACT OF DEFENSE METHODS ON FEDINVERSE+KED-MI ON CELEBA

We used two latest prevailing defense methods, MID and BiDO, to train the FL models. Meanwhile, we launch the KED-MI attacks on the updated global models in each federated communication round to see if these defense training schemes can still work in FL settings. The relevant observations are summarized in Table 19. The FedInverse with KED-MI and KED-MI-HSIC attacked the FL with MID on the third training round and reached attack accuracy of 0.40 and 0.47, respectively. Similar to MID settings, KED-MI and KED-MI-HSIC attacked the FL with BiDO on the third training round and reached attack accuracy of 0.13 and 0.53, respectively. For top-5 attack accuracy, KED-MI and KED-MI-HSIC both reached 1.07 in round 1 with MID. Additionally, KED-MI reached 0.93 in round 2, and GMI-HSIC reached 2.07 in round 1 with BiDO. The attack performance decreased dramatically, which shows that the SOTA MI defense approaches can successfully defend against FedInverse+KED-MI attacks on FL settings on CelebA dataset. According to the results from MNIST; the experiments show that the SOTA MI defense approaches are data and model-oriented to defend against FedInverse+KED-MI attacks.

Table 19: FL privacy leakage indicated by Attack Acc/Acc5± standard deviation(%) and FID on CelebA via FedInverse using KED-MI and KED-MI-HSIC with two diverse defense methods: MID and BiDO. Bold values denote the best metric results obtained by KED-MI or KED-MI-HSIC throughout the FL training epoch. The symbol ↓(↑) denotes that smaller (larger) values are favored.

| Metrics | Methods | FL#R01 | FL#R02 | FL#R03 | FL#R04 | FL#R05 | FL#R06 | FL#R07 | FL#R08 | FL#R09 | FL#R10 |
|---|---|---|---|---|---|---|---|---|---|---|---|
| *Defense Method* | | MID | | | | | | | | | |
| Accuracy ↑ | | 0.13 | 0.09 | 62.1 | 82.38 | 84.77 | 84.77 | 85.2 | 85.8 | 85.6 | 86.13 |
| Attack Acc ↑ | KED-MI | **0.40±0.14** | 0.13±0.18 | 0.00±0.00 | 0.13±0.18 | 0.00±0.00 | 0.00±0.00 | 0.00±0.00 | 0.07±0.14 | 0.00±0.00 | 0.13±0.18 |
| | KED-MI-HSIC | 0.40±0.14 | 0.20±0.18 | **0.47±0.18** | 0.07±0.14 | 0.20±0.33 | 0.33±0.29 | 0.00±0.00 | 0.27±0.14 | 0.20±0.18 | 0.33±0.00 |
| Attack Acc5 ↑ | KED-MI | **1.07±0.46** | 0.27±0.33 | 0.07±0.14 | 0.73±0.74 | 0.13±0.29 | 0.47±0.58 | 0.27±0.33 | 0.53±0.18 | 0.60±0.44 | 0.60±0.44 |
| | KED-MI-HSIC | **1.07±0.33** | 0.60±0.33 | 0.73±0.44 | 0.87±0.92 | 0.80±0.84 | 0.67±0.48 | 0.33±0.00 | 0.87±0.44 | 0.53±0.29 | 0.73±0.44 |
| FID ↓ | KED-MI | 1066.2796 | 584.8376 | 626.5133 | **494.0431** | 702.2928 | 641.6904 | 691.3997 | 748.0618 | 748.3944 | 563.5676 |
| | KED-MI-HSIC | 1014.4789 | 585.5836 | 604.0114 | **492.146** | 725.212 | 667.3054 | 668.4476 | 755.4100 | 735.7304 | 571.0013 |
| *Defense Method* | | BiDO | | | | | | | | | |
| Accuracy ↑ | | 0.09 | 56.88 | 83.87 | 84.90 | 85.03 | 85.53 | 85.50 | 86.00 | 86.07 | 85.97 |
| Attack Acc ↑ | KED-MI | 0.00±0.00 | **0.13±0.29** | 0.00±0.00 | 0.00±0.00 | **0.13±0.29** | 0.00±0.00 | 0.00±0.00 | 0.00±0.00 | 0.00±0.00 | 0.00±0.00 |
| | KED-MI-HSIC | 0.40±0.62 | **0.53±0.64** | 0.20±0.18 | 0.13±0.18 | 0.00±0.00 | 0.27±0.33 | 0.40±0.51 | 0.60±0.43 | 0.13±0.18 | 0.40±0.14 |
| Attack Acc5 ↑ | KED-MI | 0.53±0.69 | **0.93±0.71** | 0.53±0.33 | 0.33±0.42 | 0.73±0.82 | 0.20±0.33 | 0.60±0.33 | 0.60±0.36 | 0.20±0.33 | 0.47±0.53 |
| | KED-MI-HSIC | **2.07±0.57** | 1.40±0.66 | 0.73±0.74 | 0.80±0.48 | 0.67±0.63 | 1.20±1.32 | 0.93±0.48 | 1.40±0.74 | 0.73±0.66 | 1.07±0.82 |
| FID ↓ | KED-MI | **419.2799** | 738.1139 | 876.206 | 1001.516 | 940.0084 | 594.5001 | 593.23 | 741.6032 | 624.6205 | 582.4685 |
| | KED-MI-HSIC | **415.9628** | 730±9202 | 880.1697 | 858.9476 | 946.7589 | 575.2957 | 605.4646 | 748.6841 | 596.2671 | 570.9718 |

Table 20: FL privacy leakage indicated by Attack Acc/Acc5±standard deviation(%) and FID on CIFAR10 via FedInverse using GMI and GMI-HSIC. Bold values denote the best metric results obtained by GMI or GMI-HSIC throughout the FL training epoch. The symbol ↓(↑) denotes that smaller (larger) values are favored.

| Metrics | Methods | FL#R01 | FL#R02 | FL#R03 | FL#R04 | FL#R05 | FL#R06 | FL#R07 | FL#R08 | FL#R09 | FL#R10 |
|---|---|---|---|---|---|---|---|---|---|---|---|
| Accuracy ↑ | | 69.16 | 76.36 | 80.36 | 80.76 | 70.48 | 84.44 | 82.39 | 85.88 | 86.60 | 86.60 |
| Attack Acc ↑ | GMI | 10.00±14.14 | 14.00±13.50 | 14.00±13.50 | 16.00±15.78 | 16.00±8.43 | 12.00±13.98 | 20.00±18.86 | 14.00±13.50 | 20.00±16.33 | **22.00±14.76** |
| | GMI-HSIC | 24.00±15.78 | 24.00±12.65 | 24.00±15.78 | 28.00±13.98 | 18.00±14.76 | 20.00±16.33 | 20.00±16.33 | 20.00±16.33 | 26.00±13.50 | **30.00±17.00** |
| Attack Acc5 ↑ | GMI | 54.00±13.50 | 54.00±13.50 | 54.00±13.50 | 56.00±15.78 | 52.00±19.32 | 58.00±17.51 | 62.00±14.76 | **66.00±16.47** | 58.00±11.35 | 56.00±8.43 |
| | GMI-HSIC | 62.00±14.76 | 64.00±12.65 | 62.00±11.35 | 66.00±13.50 | 66.00±16.47 | 60.00±13.33 | 64.00±12.65 | **72.00±13.98** | 62.00±14.76 | 62.00±17.51 |
| FID ↓ | GMI | 8.9866 | 8.0442 | 8.0731 | 8.1162 | 7.9437 | 7.8523 | 7.2196 | 7.8648 | 7.3911 | **7.1638** |
| | GMI-HSIC | 7.8270 | 7.2996 | 7.2069 | 7.4807 | 7.0648 | 6.6097 | 6.1103 | 6.8338 | 6.8324 | **6.0622** |

## A.16 FEDINVERSE+GMI ATTACK PERFORMANCE ON CIFAR-10

To further evaluate the FedIverse performance, we tested the FedInverse+GMI on CIFAR-10 Krizhevsky et al. (2009). The results of FedInverse attack using GMI and GMI-HSIC on CIFAR-10 are illustrated in Table 20. The GMI achieved the highest attack accuracy in FL training round ten, which is 22.00, and GMI-HSIC further improved the attack accuracy to 30.00, which is an improvement of 36%. For top-5 attack accuracy, GMI reached the pick of 66.00 in FL training round eight, and GMI-HSIC is 72.00, which improves 9%.

## A.17 FEDINVERSE+KED-MI ATTACK PERFORMANCE ON CIFAR-10

Comparing to FedInverse+GMI, FedInverse+KED-MI successfully attacked the FL settings. The performance evaluation of FedInverse using KED-MI attack on CIFAR10 is presented in Table 21. The FedInverse+KED-MI attack accuracy achieved 51.92 in FL training round seven.

Table 21: FL privacy leakage indicated by Attack Acc/Acc5±standard deviation(%) and FID on CIFAR10 via FedInverse using KED-MI and KED-MI-HSIC. Bold values denote the best metric results obtained by KED-MI or KED-MI-HSIC throughout the FL training epoch. The symbol ↓(↑) denotes that smaller (larger) values are favored.

| Metrics | Methods | FL#R01 | FL#R02 | FL#R03 | FL#R04 | FL#R05 | FL#R06 | FL#R07 | FL#R08 | FL#R09 | FL#R10 |
|---|---|---|---|---|---|---|---|---|---|---|---|
| Accuracy ↑ | | 69.16 | 76.36 | 80.36 | 80.76 | 70.48 | 84.44 | 82.39 | 85.88 | 86.60 | 86.60 |
| Attack Acc ↑ | KED-MI | 24.01±11.85 | 42.85±14.18 | 43.44±12.76 | 32.77±12.98 | 48.53±15.41 | 37.03±12.95 | **51.92±16.49** | 24.41±8.27 | 35.28±12.85 | 21.57±9.44 |
| | KED-MI-HSIC | 39.16±10.79 | 49.13±12.20 | 44.20±14.40 | 41.83±13.54 | 51.49±16.27 | 40.01±13.73 | **63.25±13.22** | 36.09±13.32 | 40.93±12.93 | 24.75±12.29 |
| Attack Acc5 ↑ | KED-MI | 81.88±7.05 | 73.27±11.93 | 82.84±6.99 | 77.27±10.89 | 88.05±5.92 | **98.24±4.96** | 92.85±3.37 | 75.89±12.72 | 72.51±10.13 | 71.36±12.86 |
| | KED-MI-HSIC | 87.28±7.10 | 81.44±7.61 | 85.15±8.61 | 97.23±6.39 | 96.84±7.15 | **99.09±4.12** | 93.65±4.70 | 86.75±8.85 | 80.51±9.45 | 74.61±15.72 |
| FID ↓ | KED-MI | 8.0518 | 10.2236 | 7.5784 | 5.7769 | 8.3396 | **5.5596** | 5.5913 | 9.4793 | 7.4885 | 10.4272 |
| | KED-MI-HSIC | 5.7406 | 6.0592 | 5.3121 | 5.3495 | 7.6287 | 5.1709 | **4.5572** | 6.9419 | 6.1933 | 10.1933 |

FedInverse+KED-MI-HSIC further pushed the attack accuracy to 63.25, which is a 21% improvement. For top-5 attack accuracy, both FedInverse+KED-MI and FedInverse+KED-MI-HSIC achieved extremely high performance, which are 98.24 and 99.09, respectively.

## A.18    IMPACT OF DEFENSE METHODS ON FEDINVERSE+GMI ON CIFAR-10

We used two latest prevailing defense methods, MID and BiDO, to train the FL models. Meanwhile, we launch the GMI attacks on the updated global models in each federated communication round to see if these defense training schemes can still work in FL settings. The relevant observations are summarized in Table 22. The FedInverse with GMI and GMI-HSIC attacked the FL with MID on the ninth training round and reached attack accuracy of 24.00 and 28.00, respectively. Similar to MID settings, GMI and GMI-HSIC successfully attacked the FL with BiDO on the ninth training round and reached attack accuracy of 28.00 and 38.00, respectively. For top-5 attack accuracy, GMI reached 66.00 in round 6, and GMI-HSIC reached 68.00 in round 6 with MID, and GMI reached 66.00 in round 9, and GMI-HSIC reached 70.00 in round 9 with BiDO defense settings. The results show that the SOTA defense approaches has less impact on defending against FedInverse+GMI attacks on FL settings on CIFAR-10 dataset.

## A.19    IMPACT OF DEFENSE METHODS ON FEDINVERSE+KED-MI

The relevant observations are summarized in Table 23. The FedInverse with KED-MI and KED-MI-HSIC successfully attacked the FL with MID on the ninth training round and reached attack accuracy of 45.17 and 63.75, respectively. Similar to MID settings, KED-MI and KED-MI-HSIC successfully attacked the FL with BiDO and reached attack accuracy of 63.75 and 72.25, respectively. For top-5 attack accuracy, KED-MI reached 97.43 in round 10, and KED-MI-HSIC also reached 99.23 in round 10 with MID. In addition, with BiDO defense settings, KED-MI reached 87.53 in round 7, and KED-MI-HSIC also reached 99.89 in round 4. The results show that the SOTA defense approach cannot defend against FedInverse+KED-MI attacks on FL settings on CIFAR-10 dataset.

# B    MORE RELATED WORKS

## B.1    FEDERATED LEARNING

With the rapid development of machine learning technology, applications such as automatic driving, face recognition, and natural language processing have brought great convenience to people's lives. Machine learning is data-driven, which learns a model from a large amount of data to achieve various tasks. Traditional machine learning adopts a centralized architecture that collects training data, trains models, and deploys them in various application scenarios after the model training is completed Zhang et al. (2021). Although centralized machine learning has excellent learning performance, it also faces problems such as data silos and data privacy problems. Data silos refer to the phenomenon that data is stored in different forms and managed by different agencies, resulting in these data that

Table 22: FL privacy leakage indicated by Attack Acc/Acc5± standard deviation(%) and FID on CIFAR10 via FedInverse using GMI and GMI-HSIC with two diverse defense methods: MID and BiDO. Bold values denote the best metric results obtained by GMI or GMI-HSIC throughout the FL training epoch. The symbol ↓(↑) denotes that smaller (larger) values are favored.

| Metrics | Methods | FL#R01 | FL#R02 | FL#R03 | FL#R04 | FL#R05 | FL#R06 | FL#R07 | FL#R08 | FL#R09 | FL#R10 |
|---|---|---|---|---|---|---|---|---|---|---|---|
| *Defense Method* | | MID | | | | | | | | | |
| Accuracy ↑ | | 66.67 | 75.22 | 79.92 | 53.80 | 82.39 | 84.84 | 84.56 | 86.11 | 87.14 | 79.97 |
| Attack Acc ↑ | GMI | 12.00±10.33 | 18.00±11.35 | 12.00±10.33 | 16.00±18.38 | 16.00±12.65 | 20.00±9.43 | 18.00±19.89 | 20.00±13.33 | **24.00±20.66** | 16.00±15.78 |
| | GMI-HSIC | 18.00±11.35 | 18.00±14.76 | 18.00±14.76 | 16.00±15.78 | 24.00±12.65 | 24.00±18.38 | 22.00±14.76 | 24.00±22.71 | **28.00±19.32** | 22.00±14.76 |
| Attack Acc5 ↑ | GMI | 52.00±16.87 | 56.00±15.78 | 50.00±17.00 | 62.00±22.01 | 58.00±14.76 | **66.00±16.47** | 58.00±14.76 | 54.00±18.97 | 62.00±14.76 | 62.00±14.76 |
| | GMI-HSIC | 52.00±19.32 | 56.00±15.78 | 62.00±11.35 | 64.00±20.66 | 60.00±16.33 | **68.00±13.33** | 62.00±14.76 | 62.00±19.89 | 60.00±18.86 | 64.00±12.65 |
| FID ↓ | GMI | 8.1742 | 8.6426 | 8.2033 | 7.7860 | 6.8375 | 7.8042 | 7.1293 | 7.4087 | **6.7212** | 8.4300 |
| | GMI-HSIC | 7.5186 | 7.5598 | 7.3069 | 7.2506 | 6.9614 | 7.0091 | 7.4685 | **6.4709** | 6.6456 | 8.3597 |
| *Defense Method* | | BiDO | | | | | | | | | |
| Accuracy ↑ | | 71.16 | 31.08 | 83.52 | 82.86 | 83.02 | 86.32 | 86.58 | 87.40 | 87.36 | 86.28 |
| Attack Acc ↑ | GMI | 4.00±8.43 | 18.00±17.51 | 14.00±13.50 | 22.00±17.51 | 10.00±10.54 | 14.00±13.50 | 20.00±9.43 | 20.00±13.33 | **28.00±16.87** | 12.00±10.33 |
| | GMI-HSIC | 16.00±15.78 | 22.00±11.35 | 18.00±22.01 | 18.00±11.35 | 18.00±14.76 | 18.00±17.51 | 20.00±13.33 | 24.00±18.38 | **38.00±14.76** | 22.00±17.51 |
| Attack Acc5 ↑ | GMI | 52.00±16.87 | 58.00±11.35 | 50.00±21.60 | 62.00±14.76 | 54.00±16.47 | 52.00±10.33 | 60.00±13.33 | 62.00±14.76 | **66.00±13.50** | 48.00±10.33 |
| | GMI-HSIC | 58.00±17.51 | 72.00±13.98 | 58.00±14.76 | 64.00±15.78 | 54.00±23.19 | 62.00±19.89 | 62.00±19.89 | 62.00±11.35 | **70.00±10.54** | 60.00±13.33 |
| FID ↓ | GMI | 8.7736 | 8.6881 | 8.3101 | 7.9541 | 7.7629 | **6.2580** | 7.6331 | 7.2277 | 7.5603 | 7.4346 |
| | GMI-HSIC | 8.1356 | 7.6322 | 7.4147 | 7.1645 | **6.1716** | 6.9755 | 7.1390 | 7.0099 | 7.0535 | 8.1704 |

Table 23: FL privacy leakage indicated by Attack Acc/Acc5± standard deviation(%) and FID on CIFAR10 via FedInverse using KED-MI and KED-MI-HSIC with two diverse defense methods: MID and BiDO. Bold values denote the best metric results obtained by KED-MI or KED-MI-HSIC throughout the FL training epoch. The symbol ↓(↑) denotes that smaller (larger) values are favored.

| Metrics | Methods | FL#R01 | FL#R02 | FL#R03 | FL#R04 | FL#R05 | FL#R06 | FL#R07 | FL#R08 | FL#R09 | FL#R10 |
|---|---|---|---|---|---|---|---|---|---|---|---|
| *Defense Method* | | MID | | | | | | | | | |
| Accuracy ↑ | | 66.67 | 75.22 | 79.92 | 53.80 | 82.39 | 84.84 | 84.56 | 86.11 | 87.14 | 79.97 |
| Attack Acc ↑ | KED-MI | 23.31±7.50 | 30.79±10.45 | 35.60±12.75 | 37.73±9.23 | 38.20±10.57 | 41.80±14.41 | 30.81±14.18 | 27.27±13.15 | **45.17±11.57** | 35.45±13.86 |
| | KED-MI-HSIC | 29.09±11.54 | 43.27±9.08 | 40.52±12.70 | 45.88±13.20 | **50.84±14.05** | 48.80±14.29 | 41.72±11.79 | 49.92±11.46 | 49.92±11.46 | 43.76±13.98 |
| Attack Acc5 ↑ | KED-MI | 68.41±12.36 | 82.53±7.21 | 95.51±7.76 | 88.23±3.71 | 85.71±11.69 | 94.15±8.46 | 92.25±8.32 | 88.17±11.72 | 85.91±9.55 | **97.43±5.84** |
| | KED-MI-HSIC | 77.12±13.22 | 96.67±5.37 | 97.15±4.86 | 92.29±9.22 | 98.57±5.07 | 96.89±6.59 | 94.19±9.66 | 87.41±10.81 | 96.63±7.45 | **99.23±3.30** |
| FID ↓ | KED-MI | 10.0550 | 10.1391 | 7.6529 | 7.1021 | 6.1316 | 7.1951 | 5.8090 | 5.8956 | **5.0637** | 8.1110 |
| | KED-MI-HSIC | 8.0944 | 6.9088 | 5.6072 | 5.3786 | 7.4237 | **3.7721** | 4.9038 | 5.6893 | 5.0551 | 5.8501 |
| *Defense Method* | | BiDO | | | | | | | | | |
| Accuracy ↑ | | 71.16 | 31.08 | 83.52 | 82.86 | 83.02 | 86.32 | 86.58 | 87.40 | 87.36 | 86.28 |
| Attack Acc ↑ | KED-MI | 45.60±11.91 | 41.23±6.44 | 21.96±12.87 | 49.53±15.19 | 48.96±13.73 | 27.19±12.09 | 62.35±12.14 | 62.25±14.94 | **63.75±13.92** | 56.03±7.48 |
| | KED-MI-HSIC | 54.23±14.17 | 56.11±7.48 | 30.12±13.91 | 65.33±14.26 | 57.01±14.44 | 36.03±13.22 | 65.72±11.86 | 67.85±11.54 | 70.56±10.05 | **72.25±6.54** |
| Attack Acc5 ↑ | KED-MI | 79.83±1.40 | 79.91±9.18 | 83.40±6.38 | 92.15±4.38 | 83.37±7.55 | 83.15±3.63 | **87.53±3.14** | 79.72±2.21 | 79.69±2.40 | 78.59±4.99 |
| | KED-MI-HSIC | 94.27±7.88 | 94.00±2.32 | 93.03±10.11 | **99.89±1.44** | 90.09±5.76 | 84.45±10.99 | 82.16±6.24 | 85.71±6.38 | 79.97±0.42 | 86.53±0.93 |
| FID ↓ | KED-MI | 6.7435 | 8.1078 | 5.2110 | 8.6607 | 4.8133 | 5.8224 | **2.2666** | 3.7742 | 4.7176 | 5.4706 |
| | KED-MI-HSIC | 6.4173 | 7.1471 | 5.8914 | 3.8114 | 4.0443 | 5.8198 | **3.1937** | 4.0026 | 3.7809 | 3.5632 |

cannot be integrated and utilized. The data privacy problem is because the data owner does not want the data to be shared with a third party, making it difficult for centralized machine learning to utilize all the data. In response to the above problems, FL was proposed McMahan et al. (2017).

FL uploads model parameters to the central server instead of training data, thereby ensuring the data privacy of participants and reducing communication costs. Although FL has made significant benefits to the fields of the Internet of Things Savazzi et al. (2020), network security Chen et al. (2022), and medical care Huang et al. (2019), but it also faces some challenges. First of all, the global model in FL can be poisoned by uploading malicious parameters to the server Zhang et al. (2019). In Zhang et al. (2019), the malicious party leverages the global model as a discriminator to train the GAN, where the generated data can be used to poison the global model. The poisoning attack aims to reduce the performance of the global model rather the leak user privacy. For obtaining user privacy, attackers can recover the local data from the gradients Zhu et al. (2019) when all participants upload the trained model gradients to the central server. However, no studies pay attention to the data leakage problem when the attackers are pretended to be benign users. This paper mainly discusses the vulnerability of federated learning from the perspective of attackers obtaining user privacy as the FL participants.

## B.2 GRADIENT INVERSION ATTACKS

Pasquini et al. (2022) discusses the "gradient suppression attack", a method where a malicious server exploits model inconsistencies to bypass Secure Aggregation and extract a specific user's model update. Central to this attack is the "dead-layer trick," which manipulates ReLU layers by inducing the dying-ReLU phenomenon, where a network's parameter derivatives become zero. This is achieved through malicious parameters that exploit the ReLU function's non-differentiability, allowing the server to nullify the model updates of non-targeted users. Consequently, the server isolates and leaks the targeted user's model update, undermining the integrity of the model training process. Moreover, Boenisch et al. (2023) investigates an attack on Federated Learning (FL) systems that use Distributed Differential Privacy (DDP) and Secure Aggregation (SA). The attack involves circumventing SA by introducing Sybil devices, controlled by the server, into the FL process. These devices manipulate the outcome by returning arbitrary gradients, allowing the server to isolate and extract the target user's gradients. The paper also points out the limitations of DDP, where the noise added for privacy is often insufficient, especially compared to Local Differential Privacy. By exploiting these vulnerabilities, the attacker can reconstruct individual users' training data. The paper discusses using "trap weights" to create redundancy in the data, facilitating higher-fidelity reconstruction. Experiments with datasets like CIFAR10 and IMDB demonstrate the attack's effectiveness, using techniques like similarity clustering and leveraging gradient sparsity to improve the quality of reconstructed data. The study further notes that higher gradient norms lead to more significant data leakage, which the attack exploits. This comprehensive analysis reveals critical weaknesses in FL systems enhanced with DDP and SA, showing that private user data can still be extracted despite these privacy measures. Additionally, Zhao et al. (2023) presents a sophisticated attack in Federated Learning (FL) environments, greatly surpassing previous methods in data leakage efficiency. LOKI can leak 76-86% of data samples in a single training round, targeting both FedAVG and FedSGD systems. The attack, designed for a cross-device FL setting with secure aggregation, manipulates the model architecture and parameters, enabling data recovery from hundreds of clients. A key tactic is using customized convolutional kernels to create separate identity mapping sets, maintaining distinct weight gradients for different clients. LOKI resolves scalability issues in FedAVG by introducing a convolutional scaling factor (CSF), which aids in precision during reconstruction and addresses neuron activation overlap. The attack also cleverly utilizes biases in the FC layer to learn and utilize data characteristics like average pixel intensity. This technique enables the identification of data ownership post-aggregation and requires significantly fewer parameters compared to previous methods, enhancing efficiency. LOKI is effective even in the absence of full model inconsistency among clients, marking a significant advancement in data reconstruction techniques in FL systems.

## B.3 MODEL INVERSION ATTACKS

Unlike gradient inversion attacks, model inversion attacks do not need to reconstruct original images from the feature maps, but gradient inversion attacks need to. According to different attack strategies, privacy attacks on machine learning can be divided into membership inference attacks, model inversion attacks, and parameter extraction attacks. The membership inference attack refers to the attacker trying to determine whether a piece of information exists in the training data set of the global model Shokri et al. (2017). Model parameter extraction means that when the global model parameters are not public, the attacker knows part of the model structure information and attempts

to access the global model to get the parameters Ateniese et al. (2015). Model inversion attack refers to inverting some or all attribute values of a target data in the training set through the model's output Liu et al. (2020). Among them, membership inference attacks and model inversion attacks can be regarded as direct attacks, and model parameter extraction attacks are indirect.

This paper mainly focuses on model inversion attacks. The first model inversion attack was proposed by Fredrikson et al. Fredrikson et al. (2014). They used demographic information as auxiliary information and a linear regression model of drug dosage as the global model to recover patient genomic information. This research demonstrates that even if an attacker only has access privilege to the global model, it is possible to obtain users' sensitive data. Hitaj et al. Hitaj et al. (2017) proposed a model inversion attack in collaborative learning scenarios. In this attack, the attacker is a participant in collaborative learning. By actively participating in the training of the global model, the model parameters are obtained so that the attack under white-box conditions can be realized. The experimental results show that as long as the local model accuracy of the participant is high, a good attack performance can be achieved. Ateniese et al. Ateniese et al. (2015) constructed a new meta-classifier (meta-classifier) and trained it to attack other classifiers to obtain sensitive information about their training data sets.

Due to their excellent data generation ability, generative adversarial networks are widely used in various deep learning tasks. Wang et al. Wang et al. (2019) proposed a model inversion attack for FL. This method designed a multi-task generative confrontation model as the attack model and successfully realized the user-level privacy attack. In addition, some optimization-based methods have been proposed, such as GMI Zhang et al. (2020), KED-MI Chen et al. (2021), and VMI Wang et al. (2021a), which obtain private data in the global model by training GAN. The details of these attacks will be described in the next section.

### B.4 STATISTICAL DEPENDENCY MEASURES

Mutual information Hutter (2001) and Hilbert-Schmidt independency criterion (HSIC) Gretton et al. (2005) are statistic dependency measure metrics that are well established in statistics. Unlike mutual information, the HSIC does not need to estimate two variables' probability density but directly convert it into sampling. Due to its effectiveness and high efficiency, HSIC is widely used in machine learning, such as dimensionality reduction Zhang & Zhou (2010), feature selection Song et al. (2012), transfer learning Wang & Yang (2011), and deep learning Lopez et al. (2018). The central idea of the HSIC-based learning method is to use the HSIC to measure the dependence and achieve the solutions by maximizing or minimizing such associations Wang et al. (2021c).

## C EXTRA EXPERIMENTS ON HEALTH DATASETS AND CIFAR-10

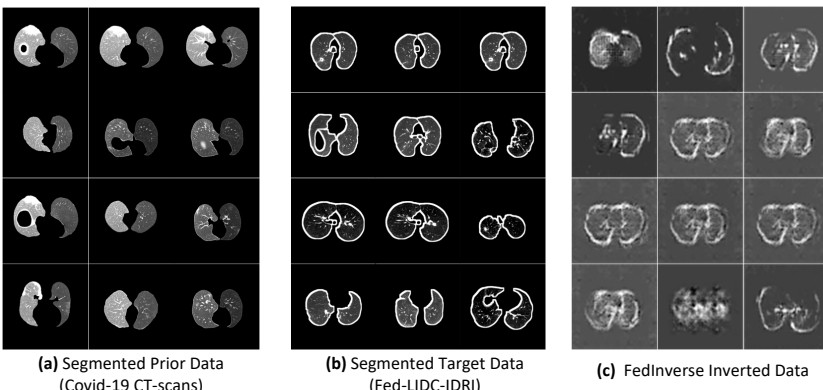

(a) Segmented Prior Data
(Covid-19 CT-scans)

(b) Segmented Target Data
(Fed-LIDC-IDRI)

(c) FedInverse Inverted Data

Figure 6: Visualization results of FedInverse on Fed-LIDC-IDRI Terrail et al. (2022) using the publicly available prior data from Covid-19 CT-Scans Kaggle (2019). It should be noted that, considering the global model in FL is a nodule detection model, FedInverse adopts the preprocessed segmentation results for original CT-scanned images as the private data distributed to federated clients.

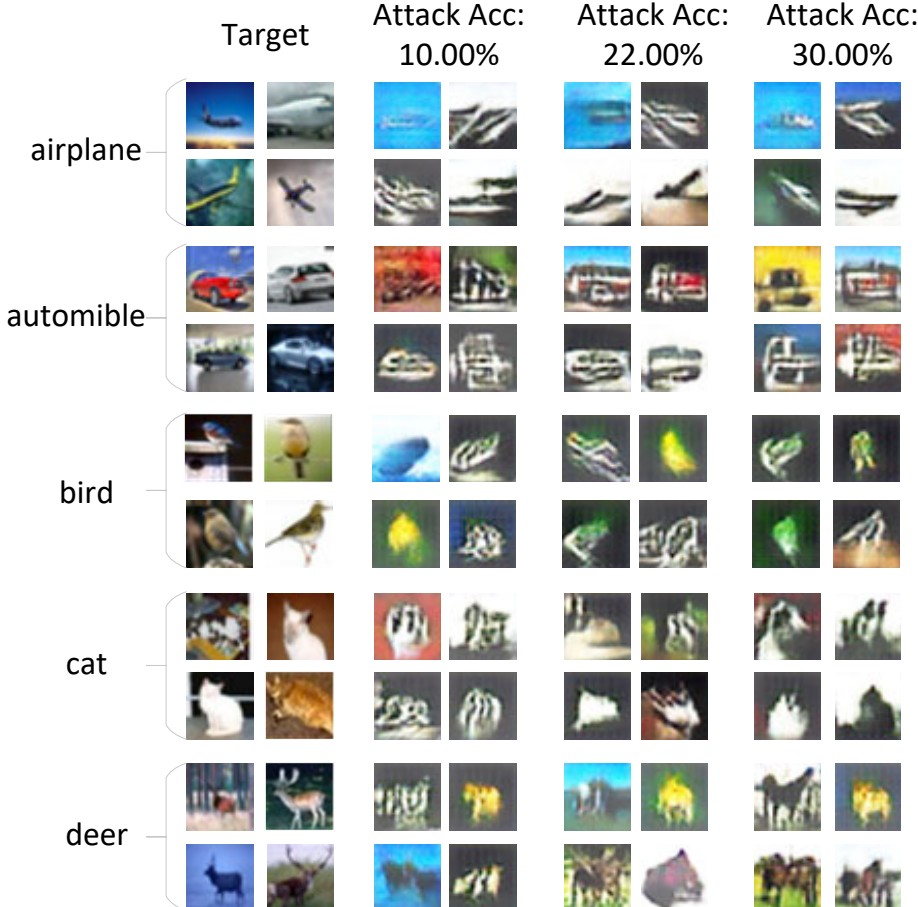

Figure 7: Visualization results of FedInverse on CIFAR-10 across different attack accuracies (10.00%, 22.00%, and 30.00%).

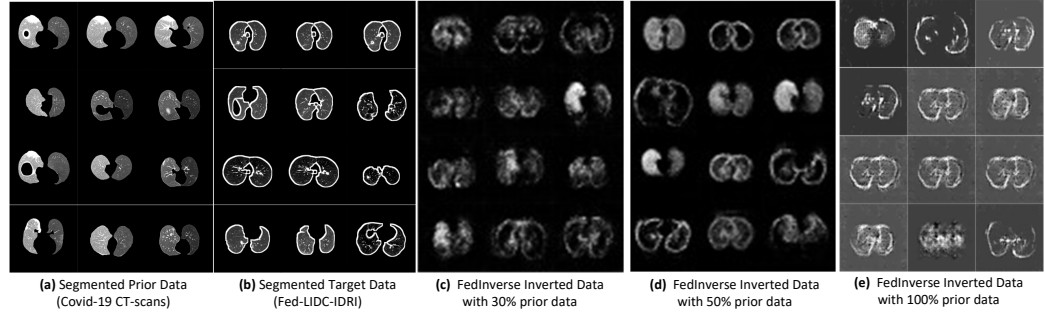

Figure 8: Visualization results of FedInverse on Fed-LIDC-IDRI Terrail et al. (2022) using the publicly available prior data from Covid-19 CT-Scans Kaggle (2019). We select a total of 665 segmented images from Kaggle (2019) as prior data. Panes (c), (d), and (e) present the inverted results with prior data proportions of 30%, 50%, and 100%, respectively.

