# OpenReview forum: "FedInverse: Evaluating Privacy Leakage in Federated Learning"
_ICLR.cc/2024/Conference — ICLR 2024 poster_

### Official Review · Reviewer_5eXT · 2023-10-14

**Soundness:** 3 good
**Presentation:** 2 fair
**Contribution:** 2 fair
**Rating:** 8
**Confidence:** 3

**Summary:**

The authors apply known model inversion attacks (GMI, KED, VMI) to FL where a clients might act malicious and try to learn sensitive training data from other participants (by applying these attacks to the global models distributed in each round). The authors furthermore propose an optimization to these attacks based on introducing more diversity in the images generated by the attacker. The evaluation on various standard image classification tasks demonstrates the effectiveness of the model inversion attacks and a mostly minor advantage for the proposed optimization.

**Strengths:**

The paper studies an important topic, showing further vulnerabilities of FL, which is often assumed to be a privacy-preserving ML paradigm.

In the discussed threat model, this might indeed be the first (or one of the first) works to reconstruct sensitive training data of other clients just from plaintext access to the global model.

The underlying model inversion attacks are clearly presented and perform well in the evaluation.

**Weaknesses:**

The work is based on the bold premise that "no studies pay attention to the data leakage problem when the attackers are pretended to be benign users", i.e., data extraction from the global models that are distributed by the central server in each round. While this might be true, a dedicated discussion of related works is missing from the main body of the paper (and only provided in Appendix B). There definetely exist works that look into data extraction from the global model, but from the perspective of a malicious aggregator that can use techniques like providing inconsistent models. For example: Boenisch et al. (arXiv:2301.04017), Pasquini et al. (CCS'22), Zhao et al. (SP'24).

The application of the existing model inversion attacks seems rather straightforward, assumes that a good auxiliary dataset is available, and the proposed diversity optimization in the evaluation does not show impressive improvements over the respective baselines.

The presentation and positioning of the suggested FedInverse framework is rather unclear to me. On the one hand, it is presented as an attack, on the other hand as an evaluation framework => is it supposed to be something that also FL operators should utilize to measure whether real benign clients might be vulnerable to such attacks?
Algorithm 1 is also unclear in this respect: Do all clients run the attack in parallel (instead of a small subset of malicious clients pretending to be benign)? The set Q_t is not passed to the FedInverse function as a parameter but assumed to be a global variable in line 19; however, the function still explicitly returns it => make clear what is the purpose of this algorithm, what is the final result of the algorithm, and how values are passed around between the functions.

**Update:** The authors have appropriately addressed my concerns in the rebuttal and provided a revised version of the paper. Therefore, I'm upgrading my score.

**Questions:**

- What is the relation to existing attacks for extracting sensitive training data from global models listed above?
- Does the attack perform well / at all if there is no good quality auxiliary dataset available? (e.g., because training is done over records of rare medical diseases, etc.)
- Is the proposed algorithm / framework meant purely for attack purposes or also as a defence / pre-check for the server to evaluate the vulnerability of participating clients?

---

> ### Author Response · Authors · 2023-11-21
> **Rebuttal for W1 (1/2)**
>
> > The work is based on the bold premise that "no studies pay attention to the data leakage problem when the attackers are pretended to be benign users", i.e., data extraction from the global models that are distributed by the central server in each round. While this might be true, a dedicated discussion of related works is missing from the main body of the paper (and only provided in Appendix B). There definetely exist works that look into data extraction from the global model, but from the perspective of a malicious aggregator that can use techniques like providing inconsistent models. For example: Boenisch et al. (arXiv:2301.04017), Pasquini et al. (CCS'22), Zhao et al. (SP'24).
>
> #### **Thank the reviewer's comments and suggestions. We will answer the weakness and question one by one in this rebuttal. We will include rebuttal discussions in the updated version to make the final version more clear to our readers. We hope the answers below solve your concerns. If your concerns were not cleared, please let us know, and we'll try to clarify them further. We also hope you can re-evaluate our paper based on the answers and consider raising your score.**
>
> #### For W1, We will add a related works summary into the main context and move some of the experiment results into the Appendix. We have further studied the suggested papers as follows and will add them to the related works.
> #### The paper ```[13]``` discusses the "gradient suppression attack", a method where a malicious server exploits model inconsistencies to bypass Secure Aggregation and extract a specific user's model update. Central to this attack is the "dead-layer trick," which manipulates ReLU layers by inducing the dying-ReLU phenomenon, where a network's parameter derivatives become zero. This is achieved through malicious parameters that exploit the ReLU function's non-differentiability, allowing the server to nullify the model updates of non-targeted users. Consequently, the server isolates and leaks the targeted user's model update, undermining the integrity of the model training process.
> #### The paper ```[14]``` investigates an attack on Federated Learning (FL) systems that use Distributed Differential Privacy (DDP) and Secure Aggregation (SA). The attack involves circumventing SA by introducing Sybil devices, controlled by the server, into the FL process. These devices manipulate the outcome by returning arbitrary gradients, allowing the server to isolate and extract the target user's gradients. The paper also points out the limitations of DDP, where the noise added for privacy is often insufficient, especially compared to Local Differential Privacy. By exploiting these vulnerabilities, the attacker can reconstruct individual users' training data. The paper discusses using "trap weights" to create redundancy in the data, facilitating higher-fidelity reconstruction. Experiments with datasets like CIFAR10 and IMDB demonstrate the attack's effectiveness, using techniques like similarity clustering and leveraging gradient sparsity to improve the quality of reconstructed data. The study further notes that higher gradient norms lead to more significant data leakage, which the attack exploits. This comprehensive analysis reveals critical weaknesses in FL systems enhanced with DDP and SA, showing that private user data can still be extracted despite these privacy measures.
> #### The paper ```[15]``` presents a sophisticated attack in Federated Learning (FL) environments, greatly surpassing previous methods in data leakage efficiency. LOKI can leak 76-86% of data samples in a single training round, targeting both FedAVG and FedSGD systems. The attack, designed for a cross-device FL setting with secure aggregation, manipulates the model architecture and parameters, enabling data recovery from hundreds of clients. A key tactic is using customized convolutional kernels to create separate identity mapping sets, maintaining distinct weight gradients for different clients. LOKI resolves scalability issues in FedAVG by introducing a convolutional scaling factor (CSF), which aids in precision during reconstruction and addresses neuron activation overlap. The attack also cleverly utilizes biases in the FC layer to learn and utilize data characteristics like average pixel intensity. This technique enables the identification of data ownership post-aggregation and requires significantly fewer parameters compared to previous methods, enhancing efficiency. LOKI is effective even in the absence of full model inconsistency among clients, marking a significant advancement in data reconstruction techniques in FL systems.

---

> ### Author Response · Authors · 2023-11-21
> **Rebuttal for W1 (2/2)**
>
> #### The major differences between our paper and the three papers listed above are listed below:
> #### - Model inversion attacks do not need to reconstruct original images from the **feature maps**, but gradients attack need. Therefore, gradient inversion attacks cannot be deployed on the clients because clients cannot obtain the gradient changes from the other clients.
> #### - Based on the attack mechanism above, model inversion attacks can be Black-box settings where the attacker can launch the attack on the global models it received in FL. However, the gradient inversion attacks are White-box settings in which the gradient changes must be obtained by the attackers.
> #### Therefore, the attacks ```[13]``` ```[14]``` ```[15]``` cannot be applied in our FL settings because our FL settings aim to evaluate privacy issues from the client's perspective. We will add the review into the main context of related works.
>
> #### ```[13]``` Pasquini, D., Francati, D. and Ateniese, G., 2022, November. Eluding secure aggregation in federated learning via model inconsistency. In Proceedings of the 2022 ACM SIGSAC Conference on Computer and Communications Security (pp. 2429-2443).
>
> #### ```[14]``` Boenisch, F., Dziedzic, A., Schuster, R., Shamsabadi, A.S., Shumailov, I. and Papernot, N., 2023, July. Reconstructing Individual Data Points in Federated Learning Hardened with Differential Privacy and Secure Aggregation. In 2023 IEEE 8th European Symposium on Security and Privacy (EuroS\&P) (pp. 241-257). IEEE.
>
> #### ```[15]``` Zhao, J.C., Sharma, A., Elkordy, A.R., Ezzeldin, Y.H., Avestimehr, S. and Bagchi, S., 2023, October. LOKI: Large-scale Data Reconstruction Attack against Federated Learning through Model Manipulation. In 2024 IEEE Symposium on Security and Privacy (SP) (pp. 30-30). IEEE Computer Society.

---

> ### Author Response · Authors · 2023-11-21
> **Rebuttal for W2**
>
> > The application of the existing model inversion attacks seems rather straightforward, assumes that a good auxiliary dataset is available, and the proposed diversity optimization in the evaluation does not show impressive improvements over the respective baselines.
>
> #### The lack of publicly available data is the current challenge for most model-inversion attackers and limits the usefulness of not only our proposed method but also most of the other public-data-assisted generative models, including GMI, KED-MI, and VMI. For evaluation purposes, FedInverse uses very limited similar but different distributed public data to launch the model inversion attacks in our FL settings. This is more realistic in real-world applications because there are still available public data that have similar but not the same distribution to the targeted data from the global model. For example, the experiments in **Appendix A.2** have been conducted using FMNIST and EMNIST as the public datasets to generate the data from MNIST.
>
> #### The experiment of diversity optimization is important to help us evaluate the vulnerability of FL under the model inversion attack via the diversity of the generated data. The results show that diversity optimization can achieve higher improvements in attack, as well as results that are at least not inferior to those without optimization. For example, some improvements show in the **Appendix A** in **Table 3** from **52.00** to **58.00** when we use FMNIST as the public dataset, **Table 4** from **86.83** to **99.80** when we use FMNIST as the public dataset, **Table 7** from **52.00** to **60.00** when (E,B) = (2, 60), **Table 8** from **78.89** to **85.09** when (E,B) = (2, 10), and **Table 17** from **60.00** to **82.80** when (E,B) = (30, 128) etc.

---

> ### Author Response · Authors · 2023-11-21
> **Rebuttal for W3**
>
> > The presentation and positioning of the suggested FedInverse framework is rather unclear to me. On the one hand, it is presented as an attack, on the other hand as an evaluation framework => is it supposed to be something that also FL operators should utilize to measure whether real benign clients might be vulnerable to such attacks? Algorithm 1 is also unclear in this respect: Do all clients run the attack in parallel (instead of a small subset of malicious clients pretending to be benign)? The set Q_t is not passed to the FedInverse function as a parameter but assumed to be a global variable in line 19; however, the function still explicitly returns it => make clear what is the purpose of this algorithm, what is the final result of the algorithm, and how values are passed around between the functions.
>
> #### Our work aims to evaluate the boundaries of privacy protection in the whole FL system from the client's perspective, whether the client can obtain data from other clients. From the reviewer's suggestions and comments, we can evaluate that the FL task can be considered secure, with no public datasets attackers can leverage. Therefore, evaluating privacy boundaries under model inversion attacks in FL is important for the FL research community.
>
> #### Only a single attacker participates in the FL to pretend to be a benign client in each communication round. We have made the changes to emphasize only one single attacker will join the FL in each communication round in **Algorithm 1**. In line 19, the set Q_t has been replaced by N, representing a Gaussian distribution. Please check the updated algorithm in **Algorithm 1** in the **Rebuttal Revision**. The final results of the **Algorithm 1** will be relevant evaluation results on Q_t across all communication rounds.

---

> ### Author Response · Authors · 2023-11-21
> **Rebuttal for Questions**
>
> > - What is the relation to existing attacks for extracting sensitive training data from global models listed above?
> > - Does the attack perform well / at all if there is no good quality auxiliary dataset available? (e.g., because training is done over records of rare medical diseases, etc.)
> > - Is the proposed algorithm / framework meant purely for attack purposes or also as a defence / pre-check for the server to evaluate the vulnerability of participating clients?
>
> - ####  As we answer the question 1 in **Rebuttal for W1 (2/2)**. We copy and paste the answers here again.
>   #### The major differences between our paper and the three papers listed above are listed below:
>   #### - Model inversion attacks do not need to reconstruct original images from the **feature maps**, but gradients attack need. Therefore, gradient inversion attacks cannot be deployed on the clients because clients cannot obtain the gradient changes from the other clients.
>   #### - Based on the attack mechanism above, model inversion attacks can be Black-box settings where the attacker can launch the attack on the global models it received in FL. However, the gradient inversion attacks are White-box settings in which the gradient changes must be obtained by the attackers.
>
>   #### Therefore, the attacks ```[13]``` ```[14]``` ```[15]``` cannot be applied in our FL settings because our FL settings aim to evaluate privacy issues from the client's perspective. We will add the review into the main context of related works.
>
>   #### ```[13]``` Pasquini, D., Francati, D. and Ateniese, G., 2022, November. Eluding secure aggregation in federated learning via model inconsistency. In Proceedings of the 2022 ACM SIGSAC Conference on Computer and Communications Security (pp. 2429-2443).
>
>   #### ```[14]``` Boenisch, F., Dziedzic, A., Schuster, R., Shamsabadi, A.S., Shumailov, I. and Papernot, N., 2023, July. Reconstructing Individual Data Points in Federated Learning Hardened with Differential Privacy and Secure Aggregation. In 2023 IEEE 8th European Symposium on Security and Privacy (EuroS\&P) (pp. 241-257). IEEE.
>
>   #### ```[15]``` Zhao, J.C., Sharma, A., Elkordy, A.R., Ezzeldin, Y.H., Avestimehr, S. and Bagchi, S., 2023, October. LOKI: Large-scale Data Reconstruction Attack against Federated Learning through Model Manipulation. In 2024 IEEE Symposium on Security and Privacy (SP) (pp. 30-30). IEEE Computer Society.
>
> - #### The lack of publicly available data is the **current challenge** for most model-inversion attackers and limits the usefulness of not only our proposed method **but also most of the other public-data-assisted generative models**, including GMI, KED-MI, and VMI. For evaluation purposes, **FedInverse uses very limited similar but different distributed public data** to launch the model inversion attacks in our FL settings. This is **more realistic** in real-world applications because there are still available public data that have similar but not the same distribution to the targeted data from the global model. For example, the experiments in **Appendix A.2** have been conducted using FMNIST and EMNIST as the public datasets to generate the data from MNIST. We have done further experiments on the medical dataset suggested by the Reviewer ```hL64``` and have added to **Appendix C Figure 6 and Figure 8**. From the evaluation point of view, the FL will be robust to model inversion attacks without good-quality auxiliary datasets available. However, it is possible that attackers can obtain the public dataset with very limited similar but different distributions. Therefore, the evaluation of the boundaries of privacy protection in FL is valuable for the research community.
>
> - #### This paper aims to **evaluate the vulnerability of the FL** where clients can obtain data through model inversion attacks from other clients without obtaining extra information. This evaluation work can help **future pre-checks from the client side or the whole FL system pre-checks via the trusted third party** because FedInverse attacks FL from the client, and clients do not know each other. Therefore, our evaluation work is important to enhance the future FL privacy.

---

> > ### Comment · Reviewer_5eXT · 2023-11-22
> > **Thanks for the rebuttal**
> >
> > Thanks for the extensive rebuttal and updating the submission PDF! All my concerns were appropriately addressed and I'm willing to increase my score during the PC discussion. Re W3, as reviewer Do36 had somewhat similar misunderstandings regarding the entire setup and purpose of the work, an extensive revision of the introduction section might be helpful.

---

> ### Author Response · Authors · 2023-11-22
>
> #### Thank you for giving valuable comments and discussions. The discussions you have contributed to this paper are constructive. We have followed your suggestions and updated the submitted rebuttal revision PDF in the introduction section to clarify the entire setup and purpose of the work.
>
> #### Please find the changes in Introduction paragraph 4: "Given that the data-leakage risk cannot be eliminated, we propose FedInverse, a novel privacy leakage evaluation method to evaluate the boundaries of privacy protection in the FL system from the participant’s perspective, whether one participant can obtain data from other participants. An attacker pretending to be a participant in FedInverse can conduct the Black-box attacks via global model prediction in each federated training round.", which emphasizes the purpose and setup of our paper.

---

### Official Review · Reviewer_Do36 · 2023-10-30

**Soundness:** 3 good
**Presentation:** 3 good
**Contribution:** 2 fair
**Rating:** 6
**Confidence:** 4

**Summary:**

This manuscript proposes FedInverse, an approach that one attacker could use to obtain the private local data of other clients as a normal participant. FedInverse generates the possible training data from observing the aggregated global model with the help of the GAN. Additionally, a regularizer using the Hilbert-Schmidt independence criterion (HSIC) is applied to generate more diverse images. The experiments show that FedInverse with HSIC has achieved progress in recovering the local data of other participants.

The rating has been changed to 6 after the rebuttle discussion.

**Strengths:**

This paper focuses on how to steal the private data of other clients in FL as a participant, and the idea of applying HSIC as a regularizer to increase diversity is helpful. Besides, for the evaluation part, plenty of factors are taken into consideration, which make the results comprehensive and easy to understand.

**Weaknesses:**

- FedInverse uses typical model inversion attacks to generate the images. However, whether those approaches could be directly applied to the aggregated model is still a problem. In the setting of the experiments, the clients have local datasets that are disjoint on the label (for example, in MNIST, the attacker has labels 5-9, while other clients have labels 0-4). However, the setting is not practical for the typical situation. As the number of participants increases, some clients will have similar data distributions, so that FedInverse may fail in this setting.
- Moreover, FedInverse generates the image from a similar distribution with target clients. However, it could not tell the source of the image exactly (from an attacker or another specific participant).
- Some metrics used in the experiments should be clear. Take the attack accuracy as an example. A well-trained model may still classify an image with some flaw as the correct class, and it could lead to results at variance with reality. Additionally, the top-5 accuracy is too weak for some experiments, especially on MNIST (with only ten labels). As for FID, the attack results of some baseline approaches(for example, applying the MI attack on local models) should be added for comparison. Otherwise, it would be hard for readers to understand the degree of privacy leakage.
- In Section 3, the description of the attack procedure is not clear enough. More details are needed.

**Questions:**

- How does FedInverse perform in a typical vertical FL setting? For example, 10+ clients with the same (or similar) data distribution. More discussions of this case should be given.
- Is there any method to remove or mitigate the effect of the attacker dataset in the final results? From Figure 4, the generator is still trying to produce an image similar to its local dataset, even with the regulation of HSIC.

---

> ### Author Response · Authors · 2023-11-21
> **Clarification for Misunderstanding of FedInverse**
>
> #### **Thanks for taking the time to review and give the comments. However, we found that the reviewer completely misunderstood the basic idea and FL settings of our proposed FedInverse evaluation rather than other Reviewers ```hL64```, ```E5kp```, and ```5eXT```. FedInverse aims to evaluate the privacy protection boundary when an attacker pretends to be a client and reconstructs data from other clients. Therefore, we will explain our idea and FL settings in the first place and try to clarify the reviewer's concerns from the weaknesses and questions. We hope the answers below solve your concerns. If your concerns were not cleared, please let us know, and we’ll try to clarify them further. We also hope you can re-evaluate our paper based on the answers and consider raising your score.**
>
> #### It is important to clarify the basic idea and FL settings of our paper. The FedInverse is to evaluate how the model inversion attacks affect FL from the client's perspective. Unlike the gradient inversion attacks suggested by the Reviewer ```5eXT``` ```[1]``` ```[2]``` ```[3]``` published recently, which assume that a malicious server can access and manipulate model layers and the gradients to attack the target clients. Our paper is the first work to evaluate the boundary of privacy protection against model inversion attack from the client perspective in FL, which is recognized by Reviewer ```hL64```, ```E5kp```, and ```5eXT```. Specifically, an attacker can join the FL and attack the aggregated global model distributed from the central server to reconstruct the data from all the other clients without a specific target. The key differences between our paper and recent gradient inversion attacks work are elaborated below.
>
> - #### FedInverse does not need to reconstruct original images from the **feature maps**, but gradients attack need. Therefore, gradient inversion attacks cannot be deployed on the clients because clients cannot obtain the gradient changes from the other clients.
> - #### Based on the attack mechanism above, FedInverse can be Black-box settings where the attacker can launch the attack on the global models it received in FL. However, the gradient inversion attacks are White-box settings in which the gradient changes must be obtained by the attackers.
>
> #### ```[1]``` Pasquini, D., Francati, D. and Ateniese, G., 2022, November. Eluding secure aggregation in federated learning via model inconsistency. In Proceedings of the 2022 ACM SIGSAC Conference on Computer and Communications Security (pp. 2429-2443).
>
> #### ```[2]``` Boenisch, F., Dziedzic, A., Schuster, R., Shamsabadi, A.S., Shumailov, I. and Papernot, N., 2023, July. Reconstructing Individual Data Points in Federated Learning Hardened with Differential Privacy and Secure Aggregation. In 2023 IEEE 8th European Symposium on Security and Privacy (EuroS\&P) (pp. 241-257). IEEE.
>
> #### ```[3]``` Zhao, J.C., Sharma, A., Elkordy, A.R., Ezzeldin, Y.H., Avestimehr, S. and Bagchi, S., 2023, October. LOKI: Large-scale Data Reconstruction Attack against Federated Learning through Model Manipulation. In 2024 IEEE Symposium on Security and Privacy (SP) (pp. 30-30). IEEE Computer Society.

---

> ### Author Response · Authors · 2023-11-21
> **Rebuttal for W1**
>
> > - FedInverse uses typical model inversion attacks to generate the images. However, whether those approaches could be directly applied to the aggregated model is still a problem. In the setting of the experiments, the clients have local datasets that are disjoint on the label (for example, in MNIST, the attacker has labels 5-9, while other clients have labels 0-4). However, the setting is not practical for the typical situation. As the number of participants increases, some clients will have similar data distributions, so that FedInverse may fail in this setting.
>
> #### FedInverse indeed generates data from the aggregated global model, where the client receives the aggregated global model from the central server and launches the attack on the global model just like the reviewer summarized in the Summary. In FedInverse, we evaluate the performance of MI attacks in FL settings to reveal data with totally different labels owned by the attackers from the aggregated global model.
>
> #### In addition, it should be noted that our experimental data settings follow the same setting in ```[1]``` ```[2]``` ```[3]``` because this setting is widely known as more challenging than the reviewer-suggested setting of revealing information about clients who have overlapped labels. We provide the detailed work process of FedInverse in **Subsection 3.1, Algorithm 1, and Figure 2**, and **Section 4** elaborates on our experimental settings. Therefore, FedInverse will not fail in the reviewers-suggested setting.
>
> #### ```[1]``` Yuheng Zhang, Ruoxi Jia, Hengzhi Pei, Wenxiao Wang, Bo Li, and Dawn Song. The secret revealer: Generative model-inversion attacks against deep neural networks. In Proceedings of the IEEE/CVF conference on computer vision and pattern recognition, pp. 253–261, 2020.
>
> #### ```[2]``` Si Chen, Mostafa Kahla, Ruoxi Jia, and Guo-Jun Qi. Knowledge-enriched distributional model inversion attacks. In Proceedings of the IEEE/CVF international conference on computer vision, pp. 16178–16187, 2021.
>
> #### ```[3]``` Xiong Peng, Feng Liu, Jingfeng Zhang, Long Lan, Junjie Ye, Tongliang Liu, and Bo Han. Bilateral dependency optimization: Defending against model-inversion attacks. In Proceedings of the 28th ACM SIGKDD Conference on Knowledge Discovery and Data Mining, pp. 1358–1367, 2022.

---

> ### Author Response · Authors · 2023-11-21
> **Rebuttal for W2**
>
> > - Moreover, FedInverse generates the image from a similar distribution with target clients. However, it could not tell the source of the image exactly (from an attacker or another specific participant).
>
> #### FedInverse is designed to generate images from a distinct distribution unrelated to potential attackers, not from a similar distribution. In **Appendix A.2**, we even test how FedInverse can generate MNIST data from the FL aggregated model using a different dataset with entirely different labels, such as FMNIST ```[1]``` and EMINST ```[2]```. Additionally, FedInverse's main contribution lies in evaluating privacy leakage in FL, regardless of whether it originates from attackers or other participants. This contribution has been recognized and supported by all other reviewers.
>
> #### ```[1]``` Han Xiao, Kashif Rasul, and Roland Vollgraf. Fashion-mnist: a novel image dataset for benchmarking machine learning algorithms. arXiv preprint arXiv:1708.07747, 2017.
>
> #### ```[2]``` Gregory Cohen, Saeed Afshar, Jonathan Tapson, and Andre Van Schaik. Emnist: Extending mnist to handwritten letters. In 2017 international joint conference on neural networks (IJCNN), pp.2921–2926. IEEE, 2017.

---

> ### Author Response · Authors · 2023-11-21
> **Rebuttal for W3**
>
> > - Some metrics used in the experiments should be clear. Take the attack accuracy as an example. A well-trained model may still classify an image with some flaw as the correct class, and it could lead to results at variance with reality. Additionally, the top-5 accuracy is too weak for some experiments, especially on MNIST (with only ten labels). As for FID, the attack results of some baseline approaches(for example, applying the MI attack on local models) should be added for comparison. Otherwise, it would be hard for readers to understand the degree of privacy leakage.
>
> #### We have compared to the SOTA baseline ```[1]``` ```[2]``` ```[3]``` ```[4]``` ```[5]``` and followed the attack accuracy used by the prior works. We didn't find more recent works on the problem mentioned by the reviewer. The metrics, such as top-5 accuracy and FID, as well as the datasets we used, such as CelebA ```[6]```, CIFAR-10 ```[7]```, and MNIST ```[8]```, have been used by almost all evaluations for MI ```[1]``` ```[2]``` ```[5]```. This is because CelebA is a very large dataset that contains 10,177 identities, 202,599 face images, and 5 landmark locations, 40 binary attribute annotations per image, where top-5 accuracy is an important metric for evaluation on this large dataset. The datasets of CIFAR-10 and MNIST provide the additional evaluation of small datasets. For the FID value, it is calculated to evaluate the similarity between generated images and real images, while the real images come from the original dataset not owned by attackers for experimental evaluation only. The metrics are very important to evaluate and compare the MI attack performance when the SOTA defense mechanism ```[5]``` has been deployed in the FL settings.
>
> #### ```[1]``` Yuheng Zhang, Ruoxi Jia, Hengzhi Pei, Wenxiao Wang, Bo Li, and Dawn Song. The secret revealer: Generative model-inversion attacks against deep neural networks. In Proceedings of the IEEE/CVF conference on computer vision and pattern recognition, pp. 253–261, 2020.
>
> #### ```[2]``` Si Chen, Mostafa Kahla, Ruoxi Jia, and Guo-Jun Qi. Knowledge-enriched distributional model inversion attacks. In Proceedings of the IEEE/CVF international conference on computer vision, pp. 16178–16187, 2021.
>
> #### ```[3]``` Kuan-ChiehWang, Yan Fu, Ke Li, Ashish Khisti, Richard Zemel, and Alireza Makhzani. Variational model inversion attacks. Advances in Neural Information Processing Systems, 34:9706–9719, 2021.
>
> #### ```[4]``` Tianhao Wang, Yuheng Zhang, and Ruoxi Jia. Improving robustness to model inversion attacks via mutual information regularization. In Proceedings of the AAAI Conference on Artificial Intelligence, volume 35, pp. 11666–11673, 2021.
>
> #### ```[5]``` Xiong Peng, Feng Liu, Jingfeng Zhang, Long Lan, Junjie Ye, Tongliang Liu, and Bo Han. Bilateral dependency optimization: Defending against model-inversion attacks. In Proceedings of the 28th ACM SIGKDD Conference on Knowledge Discovery and Data Mining, pp. 1358–1367, 2022.
>
> #### ```[6]``` Ziwei Liu, Ping Luo, Xiaogang Wang, and Xiaoou Tang. Deep learning face attributes in the wild. In Proceedings of the IEEE international conference on computer vision, pp. 3730–3738, 2015.
>
> #### ```[7]``` Alex Krizhevsky, Geoffrey Hinton, et al. Learning multiple layers of features from tiny images. 2009.
>
> #### ```[8]``` Yann LeCun, L´eon Bottou, Yoshua Bengio, and Patrick Haffner. Gradient-based learning applied to document recognition. Proceedings of the IEEE, 86(11):2278–2324, 1998.

---

> ### Author Response · Authors · 2023-11-21
> **Rebuttal for W4**
>
> > In Section 3, the description of the attack procedure is not clear enough. More details are needed.
>
> #### There are no concerns from other reviewers on the current description of our attack procedure. Could you please specify which details should be further added? We will explain the details based on your detailed questions.

---

> ### Author Response · Authors · 2023-11-21
> **Rebuttal for Questions**
>
> > - How does FedInverse perform in a typical vertical FL setting? For example, 10+ clients with the same (or similar) data distribution. More discussions of this case should be given.
> > - Is there any method to remove or mitigate the effect of the attacker dataset in the final results? From Figure 4, the generator is still trying to produce an image similar to its local dataset, even with the regulation of HSIC.
>
> - #### Our work follows the recent SOTA works with the same FL setting, such as ```[1]``` ```[2]``` ```[3]``` ```[4]``` ```[5]``` ```[6]``` ```[7]``` ```[8]``` ```[9]``` ```[10]``` etc., which is a significant contribution to the research area and recognized by other reviewers. Vertical FL is a completely different FL framework. It is a good suggestion to consider it in our future work.
>
>   #### ```[1]``` Ezzeldin, Y.H., Yan, S., He, C., Ferrara, E. and Avestimehr, A.S., 2023, June. Fairfed: Enabling group fairness in federated learning. In Proceedings of the AAAI Conference on Artificial Intelligence (Vol. 37, No. 6, pp. 7494-7502).
>
>   #### ```[2]``` Lee, R., Kim, M., Li, D., Qiu, X., Hospedales, T., Huszár, F. and Lane, N.D., 2023, November. FedL2P: Federated Learning to Personalize. In Thirty-seventh Conference on Neural Information Processing Systems.
>
>   #### ```[3]``` Fan, Z., Zhang, R., Yao, J., Han, B., Zhang, Y. and Wang, Y., 2023, November. Federated Learning with Bilateral Curation for Partially Class-Disjoint Data. In Thirty-seventh Conference on Neural Information Processing Systems.
>
>   #### ```[4]``` An, X., Shen, L., Hu, H. and Luo, Y., 2023, November. Federated Learning with Manifold Regularization and Normalized Update Reaggregation. In Thirty-seventh Conference on Neural Information Processing Systems.
>
>   #### ```[5]``` Han, S., Park, S., Wu, F., Kim, S., Zhu, B., Xie, X. and Cha, M., 2023. Towards Attack-tolerant Federated Learning via Critical Parameter Analysis. In Proceedings of the IEEE/CVF International Conference on Computer Vision (pp. 4999-5008).
>
>   #### ```[6]``` Chen, R., Wan, Q., Prakash, P., Zhang, L., Yuan, X., Gong, Y., Fu, X. and Pan, M., 2023. Workie-Talkie: Accelerating Federated Learning by Overlapping Computing and Communications via Contrastive Regularization. In Proceedings of the IEEE/CVF International Conference on Computer Vision (pp. 16999-17009).
>
>   #### ```[7]``` Zhang, C., Xiaoman, Z., Sotthiwat, E., Xu, Y., Liu, P., Zhen, L. and Liu, Y., 2023. Generative Gradient Inversion via Over-Parameterized Networks in Federated Learning. In Proceedings of the IEEE/CVF International Conference on Computer Vision (pp. 5126-5135).
>
>   #### ```[8]``` Yang, C., Zhu, M., Liu, Y. and Yuan, Y., 2023. FedPD: Federated Open Set Recognition with Parameter Disentanglement. In Proceedings of the IEEE/CVF International Conference on Computer Vision (pp. 4882-4891).
>
>   #### ```[9]``` Feng, C.M., Yu, K., Liu, N., Xu, X., Khan, S. and Zuo, W., 2023. Towards Instance-adaptive Inference for Federated Learning. In Proceedings of the IEEE/CVF International Conference on Computer Vision (pp. 23287-23296).
>
>   #### ```[10]``` Kim, H., Kwak, Y., Jung, M., Shin, J., Kim, Y. and Kim, C., 2023. ProtoFL: Unsupervised Federated Learning via Prototypical Distillation. In Proceedings of the IEEE/CVF International Conference on Computer Vision (pp. 6470-6479).
>
> - #### Firstly, it is important to emphasize that the attacker dataset used in our experiments is not derived from the local datasets of other participants. In other words, the attacker does not have access to or use datasets belonging to other participants for generating images. Each participant's dataset remains private and is not utilized by the attacker.
>   #### Secondly, the introduction of HSIC into our approach serves to enhance the attack, not to defend against it, which has been demonstrated in **Figure 4** clearly. As you have mentioned in **Figure 4**, the generator of the attacker produces images similar to the local private datasets of participants that do not belong to the attacker, which clearly validates our expectation that HSIC enhances the attack.

---

> ### Comment · Reviewer_Do36 · 2023-11-22
>
> Thanks for the reply. I make sense of the explanation of the evaluation setting. However, some content of the rebuttal still needs clarification. As the rebuttal said, FedInverse is an approach to evaluate how the model inversion attacks affect FL from the client's perspective, and the criterion is how much information about the client's private data the attack could infer from the aggregated model. My understanding is that the more information the attack gets, the more privacy of clients is leaked in FL. If so, what is the difference between FedInverse without HSIC and the mentioned MI attack, which changes the attack's identity from the server to the client? Moreover, where do their targets differ? According to Algorithm 1, All FedInverse without HSIC has to generate the image by using the pre-trained GAN provided by the previous MI attack approaches. Does that mean you could use any GAN-based MI attack approaches by changing the attacker's identity? A more convincing explanation could be presented for the revision.

---

> > ### Author Response · Authors · 2023-11-22
> >
> > #### Thanks for your prompt reply and patient review for our further clarification. To further clarify the concerns of the reviewer, we will elaborate on answers regarding the concerns below.
> >
> > #### The attack model of FedInverse is totally different from the listed three papers ```[1]``` ```[2]``` ```[3]``` (Gradient inversion attacks) in the rebuttal **Clarification for Misunderstanding of FedInverse** (see https://openreview.net/forum?id=nTNgkEIfeb&noteId=tvKozlSJxt). The reason we compare differences between FedInverse and the listed three papers is to help the reviewer clarify our contribution. We are the first paper to evaluate the boundary of privacy protection against model inversion from the client perspective in FL, which is totally different from the recent gradient inversion attack work from the server perspective in FL. The detailed major difference we have listed in the rebuttal **Clarification for Misunderstanding of FedInverse** (see https://openreview.net/forum?id=nTNgkEIfeb&noteId=tvKozlSJxt).
> >
> > #### In addition, attackers as clients in FedInverse can conduct the Black-box attacks (model inversion attacks) via model prediction because the attackers have limited access to the information in FL, especially the information from other clients. Gradient inversion attacks listed in the three papers ```[1]``` ```[2]``` ```[3]``` can only do the White-box attack from the server perspective because the attacker needs to obtain the **feature map of the samples** to launch the gradient inversion attack.
> >
> > #### HSIC is the optimization method for model inversion attack purposes, which is used to improve the diversity of the generated data from the GAN. We evaluate the impact on the effectiveness of the privacy leakage from the diversity of the generated data in FL. Moreover, the HSIC optimization is applicable to existing model inversion attacks (not for the gradient inversion attacks as in the listed papers), and embedded into any GAN-based model inversion attacks.
> >
> > #### In conclusion, the attack will not change identity in FedInverse, acting as the client as is always. Also, the introduction of HSIC does not change the attacker's identity. It is worth noting that FedInverse without HSIC and with HSIC are all deployed on the client side. The novelty and contribution of our paper are recognized by Reviewer ```hL64```, ```E5kp```, and ```5eXT``` as the first work to evaluate the boundary of privacy protection against model inversion attacks from the client perspective in FL.
> >
> > #### We would much appreciate it if you could re-evaluate our paper based on the answers above.
> >
> > #### ```[1]``` Pasquini, D., Francati, D. and Ateniese, G., 2022, November. Eluding secure aggregation in federated learning via model inconsistency. In Proceedings of the 2022 ACM SIGSAC Conference on Computer and Communications Security (pp. 2429-2443).
> > #### ```[2]``` Boenisch, F., Dziedzic, A., Schuster, R., Shamsabadi, A.S., Shumailov, I. and Papernot, N., 2023, July. Reconstructing Individual Data Points in Federated Learning Hardened with Differential Privacy and Secure Aggregation. In 2023 IEEE 8th European Symposium on Security and Privacy (EuroS&P) (pp. 241-257). IEEE.
> > #### ```[3]``` Zhao, J.C., Sharma, A., Elkordy, A.R., Ezzeldin, Y.H., Avestimehr, S. and Bagchi, S., 2023, October. LOKI: Large-scale Data Reconstruction Attack against Federated Learning through Model Manipulation. In 2024 IEEE Symposium on Security and Privacy (SP) (pp. 30-30). IEEE Computer Society.

---

> ### Comment · Reviewer_Do36 · 2023-11-23
>
> Thanks to the authors. The discussion is helpful and has addressed most of my concerns. I am happy to change my rating. Please consider revising the paper regarding the reviewers' comments.

---

> > ### Author Response · Authors · 2023-11-23
> >
> > #### Thank the reviewer for the valuable comments and discussion.
> > - #### We have updated all the content regarding the discussion with different reviewers in the rebuttal revision PDF.
> > - #### Please find the change records for Rebuttal Revision from https://openreview.net/forum?id=nTNgkEIfeb&noteId=Xqdqp8TXdp

---

### Official Review · Reviewer_E5kp · 2023-10-31

**Soundness:** 3 good
**Presentation:** 3 good
**Contribution:** 3 good
**Rating:** 8
**Confidence:** 4

**Summary:**

This paper focuses on Federated Learning (FL), a privacy-enhancing distributed machine learning approach. It highlights a crucial issue: the potential for Model Inversion (MI) attacks in FL, where attackers can extract data from other participants. Existing defenses are inadequate. To address this, the authors propose FedInverse, which uses the Hilbert-Schmidt independence criterion (HSIC) to assess FL model vulnerability to MI attacks. Experiments with typical MI attackers confirm FedInverse's effectiveness in evaluating data leakage risks in FL systems.

**Strengths:**

+ A novel method called FedInverse is proposed to comprehensively evaluate the privacy risks of FL in response to model inversion attacks.
+ The research question is well defined and valuable to the research community.
+ A thorough and comprehensive case study.

**Weaknesses:**

- The diversity of the evaluated benchmark datasets needs to be further augmented.
- Potentially mature defense mechanisms require further consideration.

**Questions:**

This paper is well written and organized and has been thoroughly and comprehensively evaluated. However, the following minor issues still need to be considered:

- As mentioned in the weaknesses of this paper, we need to augment the diversity of the benchmark dataset. It would be better if the authors considered more tasks, such as NLP tasks.

- More advanced defense mechanisms need to be added to evaluate the effectiveness of the proposed attacks. Although the authors demonstrate the performance of the proposed attack against two common defense schemes in the appendix, the following stronger defenses still need to be considered:

[1] Huang Y, Gupta S, Song Z, et al. Evaluating gradient inversion attacks and defenses in federated learning[J]. Advances in Neural Information Processing Systems, 2021, 34: 7232-7241.

[2] Li J, Rakin A S, Chen X, et al. Ressfl: A resistance transfer framework for defending model inversion attack in split federated learning[C]//Proceedings of the IEEE/CVF Conference on Computer Vision and Pattern Recognition. 2022: 10194-10202.

The main reason is that the BiDO solution considered in this article is not a defense solution tailored for FL.

---

> ### Comment · Reviewer_E5kp · 2023-11-21
> **Rebuttal**
>
> Dear Authors,
>
> I have noticed that other reviewers have many concerns about this paper, and the authors should consider responding to their comments promptly so that we can make a better decision.
>
> Best regards,
> E5kp

---

> ### Author Response · Authors · 2023-11-21
> **Rebuttal for Weakness and Questions**
>
> > This paper is well written and organized and has been thoroughly and comprehensively evaluated. However, the following minor issues still need to be considered:
> > - As mentioned in the weaknesses of this paper, we need to augment the diversity of the benchmark dataset. It would be better if the authors considered more tasks, such as NLP tasks.
> > - More advanced defense mechanisms need to be added to evaluate the effectiveness of the proposed attacks. Although the authors demonstrate the performance of the proposed attack against two common defense schemes in the appendix, the following stronger defenses still need to be considered:
>
> >  [1] Huang Y, Gupta S, Song Z, et al. Evaluating gradient inversion attacks and defenses in federated learning[J]. Advances in Neural Information Processing Systems, 2021, 34: 7232-7241.
>
> >  [2] Li J, Rakin A S, Chen X, et al. Ressfl: A resistance transfer framework for defending model inversion attack in split federated learning[C]//Proceedings of the IEEE/CVF Conference on Computer Vision and Pattern Recognition. 2022: 10194-10202.
>
> >  The main reason is that the BiDO solution considered in this article is not a defense solution tailored for FL.
>
> - #### **Thank you for reviewing our paper and giving valuable comments. Due to the weaknesses related to the questions, we will mainly answer the questions here.**
>   #### In the main context, we have done our experiments on one large dataset CelebA, which is a face recognition dataset, and a small dataset MNIST, which is a handwriting digits dataset. In addition, We have done more experiments on other datasets, such as CIFAR-10 in **Appendix A.16 to A.19**, which consists of diverse classes representing airplanes, cars, birds, cats, deer, dogs, frogs, horses, ships, and trucks with 6,000 images of each class. We extend the diversity of the dataset in our answers, for example, the health dataset or other FL datasets suggested by Reviewer ```hL64``` and ```5eXT``` in **Appendix C, Figure 6 and Figure 8**. We are also planning to investigate the FL with NLP tasks because the privacy leakage issues in LLM attract the attention of the research community.
>
> - #### Thank you for your comments and suggestions. We will explain the difference between model inversion attacks, gradient inversion attacks ```[1]```, and split federated learning ```[2]```. The defense methods of gradient inversion attacks cannot be used in the model inversion attacks because of the following:
>   #### - model inversion attacks do not need to reconstruct original images from the **feature maps**, but gradients attack need. Therefore, gradient inversion attacks cannot be deployed on the clients because clients cannot obtain the gradient changes from the other clients.
>   #### - Based on the attack mechanism above, model inversion attacks can be Black-box settings where the attacker can launch the attack on the global models it received in FL. However, the gradient inversion attacks are White-box settings in which the gradient changes must be obtained by the attackers.
>   #### The attacks in split federated learning cannot be used in our setting because the defense methods proposed in ```[2]``` are to prevent the clients from using the activation to generate data from the model during its special training scheme as a White-box setting. However, FedInverse is to target the trained models as a Black-box setting. Therefore, the defense method presented in ```[2]``` is not suitable for our setting. We will add reviews of SFL defense methods in our discussion of defense.
>   #### In our future work, we will study and investigate more about the defense methods for MI attacks in FL.
>
> #### ```[1]``` Huang Y, Gupta S, Song Z, et al. Evaluating gradient inversion attacks and defenses in federated learning[J]. Advances in Neural Information Processing Systems, 2021, 34: 7232-7241.
>
> #### ```[2]``` Li J, Rakin A S, Chen X, et al. Ressfl: A resistance transfer framework for defending model inversion attack in split federated learning[C]//Proceedings of the IEEE/CVF Conference on Computer Vision and Pattern Recognition. 2022: 10194-10202.

---

> > ### Comment · Reviewer_E5kp · 2023-11-22
> > **Response to Authors**
> >
> > Thanks to the authors for their detailed and timely responses! The authors' responses have addressed most of my concerns, so I will maintain my score.

---

> > > ### Author Response · Authors · 2023-11-22
> > >
> > > #### Thank you for your valuable comments and suggestions, your suggestions are important to our future work.

---

### Official Review · Reviewer_hL64 · 2023-11-01

**Soundness:** 2 fair
**Presentation:** 2 fair
**Contribution:** 2 fair
**Rating:** 6
**Confidence:** 4

**Summary:**

- The paper studies model inversion (MI) attacks in federated learning settings, where malicious clients (participants) would apply existing model inversion techniques on the global model shared by the server. When performing MI attacks, the malicious clients would look like benign clients since they can still send model updates as such.
- To perform the attack, the malicious clients would train GAN models for model inversion, following prior work (GMI, KED-MI, VMI); a typical instantiation is to train GANs on public data with similar data distributions as the FL training setup and then optimize the generator for low log-likelihood (loss) under the shared global FL model.
- The paper also proposes a “diversity optimization” technique using the Hilbert-Schmidt independency criterion as a regularizer. The main idea is to encourage the GAN generated outputs to be more diverse, and the paper shows that this technique qualitatively improves the inverted data and quantitatively improves the attack success.
- The paper evaluates the proposed method on three datasets (CelebA, MNIST, and CIFAR-10) and shows that it can achieve high “attack accuracy” (measured by a separate model trained to tell apart generated vs real images) and low “Frechet inception distance” (which measures the similarity between real and fake images within the embedding space of a CNN). The experiments also show that the proposed HSIC regularizer was essential for obtaining high attack accuracy.

**Strengths:**

- The high-level idea of the paper is simple: with some modifications (HSIC regularizer in this case), existing model inversion attacks in the standard ML literature can easily carry over to federated settings, where the attackers are malicious clients and the target network is the global, shared model from the central server in FL.
- There is also originality in studying a new attack surface in FL that isn’t easily detectable from by the server (cf. attacks that require malicious clients to send out-of-distribution updates). The problem of interest is important and worth exploring.
- The paper is also reasonably easy to follow.

**Weaknesses:**

[W1] The use of public-data assisted generative models, particularly following prior work like GMI, KED-MI, or VMI, implicitly assumes that the client data distributions can be found in the public domain. However, in many practical FL settings, this assumption is often unwarranted. Many settings where FL is helpful—such as medical images across hospitals [1]—are often where the data distributions aren’t covered by publicly available data.

[W2] The experiments can be improved in terms of both quality and quantity.

- The paper directly takes a split of the federated training set as the “public data” for training the GANs. This in-distribution nature weakens the experimental support for the proposed techniques.
- The evaluated datasets are fairly small; there are large federated dataset such as iNaturalist [2] that would help make the results more convincing. Indeed, by looking at Figure 4(a), the reader may argue that the model-inverted examples do not look much alike the true examples; what happens if the dataset and the FL setup is scaled up (e.g. more clients, larger local batch sizes)?

[W3] The evaluation metrics can be made more rigorous.

- The bulk of the experimental results rely on having a good “evaluation classifier” which tells apart real vs generated (model-inverted) images and whose embeddings are good enough to give meaningful Frechet inception distances (Section 4.4). However, it is unclear how such a classifier was trained.
- The core issue is perhaps that the possible plausible deniability that the model-inverted images are indeed part of the training set. I.e. just because the evaluation classifier says the generated images are realistic, can we be sure that they actually are? Is the similarity provable? Again, Fig. 4(a) does not seem to support the use of proposed metrics.

[1] FLamby: Datasets and Benchmarks for Cross-Silo Federated Learning in Realistic Healthcare Settings. NeurIPS 2022 Datasets and Benchmark.  https://arxiv.org/abs/2210.04620

[2] https://www.tensorflow.org/federated/api_docs/python/tff/simulation/datasets/inaturalist

**Questions:**

- I would appreciate if the authors provide more visualization results of the generated samples, e.g. on CIFAR-10, as well as across different attack success rates.
- Consider using different citation commands `\citet` , `\cite`, etc. to make the formatting of in-text references consistent.

---

> ### Comment · Reviewer_hL64 · 2023-11-21
> **Looking forward to rebuttal**
>
> Dear authors, it appears on my side that there has not been a rebuttal to the reviews. Since the reviewer-author rebuttal/discussion will end on November 22nd AOE, this is a gentle reminder that to respond to reviews or comment that there won't be a rebuttal. Thank you.

---

> ### Author Response · Authors · 2023-11-21
> **Rebuttal for W1**
>
> #### **Thank you for your kind reminder and for taking the time to review our paper and provide valuable comments. Sorry for the late reply. We took some time to conduct experiments on the suggested dataset from the reviewers. We hope the answers below solve your concerns. If your concerns were not cleared, please let us know, and we'll try to clarify them further. We also hope you can re-evaluate our paper based on the answers and consider raising your score.**
>
> > [W1] The use of public-data assisted generative models, particularly following prior work like GMI, KED-MI, or VMI, implicitly assumes that the client data distributions can be found in the public domain. However, in many practical FL settings, this assumption is often unwarranted. Many settings where FL is helpful—such as medical images across hospitals [1]—are often where the data distributions aren’t covered by publicly available data.
>
> #### The lack of publicly available data is the current challenge for most model-inversion attackers and limits the usefulness of not only our proposed method but also most of the other public-data-assisted generative models, including GMI, KED-MI, and VMI. FedInverse uses very limited similar but different distributed public data to launch the model inversion attacks in our FL settings. This is more realistic in real-world applications because there are still available public data that have similar but not the same distribution to the targeted data from the global model. For example, the experiments in **Appendix A.2** have been conducted using FMNIST ```[1]```  and EMNIST ```[2]```  as the public datasets to generate the data from MNIST. Referring to the comments from Reviewer ```5eXT```, our work aims to evaluate the boundaries of privacy protection in the whole FL system from the client's perspective, whether the client can obtain data from other clients. From the reviewer's suggestions and comments, we can evaluate that the FL task can be considered secure if no public datasets can be leveraged by attackers. Therefore, evaluating privacy boundaries under model inversion attacks in FL is important for the FL research community.
>
> #### Following the reviewers' suggestions, we conducted the experiments and reviewed the feasibility of our FedInverse in the health domain by using the COVID-19 CT Scan Dataset ```[3]``` to reveal Fed-LIDC-IDRI in FLamby ```[4]```. The initial results have been updated to the rebuttal revision in **Appendix C, Figure 6**, which shows the performance of FedInverse on Fed-LIDC-IDRI in FLamby ```[4]``` using the COVID-19 CT Scan Dataset ```[3]``` as the public dataset. However, our FedInverse aims to evaluate the privacy leakage issue in FL. We need to review more literature in the medical image domain so that we can understand how privacy is leaked through reversed CT Scans. We will add the detailed results and the dataset in the Appendix with the final submission.
>
> #### ```[1]``` Han Xiao, Kashif Rasul, and Roland Vollgraf. Fashion-mnist: a novel image dataset for benchmarking machine learning algorithms. arXiv preprint arXiv:1708.07747, 2017
> #### ```[2]``` Gregory Cohen, Saeed Afshar, Jonathan Tapson, and Andre Van Schaik. Emnist: Extending mnist to handwritten letters. In 2017 international joint conference on neural networks (IJCNN), pp.2921–2926. IEEE, 2017.
> #### ```[3]``` Kaggle. Covid-19 ct scans. https://www.kaggle.com/datasets/andrewmvd/covid19-ct-scans/., 2019.
> #### ```[4]``` FLamby: Datasets and Benchmarks for Cross-Silo Federated Learning in Realistic Healthcare Settings. NeurIPS 2022 Datasets and Benchmark. https://arxiv.org/abs/2210.04620

---

> ### Author Response · Authors · 2023-11-21
> **Rebuttal for W2**
>
> > [W2] The experiments can be improved in terms of both quality and quantity.
> > - The paper directly takes a split of the federated training set as the “public data” for training the GANs. This in-distribution nature weakens the experimental support for the proposed techniques.
> > - The evaluated datasets are fairly small; there are large federated dataset such as iNaturalist [2] that would help make the results more convincing. Indeed, by looking at Figure 4(a), the reader may argue that the model-inverted examples do not look much alike the true examples; what happens if the dataset and the FL setup is scaled up (e.g. more clients, larger local batch sizes)?
>
> - #### As is recognized by the reviewer, current model inversion attacks cannot avoid using public datasets. However, in assessing the risks of privacy leakage in FL from clients, determining the privacy protection boundary remains essential and is indeed the primary contribution of our paper. In our evaluation, FedInverse can overcome this in-distribution nature caused by using public datasets. Take the experiments on MNIST by using EMNIST as a public prior dataset; for example, we use EMNIST of letter-images ("A-Z") to reverse MNIST digits-images ("0-4"). This means FedInverse can generate out-distribution images with high accuracy (see Appendix A.2.2--Table 4).
>
> - #### In our paper, we evaluated on both large and small datasets. For example, CelebA ```[1]``` is a very large dataset that contains 10,177 identities, 202,599 face images, and 5 landmark locations, 40 binary attribute annotations per image. The dataset of CIFAR-10 ```[2]``` and MNIST ```[3]``` provides an additional evaluation of small datasets. For FedInverse evaluation on iNaturalist, we spent time on data processing and training a newly designed MI attack model for iNaturalist privacy leakage evaluation purposes. However, considering the dataset size, volume of classes, and very large images, it will take time for the new attack model design and training purposes. Therefore, we will update the results in the Appendix once the experiment on iNaturalist is done.
>   #### In regard to the concerns of the reviewer, "model-inverted examples do not look much alike the true examples." The purpose of a model inversion attack is not to generate the exact same image that humans recognize. The biometric identity (e.g. face) can be successfully reconstructed to break into otherwise secure systems even if humans do not recognize the model-inverted examples look much alike the true examples ```[4]```. Therefore, in addition to demonstrating the inverted images, the extent of generated images leaking privacy is numerically evaluated by the evaluation classifier. A high accuracy of the classification results from the evaluation classifier means the generated images contain critical biometric identities that can be used to intrude on other systems. We will add one sentence in the main context to explain the privacy leakage issue for face recognition.
>   #### We also evaluated the impact of FL setup changes in our paper, including the number of clients and local batch sizes. In **Appendix A.3, A.4, A.10, A.11**, we evaluated the impact of FedInverse with different numbers of clients on both CelebA and MNIST datasets. In **Appendix A.5, A.6, A.12, A.13**, we evaluated the impact of FedInverse with different local batch sizes on both CelebA and MNIST datasets.
>
> #### ```[1]``` Ziwei Liu, Ping Luo, Xiaogang Wang, and Xiaoou Tang. Deep learning face attributes in the wild. In Proceedings of the IEEE international conference on computer vision, pp. 3730–3738, 2015.
> #### ```[2]``` Alex Krizhevsky, Geoffrey Hinton, et al. Learning multiple layers of features from tiny images. 2009.
> #### ```[3]``` Yann LeCun, L´eon Bottou, Yoshua Bengio, and Patrick Haffner. Gradient-based learning applied to document recognition. Proceedings of the IEEE, 86(11):2278–2324, 1998.
> #### ```[4]``` Kuan-ChiehWang, Yan Fu, Ke Li, Ashish Khisti, Richard Zemel, and Alireza Makhzani. Variational model inversion attacks. Advances in Neural Information Processing Systems, 34:9706–9719, 2021

---

> ### Author Response · Authors · 2023-11-21
> **Rebuttal for W3**
>
> > [W3] The evaluation metrics can be made more rigorous.
> >- The bulk of the experimental results rely on having a good “evaluation classifier” which tells apart real vs generated (model-inverted) images and whose embeddings are good enough to give meaningful Frechet inception distances (Section 4.4). However, it is unclear how such a classifier was trained.
> >- The core issue is perhaps that the possible plausible deniability that the model-inverted images are indeed part of the training set. I.e. just because the evaluation classifier says the generated images are realistic, can we be sure that they actually are? Is the similarity provable? Again, Fig. 4(a) does not seem to support the use of proposed metrics.
>
> - ####  The evaluation classifier is well-trained by using the whole dataset with a very high testing accuracy which achieved around 98%. We will add this claim in the main context.
> - #### The existing MI inversion research, including GMI ```[1]```, KED-MI ```[2]```, VMI ```[3]``` and BiDO ```[4]``` use the FID to measure the similarity between biometric identity. As we answered the questions from **Rebuttal for W2**, the biometric identity can be successfully reconstructed to break into otherwise secure systems even if humans do not recognize the model-inverted examples look much alike the true examples ```[3]```. Therefore, in addition to demonstrating the inverted images, the extent of generated images leaking privacy is numerically evaluated by the evaluation classifier. A high accuracy of the classification results from the evaluation classifier means the generated images contain critical biometric identities that can be used to intrude on other systems. We will add one sentence in the main context to explain the privacy leakage issue for face recognition.
>
> #### ```[1]``` Yuheng Zhang, Ruoxi Jia, Hengzhi Pei, Wenxiao Wang, Bo Li, and Dawn Song. The secret revealer: Generative model-inversion attacks against deep neural networks. In Proceedings of the IEEE/CVF conference on computer vision and pattern recognition, pp. 253–261, 2020.
>
> #### ```[2]``` Si Chen, Mostafa Kahla, Ruoxi Jia, and Guo-Jun Qi. Knowledge-enriched distributional model inversion attacks. In Proceedings of the IEEE/CVF international conference on computer vision, pp. 16178–16187, 2021.
>
> #### ```[3]``` Kuan-ChiehWang, Yan Fu, Ke Li, Ashish Khisti, Richard Zemel, and Alireza Makhzani. Variational model inversion attacks. Advances in Neural Information Processing Systems, 34:9706–9719, 2021
>
> #### ```[4]``` Xiong Peng, Feng Liu, Jingfeng Zhang, Long Lan, Junjie Ye, Tongliang Liu, and Bo Han. Bilateral dependency optimization: Defending against model-inversion attacks. In Proceedings of the 28th ACM SIGKDD Conference on Knowledge Discovery and Data Mining, pp. 1358–1367, 2022.

---

> ### Author Response · Authors · 2023-11-21
> **Rebuttal for Questions**
>
> >  - I would appreciate if the authors provide more visualization results of the generated samples, e.g. on CIFAR-10, as well as across different attack success rates.
> > - Consider using different citation commands \citet,  \cite, etc. to make the formatting of in-text references consistent.
>
> - #### Thanks for the reviewer giving the comments. Please see the experiment results on CIFAR-10 of the generated samples across different attack success rates shown in **Appendix C, Figure 7**.
> - ####  Thank you for your comments. We have double-checked the citation format. All the citations are using \citet{}.

---

> ### Comment · Reviewer_hL64 · 2023-11-21
> **Thank you for the rebuttal**
>
> I appreciate the authors for providing a detailed rebuttal, particularly the additional experimental results and pointers to experiments presented in the Appendix. I have read the responses and most of my concerns are resolved. I'm raising my score to 6, with a strong suggestion that the authors include the rebuttal discussions in the updated version.

---

> > ### Author Response · Authors · 2023-11-21
> > **We will include the rebuttal discussions in an updated version later.**
> >
> > The discussions you have contributed to this paper are very helpful. We will include rebuttal discussions in the updated version to make the final version more clear to our readers.
> > We are now adjusting the organization of this paper to include comments from other reviewers as well. Once we update a decent version that includes your discussion, we will let you know in the comments.

---

### Author Response · Authors · 2023-11-22
**Changes to the Rebuttal Revision**

#### Dear Reviewers, refer to the discussion in the rebuttal. We will list all the changes we have made in the rebuttal revision here.
#### **Main Context**
#### The content added is based on reviews:
- #### ```[hL64]``` (Section 3.2) Sentence "Even the inverted image cannot be completely alike the original image. However, the biometric identity (e.g. face) can be successfully reconstructed to break into otherwise secure systems even if humans do not recognize the model-inverted examples look much alike the true examples" has been added in Section 3.2
- #### ```[5eXT]``` (Section 6)The brief Related works have been added in Section 6
- #### ```[hL64]``` (Section 4.4) Sentence "The evaluation classifier is well-trained by using the whole dataset with a very high testing accuracy which achieved around 98%" is added in Section 4.4
- #### ```[5eXT]``` (Algorithm 1) We have made the changes to emphasize only one single attacker will join the FL in each communication round in Algorithm 1. In line 19, the set Q_t has been replaced by N, representing a Gaussian distribution.
- #### ```[Do36]``` ```[5eXT]``` (Introduction) We have made the changes in Introduction paragraph 4 as "Given that the data-leakage risk cannot be eliminated, we propose FedInverse, a novel privacy leakage evaluation method to evaluate the boundaries of privacy protection in the FL system from the participant’s perspective, whether one participant can obtain data from other participants. An attacker pretending to be a participant in FedInverse can conduct the Black-box attacks via global model prediction in each federated training round.", which emphasizes the target and setup of our paper.

#### **Appendix**

#### The content added is based on reviews:
- ####  ```[hL64][5eXT]``` (Appendix C) Experiment results using COVID-19 CT Scan Dataset to reveal Fed-LIDC-IDRI in FLamby have been added in Figure. 6 and Figure. 8
- #### ```[hL64]``` (Appendix C) Experiment results of FedInverse on CIFAR-10 across different attack accuracies (10.00%, 22.00%, and 30.00%) have been added in Figure. 7
- #### ```[5eXT]``` (Appendix B.2)The review of gradient attacks has been added in Appendix B.2.

#### **Reference**

- #### ```[1]``` Pasquini, D., Francati, D. and Ateniese, G., 2022, November. Eluding secure aggregation in federated learning via model inconsistency. In Proceedings of the 2022 ACM SIGSAC Conference on Computer and Communications Security (pp. 2429-2443).
- #### ```[2]``` Boenisch, F., Dziedzic, A., Schuster, R., Shamsabadi, A.S., Shumailov, I. and Papernot, N., 2023, July. Reconstructing Individual Data Points in Federated Learning Hardened with Differential Privacy and Secure Aggregation. In 2023 IEEE 8th European Symposium on Security and Privacy (EuroS&P) (pp. 241-257). IEEE.
- #### ```[3]``` Zhao, J.C., Sharma, A., Elkordy, A.R., Ezzeldin, Y.H., Avestimehr, S. and Bagchi, S., 2023, October. LOKI: Large-scale Data Reconstruction Attack against Federated Learning through Model Manipulation. In 2024 IEEE Symposium on Security and Privacy (SP) (pp. 30-30). IEEE Computer Society.

---

### Meta-Review · Area_Chair_HghW · 2023-12-08

**Metareview:**

This paper studies the problem of model inversion in FL. The authors identify a privacy vulnerability in FL, where the attacker can act as a benign participant and launch model inversion attack against other participants. The authors also propose a new HSIC regularizer that promotes diversity in existing inversion attacks. Evaluated on datasets such as CelebA, MNIST and CIFAR, the authors show that existing model inversion attacks can be highly successful at recovering private data of other participants.

While the initial results showed promise, reviewers recommended several ways for further improving the paper:
1. Experiment using larger scale datasets.
2. Analyzing the effect of distribution shift between private and public data.
3. More appropriate evaluation metrics for capturing privacy leakage.

The author rebuttal addressed most of these issues and the reviewers unanimously recommended acceptance in light of this.

**Justification For Why Not Higher Score:**

Technical contribution for the paper is on the weaker side, with the only novel component being the HSIC regularizer. The result being specific to FL means the paper's significance is not enough for a spotlight presentation.

**Justification For Why Not Lower Score:**

Reviewers unanimously recommended acceptance after the author rebuttal.

---

### Decision · Program_Chairs · 2024-01-16

Accept (poster)